# COMPASS: COntinual Multilingual PEFT with Adaptive Semantic Sampling

**Noah Flynn**                                                                      *noahflynn@berkeley.edu*
*UC Berkeley*

**Reviewed on OpenReview:** *https://openreview.net/forum?id=oapsbIO1Bd*

## Abstract

Large language models (LLMs) often exhibit performance disparities across languages, with naive multilingual fine-tuning frequently degrading performance due to negative cross-lingual interference. To address this, we introduce COMPASS (COntinual Multilingual PEFT with Adaptive Semantic Sampling), a novel data-centric framework for adapting LLMs to target languages. COMPASS leverages parameter-efficient fine-tuning (PEFT) by training lightweight, language-specific adapters on a judiciously selected subset of auxiliary multilingual data. The core of our method is a distribution-aware sampling strategy that uses multilingual embeddings and clustering to identify semantic gaps between existing training data and a target usage distribution. By prioritizing auxiliary data from under-represented semantic clusters, COMPASS maximizes positive cross-lingual transfer while minimizing interference. We extend this into a continual learning framework, COMPASS-ECDA, which monitors for data distribution shifts in production and dynamically updates adapters to prevent model staleness, balancing adaptation to new data with the preservation of existing knowledge. Across three different model architectures (Phi-4-Mini, Llama-3.1-8B, and Qwen2.5-7B) and multiple challenging multilingual benchmarks (Global-MMLU, MMLU-ProX), including unseen long-context tasks (OneRuler), we demonstrate that COMPASS consistently outperforms baseline methods guided by linguistic similarity, providing an effective, efficient, and sustainable solution for developing and maintaining high-performing multilingual models in dynamic environments.

## 1 Introduction

Large language models (LLMs) demonstrate remarkable capabilities across diverse natural language tasks, but extending these to multiple languages, especially low-resource languages (LRLs), remains challenging. State-of-the-art LLMs overfit to English, exhibiting noticeable biases (e.g., gender, race, caste, religion) and unusual behaviors with less-represented languages not seen during training (Khandelwal et al., 2023; Kotek et al., 2023; Vashishtha et al., 2023; Khondaker et al., 2023). Closed datasets monopolize LRLs (Longpre et al., 2024), while open datasets and NLP breakthroughs consolidate around a few data-rich languages (Lucy et al., 2024; Zhang et al., 2022). The scarcity of openly available, high-quality, human-generated non-English data widens the gap for under-represented languages (Longpre et al., 2024; Lucy et al., 2024; Zhang et al., 2022; Sun et al., 2023; Zampieri et al., 2017). This data scarcity contributes to broader systemic issues in multilingual NLP, including cross-lingual security vulnerabilities (Yong et al., 2024; Deng et al., 2024), tokenization inefficiencies for non-Latin scripts (Cui et al., 2024; Ji et al., 2023), privacy risks (Lukas et al., 2023; Li et al., 2024), and cultural mismatches (e.g., overemphasizing Western-centric concepts) or translation artifacts that arise when machine-translating English datasets (Vanmassenhove et al., 2021; Bizzoni et al., 2020; Chen et al., 2025). While these challenges exacerbate inequities in global AI deployment, they are diverse and multifaceted problems requiring distinct solutions. This work focuses on a specific but fundamental challenge underlying poor LRL performance: distributional mismatch between training data and real-world usage patterns.

Fine-tuning on multilingual corpora for language adaptation offers a path to reducing linguistic inequality, but naive inclusion of data across multiple languages often degrades performance on the target language (Wang et al., 2020b). This negative interference stems from distributional misalignment: multilingual model training data often differs from real-world usage patterns (i.e., live production interactions) due to sampling biases, topical variations, or imbalanced language contributions. When training data over-exposes models to irrelevant patterns and under-exposes them to crucial ones for the target language, the model learns suboptimal representations. While linguistic differences (syntax, vocabulary, semantics) and cultural factors (translational ambiguity, localization nuances) contribute to cross-lingual interference, the fundamental issue that COMPASS targets is that languages "compete" for model capacity when their training distributions do not align with their respective usage distributions. Consequently, monolingual models often outperform massively multilingual models on language-specific tasks (Mistral, 2025; Pires et al., 2023; Cañete et al., 2023; Sarti & Nissim, 2022; Martin et al., 2019; Chan et al., 2020; Lee et al., 2021; Nguyen & Nguyen, 2020).

A straightforward alternative is tuning individual models for each language using only monolingual data. However, this one-model-per-language approach incurs large memory overhead, and diverse, human-curated task data is scarce for many languages, especially those lacking sufficient native or fluent speaker involvement in data creation. Discarding all non-target language data avoids interference but sacrifices the benefits of cross-lingual transfer, where related language data can yield synergistic improvements.

A key research question emerges: can we selectively incorporate cross-lingual data to achieve positive transfer for a target language while avoiding harmful interference? We posit the answer is yes – if additional data are strategically chosen based on distributional similarity to the target language's requirements. In practice, multilingual model training data often differs distributionally from real-world usage data (i.e., live production interactions) due to sampling biases, topical variations, or imbalanced language contributions. This mismatch can over-expose models to irrelevant patterns and under-expose them to crucial ones for the target language. Addressing this distributional mismatch is crucial for effective cross-lingual transfer and can be achieved while still incorporating cross-lingual data for positive transfer.

We introduce **CO**ntinual **M**ultilingual **P**EFT with **A**daptive **S**emantic **S**ampling (COMPASS), a novel data sampling strategy to adapt pre-trained language models to specific target languages. COMPASS does not directly tackle tokenization inefficiencies, safety vulnerabilities, or inherent linguistic structure differences – these remain important open problems. Instead, COMPASS operates on the principle that better coverage of the target distribution, achieved through semantically-guided auxiliary data selection, can improve model performance by reducing task-irrelevant noise and ensuring exposure to under-represented usage patterns. This improved distributional alignment may indirectly reduce some symptoms of linguistic interference (e.g., by filtering irrelevant cross-lingual examples), but the method is fundamentally task-distribution-driven rather than linguistically-informed.

Recognizing that language use and data distributions are dynamic in real-world settings (e.g., new jargon, shifting user preferences or user pools, seasonality), COMPASS is designed both for effective initial adaptation and subsequent continual learning. We use parameter-efficient fine-tuning with weight-decomposed low-rank adaptation (DoRA) (Liu et al., 2024), enabling a single shared base model with lightweight, language-specific adjustments. This facilitates multi-adapter deployment without training a full model per language. Our core idea is to fine-tune a DoRA adapter for each target language using a judiciously selected subset of multilingual data. This selection is guided by analyzing the target language's expected "live" usage distribution, approximated using a proxy development set with simulated biases.

In production, an initial language adapter may perform well initially, but its performance can degrade over time as incoming queries diverge from the offline training distribution. This is particularly relevant when bootstrapping models for LRLs in production, where initial user data for emerging markets is scarce, and model deployment is expected to generate more informative data for improvement. We propose a continual learning extension that periodically updates adapters with fresh data to match distribution shifts observed in incoming user requests. We simulate distribution shift scenarios (e.g., a sudden influx of queries on a new, challenging topic) and demonstrate that our adaptive retraining procedure recovers performance on new data while preserving it on original data. This transforms COMPASS from a one-time fine-tuning technique

into a sustainable, long-term solution for multilingual model maintenance, fostering continuous improvement through real-world usage.

Our contributions are: (1) a novel distribution-aware sampling strategy for multilingual model adaptation, using clustering and embedding-based distribution comparison to guide cross-lingual data selection; (2) application of this strategy to train adapters for multiple languages, achieving strong empirical gains on multilingual, human-generated benchmarks across various architectures with minimal negative transfer; (3) extensive experiments, including studies on language affinity, data budget optimization, cross-task generalization, and component-wise ablations to validate and analyze our approach; (4) an extension of the method to adapt to distribution shifts over time.

## 2 Related Works

Our work is situated at the confluence of several research directions: mitigating negative cross-lingual transfer, parameter-efficient model adaptation, dynamic data selection, and continual learning.

### 2.1 Mitigating Negative Cross-Lingual Transfer

A central challenge in building capable multilingual systems is the "curse of multilinguality". While multilingual models are trained on a vast number of languages, they exhibit degraded performance on specific, especially lower-resource, languages when compared to their monolingual counterparts, underscoring the need for more effective, language-specific adaptation strategies (Xu et al., 2025). This phenomenon arises from an implicit competition among languages for a finite set of shared model parameters, which can lead to destructive interference between linguistic representations and suboptimal performance for any single language (Wang et al., 2020b).

A range of strategies has been developed to mitigate interference, including interventions in the optimization process. Methods like Project Conflicting Gradients (PCGrad) (Yu et al., 2020) intervene during training to project gradient updates into a space where they do not conflict between languages. More recently, CONGRAD (Li et al., 2025) operationalizes this principle as a data selection strategy for multilingual preference alignment. It computes an aggregated cross-lingual gradient direction and filters the training data to retain only those samples whose individual gradients show high cosine similarity with this global update vector. This approach is precise, but its reliance on gradient computation makes the selection process computationally intensive.

Shifts towards architectural solutions that move away from monolithic, one-size-fits-all models represent more modular or specialized architectures. The Cross-lingual Expert Language Models (X-ELM) framework, for instance, mitigates parameter competition by training multiple, smaller "expert" models, each specialized on a distinct cluster of languages (Blevins et al., 2024). These clusters can be formed using either supervised signals like linguistic typology or unsupervised methods based on TF-IDF text features, partitioning the linguistic load to reduce interference. Another architectural approach, XTransplant, proposes a dynamic, inference-time modification by "transplanting" specific model components, such as the feed-forward network (FFN) layers known to store factual knowledge, from a strong source-language context (e.g., English) to a weaker target-language context (Ye et al., 2025). These architectural solutions validate our premise that specialization is key to overcoming the curse of multilinguality. However, they introduce significant overhead. X-ELM requires training and storing multiple independent models, each of equivalent size to a seed language model, while XTransplant involves an expensive search process at inference time.

Our work, COMPASS, embraces the principle of specialization via proactive, distribution-based selection with considerations towards efficiency. By leveraging lightweight, language-specific adapters on a single shared base model, we avoid substantial storage and training costs of full model ensembles. This architectural choice positions COMPASS as a data-centric alternative that is more direct than representation alignment methods, which often rely on parallel corpora (Liu & Niehues, 2025; Zhao et al., 2025), and more efficient than reactive gradient-based filtering.

## 2.2 Parameter-Efficient Adaptation for Multilingualism

Parameter-Efficient Fine-Tuning (PEFT) has emerged as the standard methodology for adapting large pre-trained models with minimal computational and memory overhead (Aggarwal et al., 2024). Methods like Low-Rank Adaptation (LoRA) can match full fine-tuning performance while updating a fraction of the parameters (Razuvayevskaya et al., 2024). Our work specifically builds upon Weight-Decomposed Low-Rank Adaptation (DoRA), an evolution of LoRA that decouples the magnitude and direction of weight updates to enable more stable and effective training.

The application of PEFT to multilingualism enables efficient language specialization on a shared base model. The MAD-X framework introduced a modular approach with separate, stackable 'language adapters' and 'task adapters', allowing for parameter-efficient transfer to new languages and tasks (Pfeiffer et al., 2020). Similarly, Franken-Adapter proposes 'embedding surgery', where customized vocabularies are created for target languages and only the embedding layer is tuned before being integrated with an instruction-tuned base model (Jiang et al., 2025). Mix-of-Language-Experts (MoLE) architecture formalizes this approach for multilingual programming tasks by jointly optimizing a shared LoRA module for common knowledge alongside a collection of programming language-specific LoRA modules (Zong et al., 2025).

The success of these modular, parameter-efficient architectures has shifted the research bottleneck from a question of *if* we can efficiently create language-specific modules to the question of *how* we should train them for optimal performance. These architectural solutions do not inherently prescribe a strategy for selecting data from a heterogeneous, multilingual pool to train each language adapter. We posit that naively fine-tuning adapters on all available data is suboptimal. Instead, COMPASS provides a distribution-guided method to construct a judiciously selected training subset for any language-specific PEFT module – be it a full adapter in MAD-X or a new embedding layer in Franken-Adapter. By doing so, COMPASS enhances the entire ecosystem of modular multilingual models, maximizing positive cross-lingual transfer while actively minimizing interference.

## 2.3 Data Selection for Cross-Lingual Fine-Tuning

Data quality over quantity is key to PEFT-based multilingual adaptation (Liu et al., 2025). Data selection strategies range from static, heuristic-based filtering to dynamic, model-in-the-loop frameworks. Our work falls into the latter category, using model-internal signals to guide selection.

A foundational line of work uses model training dynamics to score data. Dataset Cartography, for instance, uses the model's confidence and prediction variability across epochs to map a dataset into regions of "easy-to-learn," "hard-to-learn," and "ambiguous" examples (Swayamdipta et al., 2020). This map serves as a diagnostic tool, revealing that ambiguous examples are often crucial for out-of-distribution generalization. Other methods like Variance of Gradients (VoG) and EL2N use gradient variance or error norms during training to identify challenging or easy examples (Agarwal et al., 2022). While powerful, these methods require at least a partial training run to derive their scores, posing a scalability challenge in a multi-adapter setting where one would need to repeat the process for each target language.

More advanced frameworks learn a dynamic sampling policy. Differentiable Data Selection (MultiDDS) learns a data sampling policy by optimizing for gradient alignment between sampled training data and a small, trusted validation set (Wang et al., 2020a), while Mixture-of-Skills (MoS) uses reinforcement learning to adjust sampling probabilities across pre-defined datasets ("skills") based on a set of hand-crafted reward heuristics, such as inter-dataset similarity and learning difficulty (Wu et al., 2024). MultiDDS optimizes for an indirect proxy of generalization (gradient alignment), while MoS optimizes a set of heuristic rewards that may not be robust across all data types.

In contrast, COMPASS formulates cross-lingual data selection as a problem of unsupervised domain adaptation, where the goal is to minimize the distributional mismatch between the training data and a target usage distribution. This aligns our work with prior methods that use embedding-based clustering for distribution matching, but COMPASS's novelty lies in its specific focus on identifying and filling under-represented semantic clusters to guide cross-lingual transfer. It discovers latent topics via unsupervised clustering rather than requiring pre-defined datasets (unlike MoS), and it uses the magnitude of the distributional gap itself as

the primary, non-heuristic signal for selection (unlike MultiDDS or CONGRAD). Furthermore, COMPASS unifies the goals of selecting for both similarity and diversity. By prioritizing under-represented clusters, it inherently seeks semantic diversity; by prioritizing prototypical examples within those clusters, it ensures topical similarity and relevance. As data coverage increases, it gradually introduces more ambiguous examples to enhance learning (Sorscher et al., 2023).

### 2.4 Continual Learning for Adapting to Distribution Shifts

A one-time adaptation is insufficient for real-world deployment, where data distributions evolve over time due to new topics, shifting user needs, or seasonality. Models must adapt to new data without suffering from "catastrophic forgetting", which entails degradation of performance on previously learned knowledge. Rehearsal-based methods, which replay a small buffer of past data, and regularization-based methods like Elastic Weight Consolidation (EWC), penalize changes to parameters deemed important for past tasks. Recent work has also explored more advanced signals, such as leveraging model uncertainty to guide data balancing, as seen in MultiUAT for machine translation (Wu et al., 2021), or employing causal frameworks to improve robustness (Wang & Huang, 2025).

A significant portion of continual PEFT research focuses on task-incremental learning, where a model learns a sequence of distinct tasks. Architectural methods, such as those that train separate PEFT modules for each task and compose them with a router (Araujo et al., 2024), are designed for this. While powerful, their core mechanisms (e.g., task-specific routing or parameter subspaces) are suitable to task-incremental learning and are not applicable to our problem. COMPASS tackles domain-incremental learning, where the task remains constant (e.g., instruction-following for a specific language), but the underlying data distribution shifts. Other architectural innovations like CURLoRA (Fawi, 2024), which grounds the adapter update in the original pre-trained weights via CUR decomposition, are less suited for continual adaptation where preserving knowledge from the immediately preceding state is more critical than preserving knowledge from the initial pre-trained state.

## 3 Distribution-Guided Sampling for Multilingual Adaptation

Our approach focuses on selecting relevant cross-lingual data that mirrors the target language's usage patterns, enriching the target language's training data with this auxiliary data. The core insight of our approach is that negative cross-lingual transfer occurs because the distribution of source language data differs from the target language's usage distribution. We address this by selectively sampling auxiliary language data that aligns with the target language's distribution gaps.

### 3.1 Problem Definition

We address the task of adapting a pre-trained multilingual language model to a specific target language such that its performance on that language is maximized, leveraging auxiliary data from other languages. Formally, assume we have a base model $M_{\text{base}}$ (with parameters $\Theta$) pretrained on many languages. We define a target language $\ell_t$ for which we have a set of fine-tuning data $D_t = \{(x_i, y_i)\}_{i=1}^{N_t}$ and evaluation data $E_t = \{(x_j^{(eval)}, y_j^{(eval)})\}_{j=1}^{M_t}$ which represents the model's intended usage or test distribution in language $\ell_t$. Additionally, we have a pool of auxiliary training data $D_{\text{aux}} = \bigcup_{\ell \neq \ell_t} D_\ell$ comprising data from a set of other languages (and possibly including more data from $\ell_t$ itself beyond $D_t$). Our goal is to use $D_t$ and a suitable subset of $D_{\text{aux}}$ to fine-tune the model such that performance on $E_t$ is maximized. Through a PEFT-based approach, we train a small set of additional parameters $\phi_t$ (the adapter for language $\ell_t$) while keeping $\Theta$ fixed, thus $M_{\text{adapted}}(\cdot; \Theta, \phi_t)$ is the adapted model for language $\ell_t$.

The distribution of $D_t$ (what the model sees during fine-tuning for $\ell_t$) may differ from the distribution of $E_t$ (what the model is evaluated on). In many cases, $D_t$ might be relatively small or collected in a certain manner, whereas $E_t$ (or actual user inputs in $\ell_t$) might cover a broader or different variety of content. We denote by $P_{\text{train}}(z|\ell_t)$ the distribution over inputs $z$ in the fine-tuning data for $\ell_t$, and by $P_{\text{eval}}(z|\ell_t)$ the distribution over inputs in the evaluation set (or actual use) for $\ell_t$. Our approach assumes we can use $E_t$ as a proxy for the latter (even if $y_j^{(eval)}$ labels are not used in training, the $x_j^{(eval)}$ give an idea of what

content is important). The challenge is then that $P_{\text{train}}(z|\ell_t)$ may not adequately cover regions of $P_{\text{eval}}(z|\ell_t)$. Conversely, the auxiliary data pool $D_{\text{aux}}$ (aggregated from many languages) is typically much larger and more diverse; it likely covers $E_t$'s domains, but indiscriminately adding all of $D_{\text{aux}}$ to training could introduce irrelevant regions that are not in $P_{\text{eval}}(z|\ell_t)$, thereby causing negative transfer. We need to find a subset $S \subseteq D_{\text{aux}}$ such that training on $D_t \cup S$ best approximates training on data drawn from $P_{\text{eval}}(z|\ell_t)$.

### 3.1.1 Distribution Mismatch

We quantify distribution mismatch by dividing the input space into regions and comparing the density of target data vs. auxiliary data in those regions. Let $\mathcal{Z}$ denote the space of input examples (e.g., the space of all possible text prompts). We assume the existence of a feature mapping $f : \mathcal{Z} \to \mathbb{R}^d$ that produces a meaningful vector representation (embedding) of an input. Ideally, in this feature space, texts with similar semantic content will be close together regardless of language. We do not require $f$ to be perfectly language-agnostic, but it should cluster data by topic/task more strongly than by language. In practice, $f$ will be a pretrained multilingual encoder that is not fine-tuned on our task labels, so $f$ captures general semantic information. Given this representation, we cluster the combined set of embeddings of $D_t \cup D_{\text{aux}} \cup E_t$. Let $\{C_1, C_2, \ldots, C_K\}$ be $K$ clusters in the embedding space. These clusters represent regions of semantically related content. For each cluster $C_k$, define:

- $n_t^k = |\{x_i \in D_t : f(x_i) \in C_k\}|$, the number of target-language training examples in cluster $k$.

- $n_{\text{aux}}^k = |\{x \in D_{\text{aux}} : f(x) \in C_k\}|$, the number of auxiliary examples in cluster $k$.

- $n_{\text{eval}}^k = |\{x_j^{(eval)} \in E_t : f(x_j^{(eval)}) \in C_k\}|$, the number of eval examples in cluster $k$ (our proxy for true usage in that cluster).

We then approximate the cluster-level distributions: $P_{\text{train}}(C_k|\ell_t) \approx \frac{n_t^k}{\sum_{k'} n_t^{k'}}$, and $P_{\text{eval}}(C_k|\ell_t) \approx \frac{n_{\text{eval}}^k}{\sum_{k'} n_{\text{eval}}^{k'}}$. We measure mismatch in cluster $k$ as the ratio of these: $\rho_k = \frac{P_{\text{eval}}(C_k|\ell_t)}{P_{\text{train}}(C_k|\ell_t)}$. If $\rho_k > 1$, cluster $k$ is under-represented in the training data relative to what the eval distribution would suggest (i.e., there's a "hole" in training coverage). If $\rho_k < 1$, the training set has relatively more data in $C_k$ than the eval distribution does (possibly we are over-emphasizing that region). Our aim is to adjust the training distribution by adding samples from $D_{\text{aux}}$ to better match $P_{\text{eval}}$. Specifically, we want to sample more heavily from clusters with $\rho_k > 1$.

We formalize the selection problem as finding a sampling distribution $Q$ over $D_{\text{aux}}$ (plus using all of $D_t$ by default) such that for each cluster $C_k$, the effective training probability $P_{\text{train+aux}}(C_k|\ell_t)$ moves closer to $P_{\text{eval}}(C_k|\ell_t)$. However, we also must respect a limit on how much auxiliary data we add (for efficiency and to avoid swamping the target data). Let $B$ denote a budget factor (e.g., $B = 1.0$ means we will add an amount of auxiliary data equal to $|D_t|$, $B = 2.0$ means double the target data size in auxiliary examples, etc.). We then require $\sum_{x \in D_{\text{aux}}} Q(x) = B \cdot |D_t|$. Our strategy will construct $Q$ in two stages: first allocate weights to clusters, then within each cluster allocate to individual examples.

## 3.2 Initial Adaptation Phase

Illustrated in figure 1, COMPASS includes the following process: (1) obtain a semantically meaningful, multilingual embedding-based representation of all data, (2) estimate distributional mismatches between the target language's training data and usage data (approximating the usage distribution with held-out data as an initial proxy), (3) compensate for this mismatch by ranking clusters according to the degree of mismatch, (4) weigh remaining instances with importance scores based on literature precedence, targeting prototypical or "easy" examples when there is high mismatch and targeting more "ambiguous" examples as mismatch decreases while accounting for diversity by discounting highly similar auxiliary pairs, (5) use the cluster-level and instance-level weights to augment target language training data with auxiliary language data, (6) fine-tune language-specific adapters for a shared based model, (7) route incoming queries to the appropriate adapter with a language detection model at inference-time. Algorithm 1 provides a formal summary of the core COMPASS sampling procedure.

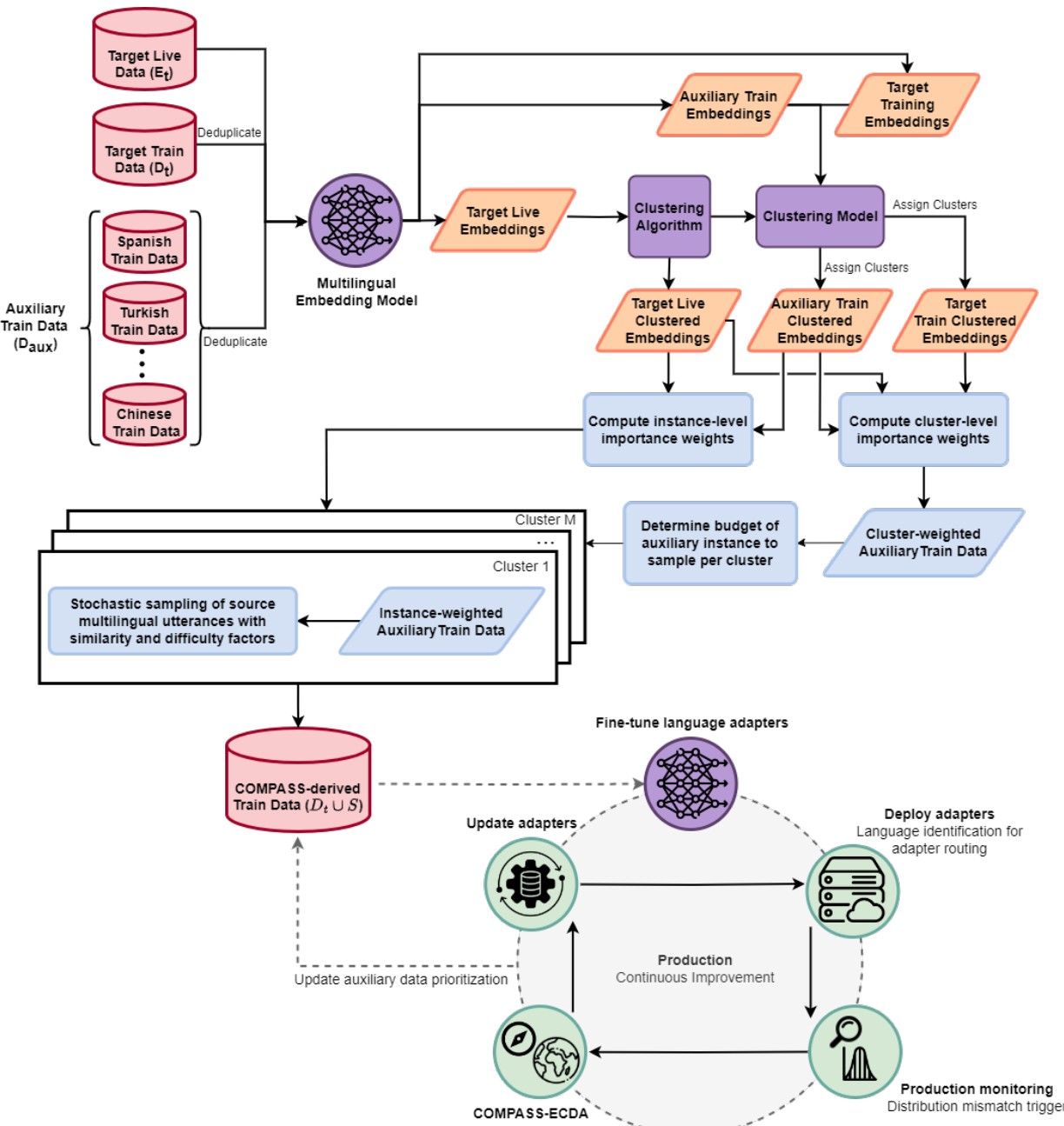

Figure 1: An overview of COMPASS for multilingual adaptation. (Top left) Data, including target language training data, a proxy for live usage data, and an auxiliary multilingual data pool, are converted into semantic representations by a multilingual embedding model. (Top right) We cluster the data into semantic groups, which are used to compute cluster-level and instance-level importance weights that guide the stochastic sampling of auxiliary data to address distributional gaps. (Bottom) Lightweight, language-specific adapters are fine-tuned and deployed, with an identification model routing incoming queries. The COMPASS-ECDA extension provides a continual learning loop that monitors for distribution shifts, triggering data re-sampling and adapter updates to maintain performance.

### 3.2.1 Embedding and Clustering

We use a pretrained multilingual embedding model $f(\cdot)$ to encode each candidate training example (both in $D_t$ and $D_{\mathrm{aux}}$) and each evaluation example in $E_t$. We then run a clustering algorithm on these embeddings; we explore several: K-means clustering, hierarchical agglomerative clustering, Taylor-Butina clustering, and HDBScan. The result is $K$ clusters $\{C_k\}_{k=1}^K$, where each cluster can be thought of as a topic or latent facet of the data (for example, one cluster might group legal/philosophical questions, etc., often with a mix of languages present).

### 3.2.2 Cluster-level Weighting

Once clusters are formed, we calculate the target-vs-eval mismatch for each cluster $k$. Since we always include the target's own data $D_t$ in training, the simplest way to compensate a deficit in cluster $k$ is to include more auxiliary examples from that cluster. We assign a cluster weight $w_k$ for sampling proportional to the mismatch ratio $\rho_k$ (or some monotonic function of it). In practice, we define:

$$w_k = \begin{cases} \frac{n_{\mathrm{eval}}^k}{n_t^k + \epsilon}, & \text{if } n_{\mathrm{eval}}^k > 0, \\ \\ 0, & \text{if } n_{\mathrm{eval}}^k = 0, \end{cases}$$

where $\epsilon = 1.0$ is a smoothing constant to avoid division by zero. This essentially says: if a cluster has eval examples but no (or few) target training examples ($n_t^k$ small), $w_k$ will be large, signaling we badly need examples of that type. If a cluster has no presence in eval ($n_{\mathrm{eval}}^k = 0$), we ideally don't want to sample from it at all (hence $w_k = 0$). If both target and eval have proportional presence, $w_k \approx 1$, meaning no special up-weighting needed (we would include auxiliary roughly in proportion). We then normalize these weights to get a probability distribution over clusters for auxiliary sampling: $\tilde{w}_k = \frac{w_k}{\sum_{j=1}^K w_j}$. This importance sampling scheme reshapes the auxiliary data distribution to closer match the eval distribution of clusters.

### 3.2.3 Practical Strategies for Bootstrapping $E_t$

In our experiments, we use held-out evaluation sets (dev sets of Global-MMLU and MMLU-ProX) as proxies for the live usage distribution $E_t$. This approximation is pragmatic for research but raises practical questions for real-world deployment, particularly in "cold-start" scenarios where a model is being adapted to a new language without extensive prior user data.

For cold-start deployment, a practical approach is to collect a few hundred representative "seed" examples through human curation. $E_t$ must span the anticipated semantic space of user queries, as COMPASS's distribution-aware sampling requires diversity across semantic clusters rather than volume within clusters. In the extreme case where no proxy is available, setting $E_t = D_t$ (using the training data itself as the usage proxy) provides a baseline. This reduces COMPASS to an approximately uniform-sampling regime in the initial phase, as cluster weights become nearly equal when train and eval distributions are identical ($\rho_k \approx 1$ for all clusters). While this eliminates distributional guidance initially, it does not harm performance relative to target-only training and allows the COMPASS-ECDA extension to refine $E_t$ over time based on observed usage. Alternatively, for languages within well-represented families and overlapping usage within defined geographic locales, usage distributions from related languages can serve as initial proxies (e.g., Spanish for Catalan within Spain), leveraging the multilingual embedding space's cross-lingual semantic similarities to borrow distributional knowledge from higher-resource relatives.

### 3.2.4 Instance-level Weighting & Sampling

Within each cluster $C_k$, not all examples are equally useful. Intuitively, examples very close to the cluster's centroid or densest region might be more representative of that cluster's theme, whereas those on the fringe might be less relevant or outliers. Thus, we score each candidate $x \in C_k$ from the auxiliary pool.

One simple choice is the inverse distance from the cluster centroid: $s(x) = \frac{1}{1 + \mathrm{dist}(f(x), \mu_k)}$, where $\mu_k$ is the centroid of cluster $k$ in embedding space and $\mathrm{dist}(\cdot, \cdot)$ uses cosine similarity with normalized embeddings.

This $s(x)$ gives higher weight to points near the center of the cluster. We would then probabilistically sample examples from cluster $k$ in proportion to $s(x)$ until we fulfill that cluster's quota. The cluster's quota of examples is determined by the cluster weight and overall budget $B$: we want $q_k = (\lfloor \tilde{w}_k \rfloor + \tilde{r}_k) \times B \times |D_t|$ examples from cluster $k$. $\tilde{r}_k$ is a random variable that is 1 with probability $\tilde{w}_k - \lfloor \tilde{w}_k \rfloor$, 0 otherwise.

Based on work in Sorscher et al. (2023), computing instance weights using the distance of each instance's embedding to its cluster decision boundary serves as a proxy for example difficulty. Following their results, it is advantageous to prioritize sampling of easier instances when there is little training data for the target language cluster. As the fraction of sampled data increases, it becomes more informative to expand sampling to more ambiguous instances further from the centroid, in the limit that instances near the decision boundary are more likely to be irrelevant or misannotated.

Thus, within each cluster $C_k$, we employ an adaptive weighting scheme that dynamically adjusts sampling preferences based on the cluster's distribution mismatch. Our core insight is that when facing severe underrepresentation (high $\rho_k$), we should prioritize prototypical examples that clearly represent the cluster's semantic content. As the mismatch decreases through sampling, we progressively favor more challenging boundary cases to improve model robustness, while remaining conservative about boundary examples since approximately 20% of examples closest to decision boundaries are expected to be hard-to-learn due to difficulty or misannotations (Swayamdipta et al., 2020).

For each candidate $x \in C_k$ from the auxiliary pool, we compute two complementary scores:

**Prototypical score**: $s_{\text{proto}}(x) = \frac{1}{1+\text{dist}(f(x), \mu_k)}$, where $\mu_k$ is the cluster centroid. This favors examples near the cluster center.

**Boundary score**: $s_{\text{boundary}}(x) = \min\{\text{dist}(f(x), \mu_j) : j \neq k\} - \text{dist}(f(x), \mu_k)$, normalized to $[0, 1]$. This favors examples near decision boundaries.

The final instance weight interpolates between these scores based on the current sampling progress, with a conservative skew towards prototypical examples:

$$s(x) = (1 - \alpha_k^2) \cdot s_{\text{proto}}(x) + \alpha_k^2 \cdot s_{\text{boundary}}(x) \tag{1}$$

where $\alpha_k = \min\left(1, \frac{\text{sampled\_count}_k}{\text{target\_quota}_k}\right)$ represents the fraction of the cluster's quota already fulfilled. The quadratic term $\alpha_k^2$ ensures that boundary examples receive reduced influence throughout the sampling process, with prototypical examples maintaining dominance until the final stages of sampling. Initially ($\alpha_k \approx 0$), we heavily favor prototypical examples. As sampling progresses ($\alpha_k \to 1$), we more cautiously incorporate boundary cases, implementing a conservative curriculum from easy to hard examples.

Given the multilingual nature of our auxiliary data pool, containing parallel or near-parallel examples across 50+ languages, redundancy within clusters can be substantial. To ensure diverse sampling while maintaining relevance to the target language, we penalize scores of instances in the same cluster that are above a similarity threshold. We construct a similarity matrix $Sim$ for all auxiliary examples within each cluster, where each entry $Sim_{i,j}$ quantifies the cosine similarity between a pair of embeddings. We establish a similarity threshold $\tau_{sim}$ (empirically set to 0.90) and derive an adjacency matrix $A$ where $A_{i,j} = 1$ if $Sim_{i,j} > \tau_{sim}$, and 0 otherwise. This formulates an undirected graph where vertices represent auxiliary instances and edges connect highly similar pairs. If an instance is sampled, we subtract a fixed penalty $\delta$ from the score $s(x')$ of any remaining neighbors $x'$.

This process yields up to $q_k$ examples from cluster $k$. In practice, some clusters might have fewer candidates than $q_k$, requiring further modifications to enforce $\sum_k |S_k| = B|D_t|$. We used a large pool of auxiliary data sufficient to avoid this issue. To mitigate this issue in settings with insufficient auxiliary data, one strategy would be to sample with replacement but institute a decay factor of $0.5^n$ after an example has been sampled $n$ times, effectively limiting each instance to at most 3 selections.

### 3.2.5 Fine-Tuning with Augmented Data

The final selected training set for language $\ell_t$ is $D_t \cup S$, where $S = \bigcup_k S_k \subset D_{\text{aux}}$. By construction, $|S| \approx B|D_t|$. In our experiments, we set $B$ in the range 0.2 to 2.0 (so the auxiliary data is at most double

---

**Algorithm 1** COMPASS: Distribution-Guided Auxiliary Data Sampling

---

**Require:** Target language data $D_t$, auxiliary data pool $D_{\mathrm{aux}}$, usage proxy $E_t$, budget $B$, embedding model
$f(\cdot)$
**Ensure:** Selected auxiliary set $S$
 1: **// Step 1: Embed and Cluster**
 2: Embed all data: $\mathcal{E} \leftarrow \{f(x) : x \in D_t \cup D_{\mathrm{aux}} \cup E_t\}$
 3: Cluster embeddings into clusters: $\{C_1, C_2, \ldots, C_K\} \leftarrow \mathrm{HDBScan}(\mathcal{E})$
 4:
 5: **// Step 2: Compute Cluster Weights**
 6: **for** each cluster $k = 1$ to $K$ **do**
 7: $\quad n_t^k \leftarrow |\{x \in D_t : x \in C_k\}|$ $\hfill \triangleright$ Target data count in cluster $k$
 8: $\quad n_{\mathrm{eval}}^k \leftarrow |\{x \in E_t : x \in C_k\}|$ $\hfill \triangleright$ Usage proxy count in cluster $k$
 9: $\quad w_k \leftarrow \begin{cases} \frac{n_{\mathrm{eval}}^k}{n_t^k + \epsilon} & \text{if } n_{\mathrm{eval}}^k > 0 \\ 0 & \text{otherwise} \end{cases}$ $\hfill \triangleright$ Mismatch ratio
10: **end for**
11: Normalize: $\tilde{w}_k \leftarrow w_k / \sum_{j=1}^K w_j$ $\hfill \triangleright$ Cluster sampling probabilities
12:
13: **// Step 3: Sample Auxiliary Data with Instance Weights**
14: $S \leftarrow \emptyset$
15: **for** each cluster $k = 1$ to $K$ **do**
16: $\quad q_k \leftarrow (\lfloor \tilde{w}_k \rfloor + \tilde{r}_k) \times B \times |D_t|$ $\hfill \triangleright$ Cluster quota ($\tilde{r}_k$ is Bernoulli)
17: $\quad \mathrm{sampled\_count}_k \leftarrow 0$
18: $\quad$ **while** $\mathrm{sampled\_count}_k < q_k$ and $|S| < B \times |D_t|$ **do**
19: $\qquad$ **for** each $x \in D_{\mathrm{aux}} \cap C_k$ **do**
20: $\qquad\quad \alpha_k \leftarrow \min(1, \mathrm{sampled\_count}_k / q_k)$ $\hfill \triangleright$ Sampling progress
21: $\qquad\quad s_{\mathrm{proto}}(x) \leftarrow 1/(1 + \mathrm{dist}(f(x), \mu_k))$ $\hfill \triangleright$ Prototypical score
22: $\qquad\quad s_{\mathrm{boundary}}(x) \leftarrow \min_{j \neq k} \mathrm{dist}(f(x), \mu_j) - \mathrm{dist}(f(x), \mu_k)$ $\hfill \triangleright$ Boundary score
23: $\qquad\quad s(x) \leftarrow (1 - \alpha_k^2) \cdot s_{\mathrm{proto}}(x) + \alpha_k^2 \cdot s_{\mathrm{boundary}}(x)$
24: $\qquad\quad s(x) \leftarrow s(x) - \delta \cdot \sum_{x' \in S \cap C_k} \mathbb{1}[\mathrm{Sim}(x, x') > \tau_{\mathrm{sim}}]$ $\hfill \triangleright$ Diversity penalty
25: $\qquad$ **end for**
26: $\qquad$ Sample $x \sim \mathrm{Categorical}(\{s(x)\})$ from $D_{\mathrm{aux}} \cap C_k$
27: $\qquad S \leftarrow S \cup \{x\}; \mathrm{sampled\_count}_k \leftarrow \mathrm{sampled\_count}_k + 1$
28: $\quad$ **end while**
29: **end for**
30: **return** $S$

---

the target data size). This yields a small, efficient fine-tuning set compared to using the entire $D_{\mathrm{aux}}$. We always include all of $D_t$ (the target's original data) to ensure no target-specific information is lost.

We fine-tune the base model on $D_t \cup S$, instantiating a new adapter $\phi_t$ for the target language $\ell_t$. During training, $\Theta$ remains frozen, and only $\phi_t$'s parameters are updated. After training, we have a specialized adapter $\phi_t$ that encodes improvements for language $\ell_t$.

### 3.2.6 Multi-language Routing

We repeat the above process for each target language of interest, obtaining adapters $\phi_{t_1}, \phi_{t_2}, \ldots$ for languages $\ell_{t_1}, \ell_{t_2}, \ldots$. At inference time, we load the base model $M_{\mathrm{base}}(\Theta)$ and all the adapters. When an input comes in, we detect its language using an inexpensive language ID model. Based on the detected language, we attach the corresponding adapter to the model and process the input. Because these adapters are lightweight, it's feasible to store a large quantity of them. Switching adapters is also efficient as we're swapping in a small set of weight delta matrices. This design means each input is handled by a model that's effectively specialized for that language.

### 3.3 Continual Adaptation Phase

Crucially, we extend COMPASS for continuous improvement. As language use and data distributions evolve in real-world scenarios (e.g., emerging markets, shifting user bases or preferences, new language- or locale-specific jargon, seasonality), we facilitate periodic updates to these adapters. Furthermore, offline proxies to estimate "live usage distribution" may not reflect real-world usage patterns, leading to misaligned optimization.

By comparing the distribution of incoming data, significant shifts can be detected, and COMPASS criteria can be reapplied to select relevant new training examples (from the incoming stream or an updated pool) to retrain or update the corresponding language-specific adapter when the clusters become stale. This allows the system to adapt to new data patterns while the targeted nature of the update, guided by distributional needs and localized to specific language adapters, might preserve performance on learned data patterns and avoid interference.

Our enhancement of COMPASS with an Elastic Consolidation and Distributional Anchoring (ECDA) update mechanism contributes not by inventing a new continual learning primitive, but through data-centric integration of established techniques into a unified adaptation lifecycle, all governed by distributional signals. First, the Jensen-Shannon divergence between cluster distributions serves as trigger for detecting significant distribution shifts (Oh et al., 2025). Second, when a shift is detected, our sampling algorithm is re-applied to select the most relevant new data for the update. Third, our rehearsal buffer is not populated randomly but with "distributional anchors" – prototypical examples from the centroids of stable, previously learned clusters. This constitutes a principled form of rehearsal that is more targeted than random sampling (Isele & Cosgun, 2018).

#### 3.3.1 Incremental Clustering

To enable meaningful distribution comparisons over time while accommodating evolving data patterns, we implement a hierarchical clustering.

In the initial adaptation phase, experiments demonstrated that HDBSCAN and K-means provided the best cluster quality. For continual adaptation, we leverage their incremental variants to maintain computational efficiency while preserving cluster stability:

**Incremental K-means**   We maintain cluster centroids from the previous iteration and update them using mini-batch gradient descent on new data. The update rule for centroid $\boldsymbol{\mu}_k$ at time $t$ is:

$$\boldsymbol{\mu}_k^{(t+1)} = (1-\eta)\boldsymbol{\mu}_k^{(t)} + \eta \cdot \text{mean}(\mathbf{X}_k^{\text{new}}) \tag{2}$$

where $\eta$ is a learning rate controlling adaptation speed and $\mathbf{X}_k^{\text{new}}$ represents new points assigned to cluster $k$.

**Incremental HDBSCAN**   Following the transductive extension of HDBSCAN*, we fix the condensed tree structure from the initial training phase and assign new points to clusters based on where they would fall in this fixed hierarchy. This preserves cluster identities while allowing membership updates. Specifically, for each new point $\mathbf{x}_{\text{new}}$, we (1) compute its core distance relative to the existing point set, (2) determine its position in the condensed tree without modifying the tree structure, and (3) assign it to the cluster corresponding to its tree position.

#### 3.3.2 Distribution Mismatch Trigger

To enable continuous adaptation, we detect when language-specific adapters require updates via distributional divergence. For each language adapter corresponding to target language $\ell_t$, we maintain a reference distribution $P_{\text{ref}}^{(\ell_t)}$ representing the cluster proportions from its most recent training cycle. Formally, if the adapter was last trained using clusters $\{C_1, C_2, \ldots, C_K\}$, then:

$$P_{\text{ref}}^{(\ell_t)}(C_k) = \frac{n_k^{\text{eval}} + n_k^{\text{aux,selected}}}{\sum_{j=1}^{K}(n_j^{\text{eval}} + n_j^{\text{aux,selected}})} \tag{3}$$

where $n_k^{\text{aux,selected}}$ denotes the number of auxiliary examples selected from cluster $k$ during the last training cycle. We continuously compute a monitoring distribution $P_{\text{mon}}^{(\ell_t)}$ from recent incoming data $\mathcal{W}_{\text{recent}}$. For each cluster $C_k$:

$$P_{\text{mon}}^{(\ell_t)}(C_k) = \frac{|\{x \in \mathcal{W}_{\text{recent}} : f(x) \in C_k\}|}{|\mathcal{W}_{\text{recent}}|} \tag{4}$$

The Jensen-Shannon divergence between these distributions provides our primary shift detection signal:

$$\text{JS}(P_{\text{ref}}^{(\ell_t)} \| P_{\text{mon}}^{(\ell_t)}) = \frac{1}{2} \cdot \text{KL}(P_{\text{ref}}^{(\ell_t)} \| M) + \frac{1}{2} \cdot \text{KL}(P_{\text{mon}}^{(\ell_t)} \| M) \tag{5}$$

where $M = \frac{1}{2}(P_{\text{ref}}^{(\ell_t)} + P_{\text{mon}}^{(\ell_t)})$ and $\text{KL}(\cdot \| \cdot)$ denotes the Kullback-Leibler divergence. An adapter update for language $\ell_t$ is triggered when:

$$\text{JS}(P_{\text{ref}}^{(\ell_t)} \| P_{\text{mon}}^{(\ell_t)}) > \theta_{\text{JS}} \tag{6}$$

where $\theta_{\text{JS}}$ is an empirically tuned threshold specific to each language.

### 3.3.3 Update: Incremental Fine-tuning with Distributional Guidance

Once the trigger condition is met, indicating a significant distribution shift for a target language $l_t$, the corresponding adapter $\phi_t$ is updated. A simple retraining on the new data could lead to catastrophic forgetting of previously learned patterns. To balance adaptation to the new distribution with the preservation of existing knowledge, we employ a hybrid strategy called Elastic Consolidation and Distributional Anchoring (ECDA). This method combines parameter-space regularization with a targeted, distributionally-guided form of rehearsal.

The update process involves incrementally fine-tuning the existing adapter $\phi_t^{(i-1)}$ to produce an updated version $\phi_t^{(i)}$. The fine-tuning is performed on a new dataset $D_{new}$, which is selected from the recent data window $\mathcal{W}_{recent}$ using the COMPASS sampling criteria. To mitigate forgetting, the optimization is guided by a composite loss function:

$$\mathcal{L}_{\text{total}} = \mathcal{L}_{\text{task}}(D_{new}) + \beta \cdot \mathcal{L}_{\text{DAR}}(\mathcal{B}_{anchor}) + \mathcal{L}_{\text{EWC}}(\phi_t^{(i)}, \phi_t^{(i-1)})$$

This objective function consists of three key components:

1. **Task Loss ($\mathcal{L}_{\text{task}}$):** Cross-entropy loss on the new data $D_{new}$, driving adaptation to the new data distribution.

2. **Distributional Anchor Replay Loss ($\mathcal{L}_{\text{DAR}}$):** This is the task loss computed on a small, fixed-size memory buffer $\mathcal{B}_{anchor}$ containing "distributional anchors" from the previous update cycle. These anchors are prototypical examples selected from the centroids of high-density clusters of the previous reference distribution. Targeted rehearsal provides stability by grounding the model in previously learned knowledge (Rolnick et al., 2019), with contribution of this loss weighted by a hyperparameter $\beta$.

3. **Elastic Weight Consolidation Loss ($\mathcal{L}_{\text{EWC}}$):** Penalizes changes to the adapter parameters that were important for the previous task distribution, providing parameter-level stability. It is adapted from Elastic Weight Consolidation (EWC) (Kirkpatrick et al., 2017) and is defined as:

$$\mathcal{L}_{\text{EWC}} = \frac{\lambda}{2} \sum_j F_j (\theta_j^{(i)} - \theta_j^{(i-1)})^2$$

    where $\theta_j$ is an individual parameter of the adapter $\phi_t$, $\lambda$ is a regularization hyperparameter, and $F_j$ is the diagonal of the Fisher Information Matrix (FIM). FIM is computed using the examples in the distributional anchor buffer $\mathcal{B}_{anchor}$, making the EWC component computationally tractable and focused on protecting parameters crucial for the most representative past data (Zhang et al., 2025).

Following each update, the anchor buffer $\mathcal{B}_{anchor}$ is repopulated with new prototypical examples from the just-learned data distribution, preparing the system for the next cycle.

## 4 Experiment Setup

### 4.1 Datasets

We use a variety of datasets for fine-tuning and evaluation, focusing on those that provide extensive multilingual coverage. Appendix B describes the extent of language coverage, categorized across script, family, subgrouping, and data resources, for each dataset.

- **Aya Dataset (Singh et al., 2024):** Our primary fine-tuning data source serving as the pool for $D_{\text{aux}}$. Aya is a large open multilingual instruction tuning dataset, consisting of 204K human-curated instruction-response examples in 65 languages. It covers a wide range of tasks and domains, from general knowledge Q&A to creative prompts. Aya was collected via an open annotation platform, and it is currently one of the most comprehensive public datasets for aligning language models in many languages. For each target language that we adapt to, we define $D_t$ as a subset of Aya (the fine-tuning target data in that language). We set $E_t$ as the dev set of Global-MMLU or MMLU-ProX questions in that language (for distribution analysis) and use the test sets of the evaluation benchmarks (detailed below) for final scoring.

- **Global MMLU (Singh et al., 2025):** For primary evaluation of COMPASS, we use Global-MMLU, which extends the Massive Multitask Language Understanding (MMLU) benchmark (Hendrycks et al., 2021) to 42 languages. MMLU is a collection of multiple-choice questions across 57 subjects (history, science, math, etc.) originally in English. Global-MMLU provides translations of these questions into many languages with the addition of culture-specific questions for each language. We use the full Global-MMLU dataset, which consists of 792 culturally sensitive and 2,058 culturally agnostic instances per language (119,900 total instances) and a dev set of 285 instances per language. This dataset is challenging as it tests knowledge and reasoning across domains. We use it both as an evaluation benchmark and as a source to simulate distribution shifts (since it has a dev and test split for each language, and content differences across languages).

- **MMLU-ProX (Xuan et al., 2025):** MMLU-ProX offers another challenging evaluation dataset. Expanding upon MMLU-Pro (Wang et al., 2024), which enhanced the original MMLU with increased complexity and answer choices, MMLU-ProX covers 29 languages with approximately 11,829 questions per language. Similar to Global-MMLU, MMLU-ProX was validated by expert human annotators for conceptual accuracy, terminological consistency, and cultural relevance. For MMLU-ProX, we use each language's available dev set and the test set of MMLU-ProX-Lite as a proxy for live data to tune adapters on COMPASS-derived Aya training data, evaluating them on test samples from the full MMLU-ProX data set (minus the test samples contained within MMLU-ProX-Lite).

- **OneRuler (Kim et al., 2025):** A multilingual benchmark designed to evaluate long-context understanding capabilities across 26 languages and context lengths of up to 128K tokens. OneRuler adapts the English-only RULER benchmark framework, featuring seven synthetic needle-in-a-haystack (NIAH) task variations: single NIAH, multi-key NIAH, multi-value NIAH, multi-query NIAH, nonexistent needle, and common word extraction (easy and hard versions).

- **Other evaluation sets:** We consider additional evaluations to probe specific aspects of our methodology in smaller capacities with limited language coverage. XNLI (Conneau et al., 2018) is a cross-lingual natural language inference dataset covering 15 languages, testing the model's ability to understand entailment, contradiction, and neutral relationships between sentences. XQuad (Artetxe et al., 2020) is a QA dataset covering 11 languages, testing reading comprehension abilities. MGSM8k (Shi et al., 2022) is a multilingual grade school math problem dataset covering 10 languages, testing mathematical reasoning capabilities. Respectively, we include each as tests of how well our adapted models handle understanding semantics in multiple languages, improve on additional QA style interactions, and whether the adapters have improved reasoning or just linguistic understanding. For evaluation on these benchmarks, we use the COMPASS-derived adapters that used Global MMLU's dev set as a reference.

## 4.2   Models

We evaluate our method on three pre-trained models to demonstrate its effectiveness and generality.

- **Phi-4-Mini-Instruct-3.8B (Microsoft et al., 2025):** a 3.8B parameter model that uses a Mixture-of-LoRAs architecture for integrating modalities, but here we use its text-only version. It has an expanded 200k vocabulary for multilingual support, but known regressions on non-English tasks compared to the earlier Phi3 iteration, possibly due to less balanced training. We treat Phi-4-mini as a primary subject for improvement, as it represents a smaller, more specialized model that could benefit from targeted fine-tuning.

- **LLaMA-3.1-Instruct-8B (Grattafiori et al., 2024):** an 8B parameter model based on a dense transformer model and known to have strong multilingual capability out-of-the-box on several high-resource languages due to extensive pre-training.

- **Qwen2.5-7B-Instruct (Qwen et al., 2025):** a 7B parameter model pre-trained on over 18T tokens across 29 languages, including many low-resource languages in our evaluation set. Provides insights into how COMPASS performs when the base model already has some exposure to target languages versus completely unseen languages.

COMPASS requires specifying an embedding model to facilitate dataset sampling and a language identification model to route inputs to the appropriate, language-specific adapter. We use Jina-Embeddings-v3-570M (Jina) (Sturua et al., 2024) to generate task-specific embeddings customized for semantic text similarity and retrieval, supporting 100 languages (Appendix A) and context lengths of up to 8192 tokens. On semantic text similarity tasks within the Massive Multilingual Text Embedding Benchmark (MMTEB; (Enevoldsen et al., 2025)), Jina best captured semantic similarity across languages while being relatively robust to syntactic and lexical differences, outperforming multilingual-e5-large-instruct (the best performing model in the initial MMTEB release). For more details, we evaluate sensitivity to encoder choice and clustering parameters in section 5.7.

In initial evaluations on COMPASS' sensitive to the choice of encoder, we compared additional embedding models (gte-multilingual-base-305M, distiluse-base-multilingual-cased-v2-135M, paraphrase-multilingual-mpnet-base-v2-278M (Zhang et al., 2024; Reimers & Gurevych, 2019; Yang et al., 2019)). Jina-Embeddings-v3-570M provided the best cross-lingual alignment and language coverage (i.e., unsupported language sentences might cluster oodly or be embedded erroneously near unrelated data points). Inference-time routing uses GlotLID-v3 (Kargaran et al., 2023) for language identification and loading of the correct adapter, enabling a unified system for all languages. GlotLID supports more than 2000 languages, including all languages used in this study, and all evaluations of COMPASS use GlotLID for adapter loading. If the language does not have a COMPASS-associated adapter, then it defaults to the pretrained model (if no fine-tuning data is available) or to the target only adapter (if fine-tuning data is available but there is no dev set for distribution matching).

## 4.3   Training Setup

We fine-tune DoRA adapters for each language in Global MMLU and MMLU-ProX, using all of the target language's data within Aya in conjunction with the COMPASS-sampled auxiliary data. To avoid data leakage from the test set, distribution approximation is done to minimize distribution discrepancy between the train and development data sets. DoRA hyperparameters were tuned on an aggregate subset of languages representing different families and resource levels: Spanish, French, Russian, Arabic, Swahili, Vietnamese, Bengali, Korean, Thai, and Yoruba. We target modules in the attention and feedforward layers with low-rank decomposition matrices of rank $r = 16$ for Phi-4-mini and Qwen2.5-7B adapters, and $r = 8$ for LLaMA-8B. We use AdamW optimizer ($\beta_1 = 0.9$, $\beta_2 = 0.999$), weight decay of 0.1, gradient clipping of 1, batch size of 128, and with learning rate of 2e-4 for Phi-4-mini and 1e-4 for LLaMA-8B and Qwen2.5-7B, with a 0.1 warmup ratio and cosine scheduler in each setting. COMPASS with LoRA adapters achieved comparable performance but using DoRA adapters resulted in more robust improvements across a broader range of

hyperparameter combinations. We limit fine-tuning to 3 epochs with early stopping. After fine-tuning, we have for each target language $\ell_t$ the adapter weights $\phi_t$. We keep the base model weights $\Theta$ unchanged (shared across all). During inference, we keep the base $W$ separate and add $\Delta W$ on the fly when the adapter is loaded.

We compare our approach to several baselines:

- Monolingual DoRA fine-tuning (Target): Fine-tune a DoRA adapter on $D_t$ alone (target language data only). This represents the scenario of no cross-lingual transfer and avoids any possible interference. We expect our method to outperform this if cross-lingual transfer is beneficial, particularly for smaller $D_t$.

- COMPASS full fine-tuning (COMPASS-FFT): Fine-tune the entire pretrained model on the COMPASS-supplied datasets, leveraging the model's full parameter space to improve at the task. While this approach allows for maximum adaptation to the target language, it entails substantial memory overhead to store a complete set of model parameters for each target language and increases overfitting risk. Consistent with prior comparisons of learning rate sensitivity of LoRA-related methods to FFT (Biderman et al., 2024), we identified optimal FFT performance with reduced learning rates of 5e-5 for Phi-4-mini, 1e-5 for LLaMA-8B, and 2e-5 for Qwen2.5-7B, along with a reduced warm-up ratio of 0.05 and a batch size of 16 across all models.

- All-data multilingual fine-tuning (All): Fine-tune a DoRA adapter on the entire multilingual dataset (Aya) for that task, i.e., combine all languages' data. This is an extreme opposite of monolingual: maximum exposure to other languages (and also much larger training set). This tests the effect of indiscriminate multilingual training. In past research, this often hurts performance on individual languages especially if the model capacity is limited, due to noise from unrelated languages. But it could provide an upper bound in some cases if more data is always helpful. We apply DoRA fine-tuning on the concatenation of $D_t$ plus all other languages in Aya. Note that this baseline effectively trains a single adapter for all languages.

- Random sampling (Random): Fine-tune on $D_t$ plus an equal amount of randomly sampled auxiliary data (from $D_{\text{aux}}$). If our method selects 2000 auxiliary examples, then this baseline also uses 2000 auxiliary examples but chosen uniformly at random from the pool of other-language examples. This isolates the effect of targeted selection vs. just adding more data.

- LangRank-guided selection (LangRank) (Lin et al., 2019): Requires training a ranking model, dependent on the tasks and datasets used for training, to exhaustively evaluate transfer potential across many different languages, and recommends transfer languages from best to worst. For example, picking the single best auxiliary language ($K = 1$) and using all its data in addition to the target language's data might equate to "just use Spanish data for Catalan". We evaluate for values of $K$ from 1 to 3, inclusive, beyond which LangRank performance saturates.

- Linguistic similarity-guided selection (LangSim) (Eronen et al., 2023): Uses linguistic similarity metrics to measure the distance between languages and choose optimal transfer language(s). eLinguistics-derived similarity (Beaufils & Tomin, 2020) was reported to result in the highest correlation with zero-shot performance on sentiment analysis, named-entity recognition, and dependency parsing tasks, but calculates a genetic proximity score based on comparing consonants, which fails in cross-script comparisons. Instead, we use an averaged vectors from lang2vec (Littell et al., 2017), which represents languages as typological, phylogenetic, and geographical vectors derived from multiple linguistic resources such as the World Atlas of Language Structures (WALS) (Dryer & Haspelmath, 2013) and Ethnologue (Eberhard et al., 2025).

### 4.3.1 Bias Simulation

In public data sets, the train data distribution often matches the test data distribution. To simulate distribution discrepancies when exposing a model to live data from real users, we simulate a subject-based sampling

bias. Global MMLU and MMLU-ProX include a categorization into 57 subjects and 14 subjects, respectively. We assign each subject to a low-sampling bucket with 20% probability. Subjects in the low-sampling bucket are reduced to 20% of their original training data size by randomly removing 50% of instances belong to the subject. Subjects not assigned to the low-sampling bucket have their training data unmodified. Bias simulation is applied separately for each language, creating different training set distributions.

To replicate the real-life scenario where the training set is composed of different data sources with different amounts of noise, we add out-of-distribution data to the Aya Data training set. We leverage the Aya Collection, a large corpus of templated and machine-translated multilingual datasets into 101 languages, and add their MLQA-en dataset (validation split), which was machine-translated using NLLB-3.3B (Team et al., 2022). MLQA-en was chosen from the Aya Collection because it was the only supplemental dataset that (1) covered all languages used in our evaluations and (2) had a high average approval ratio of 0.79 as voted on by at least 20 human annotators (e.g., average approval ratio of 0.8 would indicates that 4 out of every 5 annotations was perceived to be of good quality).

# 5 Results

## 5.1 COMPASS best balances cross-lingual transfer benefits and risks for all languages

COMPASS provides the most effective balance for multilingual adaptation, maximizing benefits of cross-lingual transfer while minimizing risks of negative interference or the high costs associated with full model fine-tuning. Table 1 shows that COMPASS effectively leverages auxiliary data selected via distribution approximation, yielding substantial gains over baseline (ZS) performance for all languages and models and outperforming monolingual tuning (Target), particularly for lower-resource languages where targeted cross-lingual transfer provides the most significant benefit by augmenting limited target-language data. Furthermore, COMPASS surpasses random auxiliary data sampling (Random), demonstrating the advantage of strategic data selection over simply increasing training data volume.

Optimizing the auxiliary data distribution to match the target task distribution, as COMPASS aims to do, may offer a more refined approach to maximizing positive transfer than relying solely on general linguistic relatedness or past transfer performance rankings. While established cross-lingual transfer strategies like LangRank and LangSim show improvements over monolingual and random baselines, COMPASS scores consistently higher. Crucially, COMPASS avoids performance degradation observed with all-data multilingual baseline (All), where indiscriminate mixing of all language data introduces substantial negative interference, resulting in performance regressions across all tasks and models.

| | Phi-4-Mini-Instruct-3.8B | | Llama-3.1-Instruct-8B | | Qwen2.5-7B-Instruct | |
| --- | --- | --- | --- | --- | --- | --- |
| | Global MMLU | MMLU-ProX | Global MMLU | MMLU-ProX | Global MMLU | MMLU-ProX |
| Pretrained (ZS) | 43.5[*] | 21.7[*] | 49.1[*] | 22.8[*] | 52.9[*] | 37.5[*] |
| COMPASS | **52.4** | **28.7** | 55.2 | **26.9** | **59.6** | **43.6** |
| Target | 44.7[*] | 23.8[*] | 50.8[*] | 23.2[*] | 54.6[*] | 38.7[*] |
| COMPASS-FFT | 49.9[*] | 27.7 | **55.5** | 26.1 | 58.1 | 42.5 |
| All | 38.8[*] | 18.5[*] | 43.8[*] | 19.6[*] | 48.0[*] | 33.7[*] |
| Random | 44.9[*] | 22.9[*] | 50.6[*] | 23.7[*] | 55.1[*] | 38.4[*] |
| LangRank | 47.3[*] | 24.5[*] | 51.8[*] | 24.4 | 56.7[*] | 39.4[*] |
| LangSim | 47.3[*] | 24.2[*] | 51.0[*] | 24.0[*] | 56.1[*] | 39.0[*] |

Table 1: Global MMLU and MMLU-ProX benchmark scores in comparison with Phi, Llama, and Qwen models across COMPASS and related methods. Superscripts denote statistical significance of COMPASS vs. each baseline via permutation tests (10,000 iterations): $*$ p<0.05.

Consistent performance gains across different base models suggest that the benefits of COMPASS approach are robust and not specific to a single architecture. Phi4-Mini improved the most, signaling that COMPASS recovers regressed multilingual performance that was previously reported on the related Multilingual-MMLU

task as a regression from 55.4% (Phi3.5-Mini) to 49.3% (Phi4-Mini). Despite the increased difficulty, COM-PASS also maintained its relative effectiveness on MMLU-ProX. Permutation tests confirm all COMPASS improvements over Target baseline achieve $p<0.05$ across models and benchmarks, with medium-to-large effect sizes (Cohen's $d = 0.72$-$0.85$ for Global-MMLU, $d = 0.61$-$0.79$ for MMLU-ProX). Importantly, COMPASS significantly outperforms linguistically-informed baselines ($p<0.05$ vs. LangRank/LangSim with medium effect sizes $d = 0.52$-$0.64$), confirming that distribution-aware selection provides benefits beyond linguistic similarity alone. Sign tests confirm that COMPASS improvements are broadly distributed across languages rather than concentrated in a few outliers (binomial test $p<0.05$ for COMPASS compared to all baselines except COMPASS-FFT).

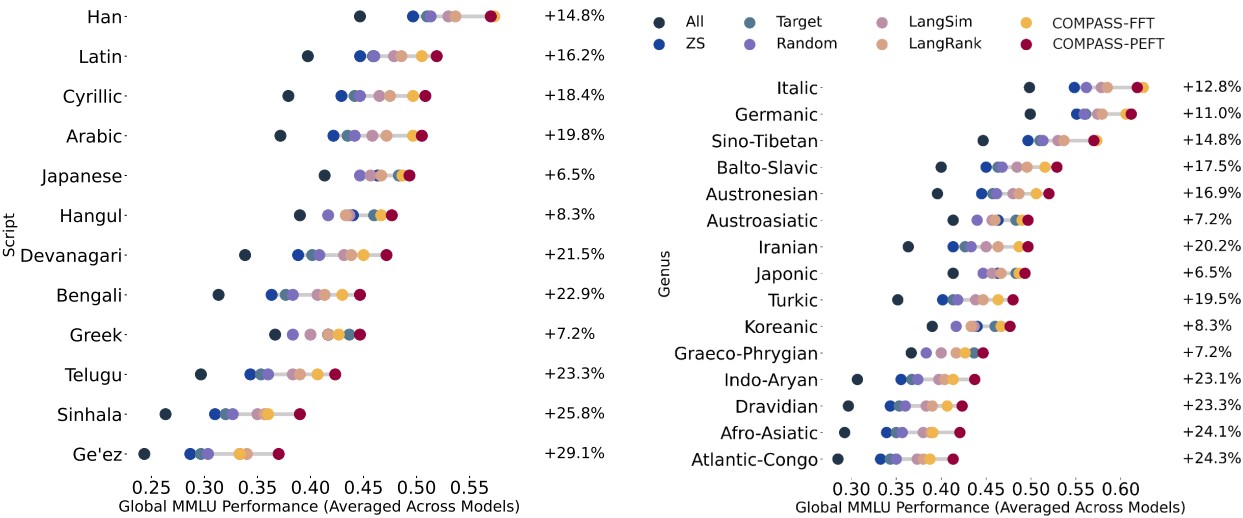

Figure 2: Performance of Phi4-Mini with COMPASS on Global MMLU, segmented by script categorization (left) and by genus categorization (right). Indo-European language family was further split into subgroupings.

COMPASS nets greatest performance gains across each script and genus represented in Global MMLU, and cross-lingual transfer was most beneficial to languages with less data (e.g., Swahili, Yoruba) where the benefit of auxiliary data is higher. Figure 2 breaks Global MMLU performance down across language script and genus categorizations. Regardless of their resource categorization, languages with no related language by script nor language family (Greek, Japanese, Korean, Vietnamese) had marginal performance gains, driven by COMPASS' cluster-level weights facilitating targeted data enrichment. Compared to COMPASS-FFT, COMPASS achieves comparable performance across most languages. While COMPASS-FFT elevates performance for the highest-resource languages, COMPASS remains competitive with greater parameter efficiency, reducing storage requirements and mitigating overfitting risks on mid- and low-resource languages, especially for afroasiatic and atlantic-congo families (see Appendix C for per-language performance).

## 5.2 Impact of Auxiliary Data Budget on Performance

To understand the relationship between the amount of auxiliary data and model performance, we investigate the effect of the budget $B$ (ratio of auxiliary data to target language data). Figure 3 plots relative improvement on Global MMLU for Phi4-mini using COMPASS, compared to a baseline fine-tuned only on target language data. This analysis covers auxiliary budgets $B$ ranging from 20% up to 200% (i.e., 2x) of the target language data size.

The optimal amount of auxiliary data varies across languages, yet performance gains can be achieved with a moderate auxiliary data budget, supporting the notion that COMPASS can reduce data overhead. This result aligns with prior analyses in multilingual instruction fine-tuning that diversifying data mixtures with multilinguality can increase data efficiency, with up to 10x fewer examples, while exhibiting comparable or

superior performance Shaham et al. (2024). For the 42 languages analyzed, the median optimal budget across languages is $B = 80\%$ (19 languages) and languages that peak at $B = 100\%$ (10 languages) or $B = 60\%$ (5 languages) show substantial gains when $B = 80\%$. Only 3 languages achieved peak performance at $B > 100\%$ (Spanish, Portuguese, Telugu). Instead of requiring vast amounts of auxiliary data, COMPASS utilizes a modest amount of strategically sampled data to enhance model performance, with many languages achieving peak performance without auxiliary data volumes that exceed their own target data size ($B \leq 100\%$).

The variability in optimal $B$ may be attributed to several factors. Languages with a richer pool of high-affinity languages within the available auxiliary data might benefit from or tolerate larger budgets as COMPASS has more relevant examples to draw from. Conversely, for languages where the base model (Phi-4-mini) already possesses stronger initial capabilities due to its pre-training, or where high-quality, highly similar auxiliary data is less abundant, performance gains might saturate more quickly or decline with increasing $B$. For instance, languages like Greek (el), English (en), Japanese (ja), and Korean (ko) show peak performance at relatively low $B$ values (20-40%) and exhibit diminishing returns or performance degradation beyond $B = 100\%$. Japanese, for example, peaks at $B = 20\%$ (+4.92%) and shows a negative relative improvement of -13.44% at $B = 200\%$.

Auxiliary budget sensitivity pronounced for language isolates and languages with unique scripts, providing insight into why these languages show marginal improvements with COMPASS. These languages, e.g., Japanese (only Japonic language) and Korean (only Koreanic language), lack closely related family members in our auxiliary data pool. This isolation manifests empirically in auxiliary budget saturation at lower thresholds than the median optimal budget of $B = 80\%$ observed across all languages. Beyond these thresholds, performance degrades sharply, where Japanese's -13.44% regression at $B = 200\%$ exemplifies how additional auxiliary data may become harmful. In contrast, well-represented families, such as Romance, Germanic, and Indo-Aryan languages, sustain performance gains across broader budget ranges, suggesting that syntactic and morphological similarities within language families enable more effective cross-lingual transfer.

While COMPASS's semantic clustering identifies topically relevant auxiliary examples for language isolates, which explains observed positive gains, we posit that syntactically incompatible examples contribute to diminishing or negative returns. At low budgets ($B = 20 - 40\%$), COMPASS samples the most semantically aligned examples, yielding meaningful improvements. At higher budgets, COMPASS exhausts high-quality semantic matches and begins sampling examples that, despite semantic relevance, introduce syntactic noise or conflicting structural patterns that interfere with target language learning. As a diagnostic tool, sharp performance degradation beyond low budgets signals insufficient linguistic affinity in the auxiliary pool.

For remaining experiments, we adopted a uniform budget of $B = 80\%$, which was near-optimal and outperformed monolingual fine-tuning across all languages. However, as indiscriminate addition of auxiliary data can be counterproductive, production settings would benefit from budget tuning for each target language to maximize potential gains. In evaluations for $B$ values of 400%, 600%, 800%, and 1000%, performance drop-offs were steep. As the volume of auxiliary data overshadowed the target language data, the adapters optimized more for the characteristics of auxiliary language content, leading to eventual regression for each target language. These diminishing returns are expected – as COMPASS samples more auxiliary data, distribution mismatch lessens and gains from language cross-pollination will plateau.

### 5.3 COMPASS detects & leverages language synergies

Multilingual training has the potential to enhance target language performance by transferring beneficial information from source languages. However, languages possess diverse properties, and their incorporation can be either advantageous or detrimental. Limiting variety and quantity of source languages mitigates the risk of negative cross-lingual transfer, but restricts potential for performance improvement.

COMPASS dynamically detects and leverages language synergies to sample data from the source language pool, converging towards a distribution that enables optimal cross-lingual transfer for a given target language. COMPASS achieves this without relying on any prior knowledge of known language synergies; all source language data is available upfront. Figure 4 visualizes the distribution of source language data sampled for each target language. Languages from the same family are selected more frequently, e.g., Italic, Germanic, and Indo-Aryan languages show strong intra-family influence, suggesting that COMPASS exploits linguistic

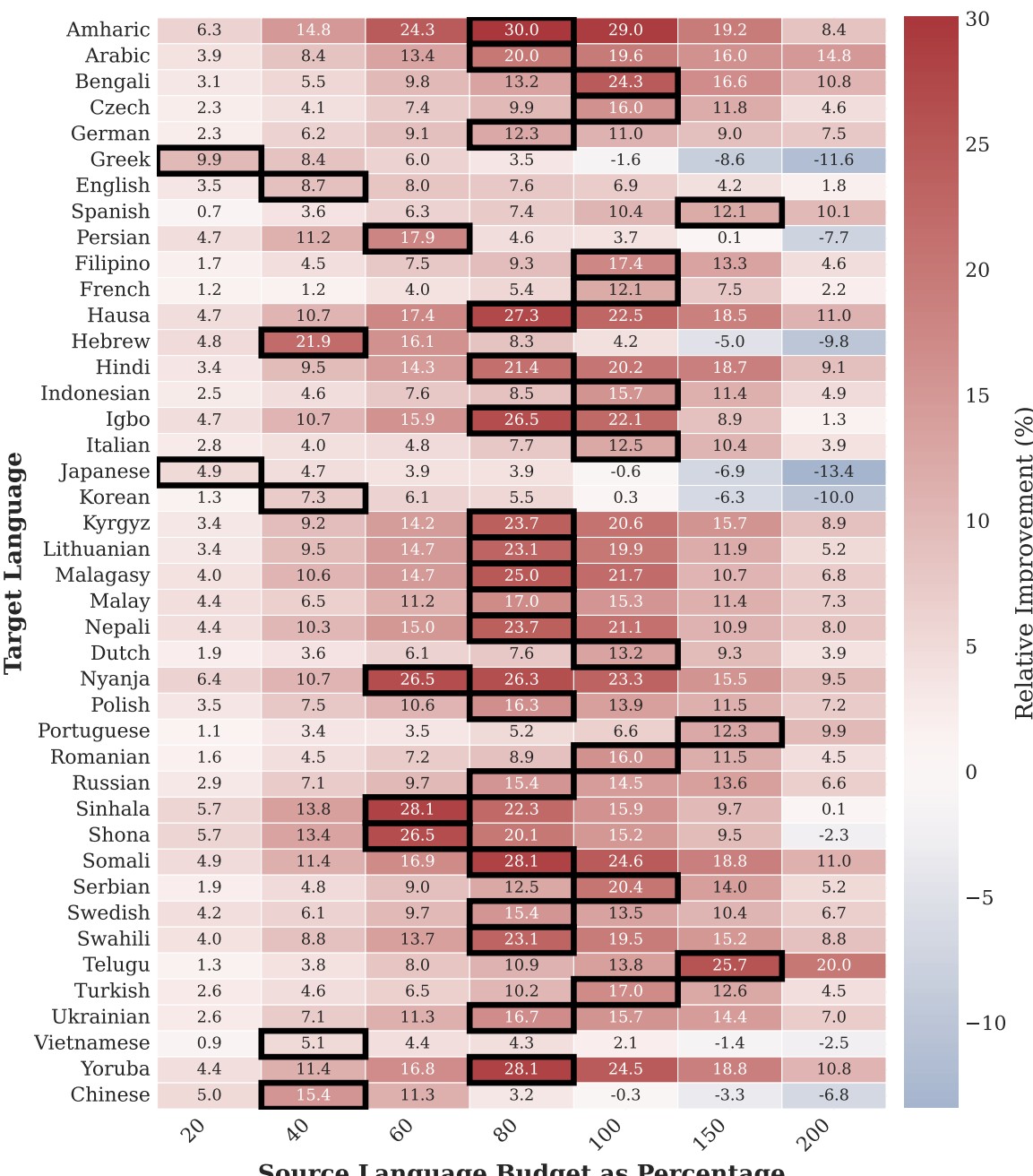

Figure 3: Global MMLU performance of Phi4-Mini with COMPASS, across a range of auxiliary budgets from 20% to 200% (i.e., 2x the size of target language fine-tuning data), relative to performance when fine-tuning on just the target language data.

relationships. Notable exceptions, which draw significant contributions from other groups, include languages within the Iranian, Sino-Tibetan, and Balto-Slavic families. The Isolate family functions as a negative control, where the observed lack of affinity is expected as none of these languages have a close linguistic relation to any other language.

Several source languages contribute to a large number of target languages without a clear family-based pattern. These "indiscriminately sampled" languages include Malagasy, Malay, Tamil, Telugu, and Sinhala. For instance, Malagasy (mg) is sampled across a wide range of target languages, including those from Afro-Asiatic, Niger-Congo, and Austronesian families—language families with no demonstrable typological relationship to Malagasy. While Jina3 supports these languages, their reported primary tuning efforts focused on a set of 30 other languages that does not include any of the indiscriminately sampled languages. The observed affect on the sampling process might be an artifact of less refined or lower-quality embeddings, entailing less precise cross-lingual similarity assessments and spurious measures of linguistic similarity.

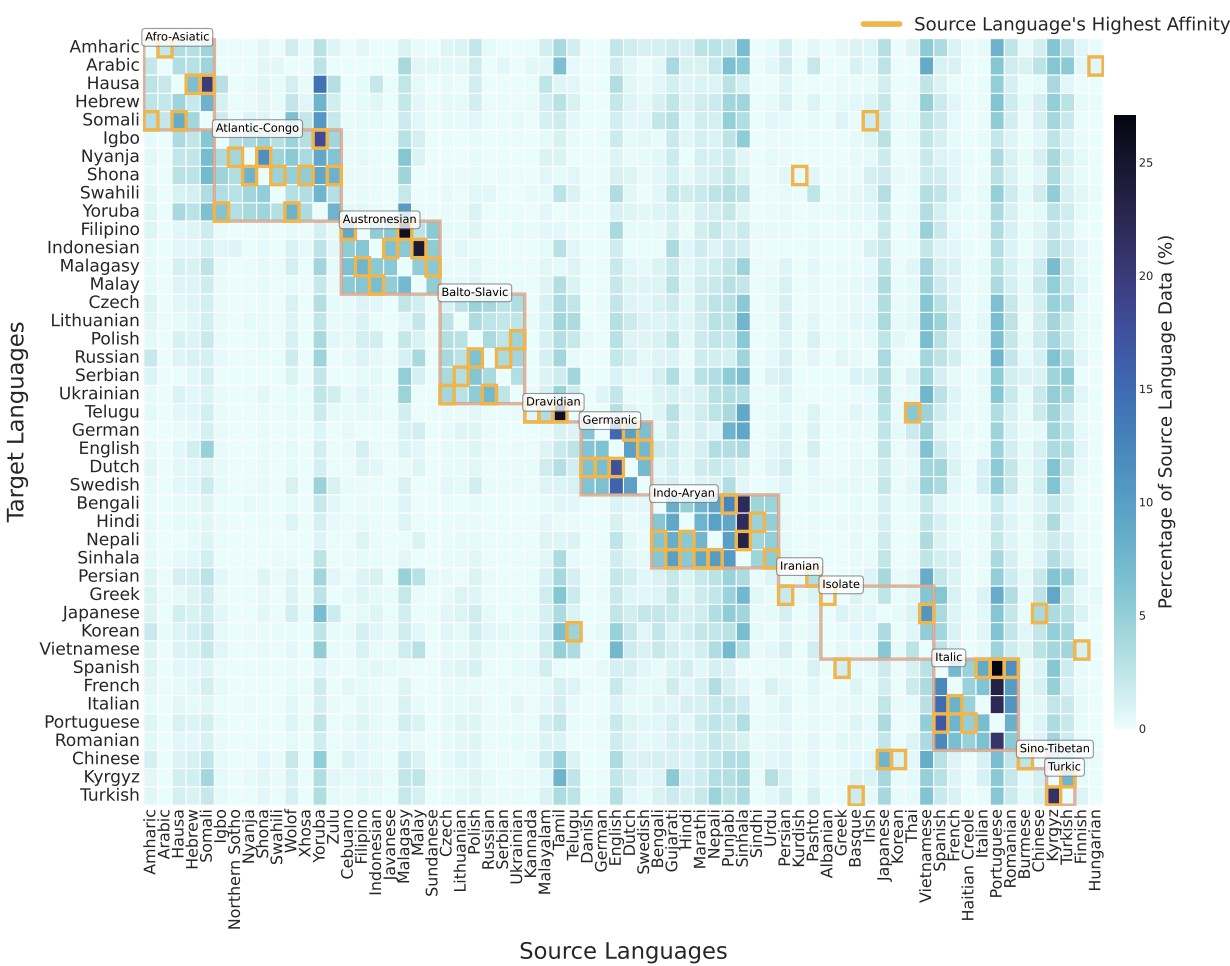

Figure 4: Heatmap of language contribution from each source language (x-axis) to each target language (y-axis). Indo-european languages were divided into further subgroupings. Isolates in this context refers to languages that are not demonstrably related to any other language in this study. The source language distribution reflects the optimum budget for the plurality of target languages.

## 5.4 COMPASS improves performance on multilingual long context tasks

We investigate whether guided fine-tuning with COMPASS and the Aya dataset (comprising short sequence lengths under 1K tokens) improves cross-lingual semantic alignment within the models. We hypothesize that this improved alignment will, in turn, enable the models' pre-existing long-context architectural mechanisms, which were predominantly trained on high-resource data, to be more effectively transferred and applied to low-resource languages during inference.

Our hypothesis is motivated by two lines of reasoning. First, multilingual models operate by learning shared representations across languages; they map words and concepts with similar meanings to nearby points in a shared vector space (Zhong et al., 2024). As we observed on other tasks, COMPASS strengthens these shared representations. Second, recent research in multilingual NMT has demonstrated that "a small amount of long-context data in a few languages is sufficient for cross-lingual length generalization, thereby inducing long-context capabilities" (Gumma et al., 2025). This suggests that the ability to handle long contexts, once learned and encoded into the model's architecture (e.g., through adoption of relative positional embeddings), can be transferred across languages. While the Aya dataset itself lacks long-context data, it provides sufficient multilingual signal that may be a prerequisite for this transfer to occur. The fine-tuning process acts as a form of continued pre-training focused on multilingual instruction following, which should solidify the cross-lingual alignments necessary for the model to apply its HRL-trained long-context machinery within LRL contexts.

**HRL Performance.** For HRLs (English, French, German, Spanish, Italian, Russian, Japanese), we observe a mix of marginal improvement or regressions, with more pronounced regressions as context length increases. These languages are already well-represented in each models' extensive pre-training corpora, which include vast amounts of long-context data (e.g., books, code repositories). Fine-tuning on the short-context Aya data is unlikely to introduce any new long-range reasoning capabilities while risking disruption of the carefully calibrated weights that govern long-context performance, leading to degradation. This is the exact problem that advanced techniques like LongRoPE2's mixed short/long training are designed to prevent (Shang et al., 2025), offering a future mitigation strategy. These slight variations in baseline performance confirm that COMPASS fine-tuning neither significantly helps nor harms these already well-optimized languages, with limited parameter modifications via DoRA contributing to observed high-resource stability. A small subset of HRLs (Hindi, Polish, Portuguese, Dutch, Serbian) show more substantial improvements at 8K and 32K context lengths before tapering off at longer context lengths.

**LRL and MRL Performance.** For LRLs and MRLs, the relative change in performance is tied to how well the target language is initially supported by the base model, with gains being largest at shorter long-context splits (8K and 32K) before consistently degrading at lengths of 64K and 128K. The baseline performance of the models in these languages on OneRuler is significantly lower than in HRLs, creating substantial room for improvement. Once the model can "think" more effectively in Swahili, it can apply its existing, but previously dormant long-context architectural machinery to a Swahili problem. Qwen2.5-7B (Vietnamese, Korean, Chinese, Swahili, Persian) and Phi4-Mini (Norwegian, Danish, Finnish, Czech, Ukrainian, Swedish, Chinese) utilize COMPASS fine-tuning to incur performance improvements on substantially more languages than Llama3.1-8B, possibly due to limitations in tokenization strategy or architectural bottlenecks. Hungarian and Tamil, for which only synthetic fine-tuning data is available with no explicit support across any of the base models, were the only languages with no improvement across any setting.

**Cross-Model Analysis.** While all models support a 128K token context length, they employ different mechanisms for length extrapolation that empirically affect fine-tuning plasticity across context lengths. Qwen2.5-7B provides the most robust bridge between short-context fine-tuning extrapolated to long-context evaluation, whereas positive long-context transfer for Llama3.1-8B was non-existent in most OneRuler settings. Despite its small size, Phi4-Mini's large multilingual vocabulary mitigates LRL tokenization issues that could plague other models. Furthermore, its usage of fractional RoPE, which leaves some attention dimensions position-agnostic, could be a built-in defense against overfitting to the positional information in the short-context training data. Despite architectural variations, the context decay pattern remains consistent: any per-language improvements diminish as context length increases, becoming negligible or negative at 128K tokens.

## 5.5 COMPASS-ECDA enables effective continual adaptation

We evaluate how different strategies perform when adapting COMPASS-trained multilingual adapters to distribution shifts. Importantly, all dynamic methods leverage COMPASS's distribution-guided selection from the Aya dataset, differing only in their approach to the stability-plasticity dilemma, isolating the adaptation mechanisms from the underlying data selection strategy.

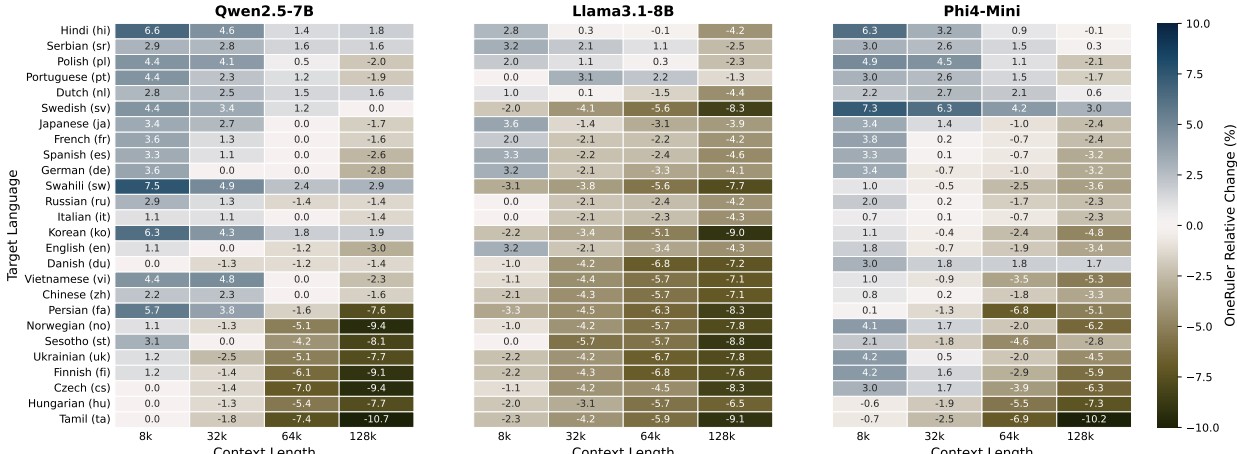

Figure 5: Relative change in performance between the baseline model with and without COMPASS on OneRuler, across language and context settings. To remain consistent with OneRuler's experimental settings, we ensure that the total number of tokens shown to each model is identical, even if some models see more text than others. Regardless, for a given model, its baseline and COMPASS augmented forms see the same amount of text.

### 5.5.1 Single-Step Domain Adaptation

To investigate the inherent tension between learning new patterns and preserving existing knowledge, we simulate a distribution shift scenario in multilingual deployment. We train COMPASS adapters for each target language using a restricted subset of Global MMLU subjects. Specifically, we use only 27 of the 57 available subjects, deliberately excluding 30 subjects that will later appear in the distribution shift. The initial training follows standard COMPASS methodology: for each language $\ell_t$, we use the Global MMLU dev set from the 27 included subjects to guide selection of training data from the training data pool, creating language-specific adapters. This represents the scenario where models are initially trained on available domain-specific data.

After initial training, we simulate a distribution shift by introducing MMLU-ProX data. Critically, we focus on MMLU-ProX questions from the 30 subjects that were excluded from initial training. This ensures a genuine shift encompassing new subject areas the model has not encountered.

All baseline methods begin from the same COMPASS-trained adapters and must adapt to the new distribution. They differ in how they select new training data from Aya and whether they employ regularization strategies. Static deployment provides a lower bound by using the original COMPASS adapter without any updates. Naive fine-tuning applies COMPASS selection using only the MMLU-ProX set for distribution guidance, ignoring the original Global MMLU distribution and representing aggressive adaptation without preservation mechanisms. Full retraining applies COMPASS selection using the combined sets from both distributions, treating them equally to find Aya samples matching the joint distribution. EWC uses COMPASS selection guided by MMLU-ProX but adds elastic weight consolidation, computing Fisher information on the original Global MMLU training data to identify and protect important parameters. Random rehearsal also uses COMPASS selection guided by MMLU-ProX, supplemented with 5% of the original training data randomly sampled and mixed with the new data.

COMPASS-ECDA employs COMPASS selection guided by MMLU-ProX, enhanced with the synergistic mechanisms defined in our ECDA methodology. We apply COMPASS-ECDA with a replay buffer of 5% based on preliminary experiments reported in D. Each method trains for 5 epochs on its selected data, with the total data budget kept constant across methods through proportional scaling.

COMPASS-ECDA achieves Pareto-optimal performance across all strategies through its integrated approach. Figure 6 presents the learning-forgetting trade-off when adapting from a subset of Global MMLU subjects

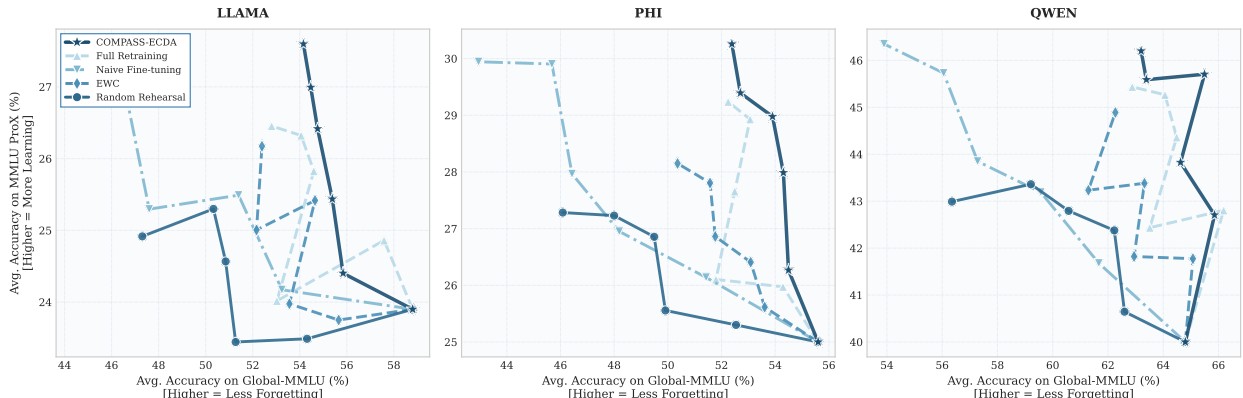

Figure 6: Learning-forgetting trade-off across strategies. COMPASS-ECDA (dark blue) achieves Pareto-optimal performance by combining distributional anchors with elastic regularization. Points represent performance after 5 epochs of adaptation. The x-axis shows retention on original Global MMLU subjects, while the y-axis shows adaptation to new MMLU-ProX domains.

to previously unseen MMLU-ProX domains. Since all non-static methods use COMPASS selection, the performance differences reflect the effectiveness of their adaptation strategies.

Even with strategic data selection, adapting to new distributions requires explicit mechanisms to preserve prior knowledge. Despite using COMPASS to select optimal training data for MMLU-ProX, naive fine-tuning drops substantially on Global MMLU. Full retraining achieves moderate performance but at greater computational cost and assuming access to all data from the initial adaptation stage, which may not be available in practice. By itself, EWC reduces forgetting but remains suboptimal in performance, while random rehearsal's naive preservation of old data fails to preserve core training samples.

The relative advantage of COMPASS-ECDA is most pronounced on the smaller Phi4-Mini model. COMPASS-ECDA achieves the highest adaptation rate while minimizing forgetting below a 5% performance regression on Global MMLU. When model capacity is limited, the quality of preserved examples via distributional anchors becomes crucial. The adaptive instance weighting within the new COMPASS selection adapts sampling strategy based on cluster coverage, initially favoring prototypical examples and progressively incorporating boundary cases to ensure stable learning of new domains.

### 5.5.2 Multi-Step Continual Learning

COMPASS-ECDA navigates multiple distribution shifts while maintaining knowledge accumulated across temporal periods. Real-world deployment involves sequential distribution shifts as user needs evolve over time. To evaluate COMPASS-ECDA's ability to handle multiple temporal shifts, we simulate a deployment trajectory spanning five distinct periods with evolving subject distributions.

We partition the 57 MMLU subjects into temporal phases representing usage evolution (detailed subject allocation in Appendix D). Period T1 establishes initial deployment with 27 foundational subjects, followed by shifts towards 10 advanced STEM domains (T2), 10 humanities and ethics subjects (T3), 10 professional domains (T4), and, finally, a return to 10 subjects from the original T1 subject distribution, simulating cyclical usage while testing retention of intervening knowledge from T2-T4.

Each period consists of approximately 2K samples per language and each transition is applied, and assessed, over 4 increments of 500 samples each. At each transition, the JS divergence trigger detects the distribution shift and initiates adapter updates (see Appendix D) for optimal trigger threshold setting). Performance is tracked throughout the adaptation process, measuring ability to adapt to new tasks and retention of previous knowledge.

Figure 7 presents performance trajectories across the five periods for Qwen2.5-7B-Instruct. COMPASS-ECDA maintains robust performance throughout the temporal sequence, maintaining greater accuracy on MMLU-ProX than all methods aside from naive fine-tuning while preserving Global MMLU performance. In contrast, naive fine-tuning adapts well to each new distribution but exhibits catastrophic forgetting on Global MMLU. EWC and random rehearsal provide intermediate solutions, achieving moderate forgetting but with weaker adaptation to new distributions than COMPASS-ECDA. Full retraining achieves the strongest performance by leveraging all accumulated data at each period, but at multiples of computational cost.

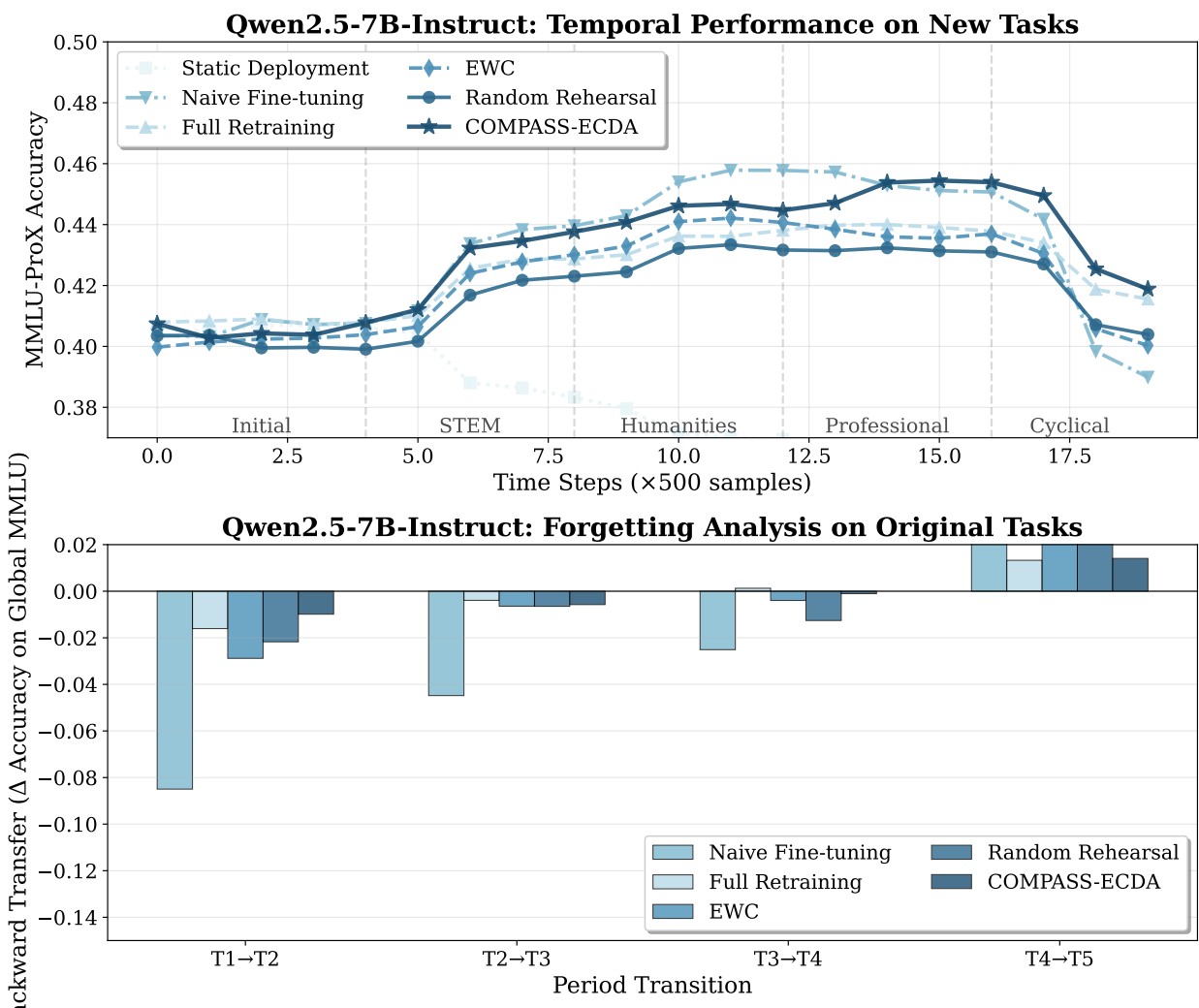

Figure 7: Temporal performance evolution for Qwen2.5-7B-Instruct. Note that time step 0 includes the initial 500 samples from the time period transition of T1 to T2 (i.e., the increments are 0-indexed on the x-axis.

The cyclical return in T5 provides insights into knowledge retention. COMPASS-ECDA recovers T1 performance while maintaining T2-T4 knowledge, demonstrating that distributional anchors successfully preserve knowledge across non-adjacent periods. Naive fine-tuning recovers T1 performance but has largely forgotten T2-T4 subjects, regressing below its initial performance state relative to T1.

When considering performance across all temporal transitions, COMPASS-ECDA achieves a superior balance of forward and backward transfer. Against naive fine-tuning, COMPASS-ECDA achieves comparable forward transfer while significantly reducing backward transfer per transition. Full retraining, despite access to all historical data and comparable forward and backward transfer, underperforms on computational ef-

ficiency (see Appendix F). EWC and random rehearsal fail to match COMPASS-ECDA's adaptation rates per transition.

Model capacity influences continual learning dynamics in our experimental setting. The smaller Phi-4-Mini model benefits most from COMPASS-ECDA's targeted preservation strategy, attributable to insufficient capacity to store essential information via DoRA, as it suffers from the largest regressions in backward transfer when using methods such as naive fine-tuning. LLaMA-3.1 demonstrates similar patterns with enhanced stability, while Qwen2.5's stronger multilingual pretraining amplifies COMPASS-ECDA's effectiveness, yielding the highest absolute performance across all periods.

The consistent pattern across architectures, aggregated over multiple target languages and averaged over 3 random seeds, validates the generalizability of COMPASS-ECDA's approach. Complete temporal trajectories and statistical comparisons for all models are provided in Appendix D.

### 5.6 COMPASS adaptation results in generalized multilingual benefits

The benefits of COMPASS generalize across other evaluation tasks, even without distribution matching (i.e., when cluster-level weights in COMPASS are not a dominant factor). The magnitude of improvement is most pronounced for languages that were initially poorly performing (MGSM) and on tasks that are similar to the fine-tuning objective (XQuAD) 8.

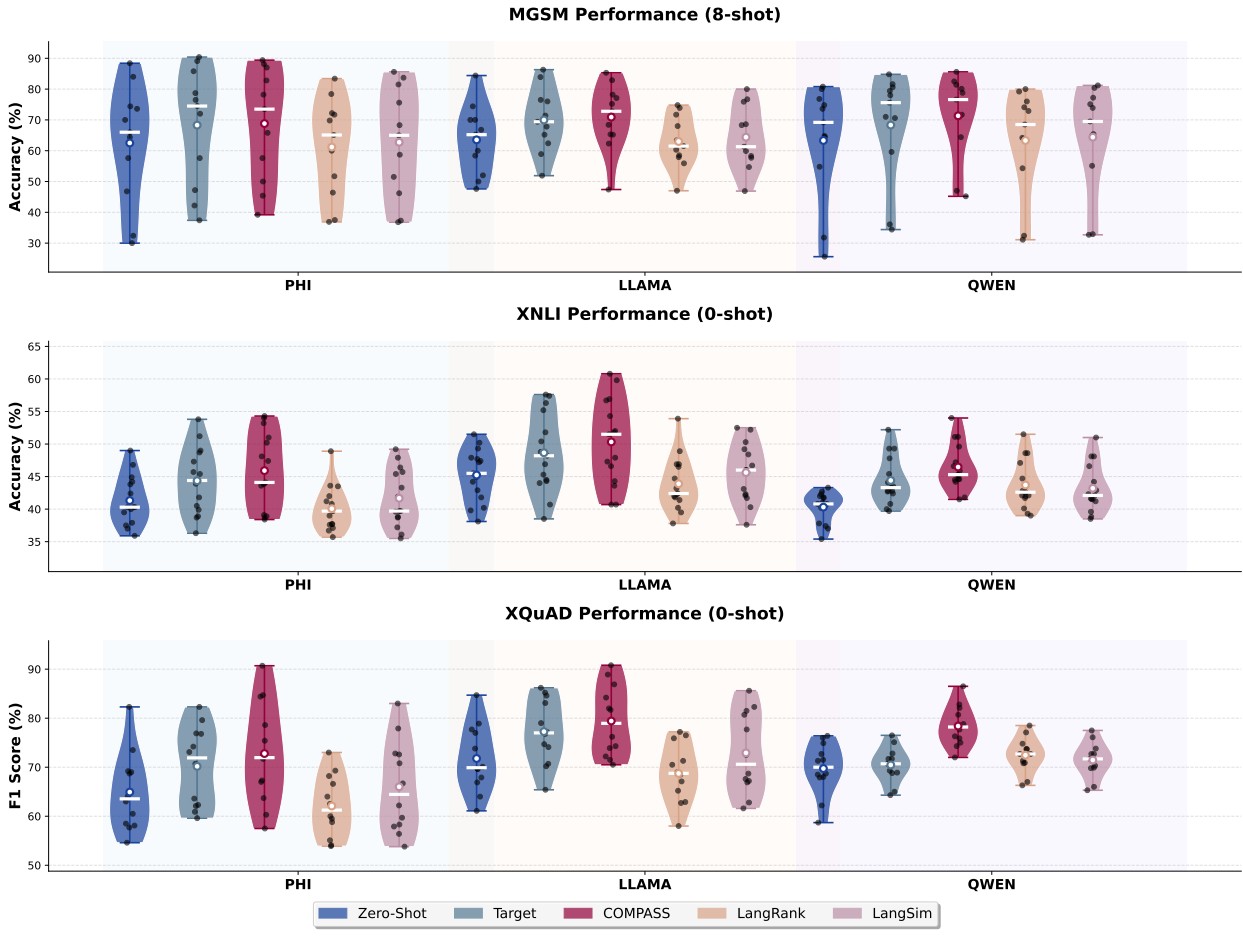

Figure 8: Model performance (Phi4-mini, LLaMA3.1, and Qwen2.5) across COMPASS, baselines, and competing approaches on three diverse multilingual evaluation tasks: MGSM, XNLI, and XQuAD. Our focus is on how performance shifts across different adaptation approaches for each model. The white dot and white line of each violin plot represent the mean and median performance, respectively.

**XQuAD.** XQuAD exhibits the most substantial performance gains among the evaluated benchmarks, underscoring COMPASS's effectiveness in adapting models for tasks that align closely with the instruction-following format of the Aya fine-tuning data. The magnitude of improvement correlates strongly with the baseline performance gap between languages. For Qwen2.5-7B, languages with weaker initial performance yet some exposure in the model's pre-training data, such as Arabic, Thai, and Vietnamese, showed exceptional gains with COMPASS, while high-resource languages like English and Chinese demonstrated more moderate but still substantial improvements. COMPASS consistently outperforms full fine-tuning on XQuAD across all models, suggesting that the targeted cross-lingual transfer approach is particularly effective for question-answering tasks where relevant knowledge may be distributed across languages. The LangRank and LangSim baselines achieve only 60-70% of COMPASS's gains, confirming that distribution-aware selection provides superior transfer compared to linguistic similarity alone.

**XNLI.** On XNLI, COMPASS demonstrates consistent improvements across all three model architectures, with the most salient improvements for languages with limited representation in the pre-training data. Bulgarian, which only had synthetic training data available, was the only language that exhibited a performance drop in the target-only setting, highlighting the challenges of fine-tuning only on data with questionable machine-translation quality. COMPASS mitigates this issue by leveraging high-quality cross-lingual data, maintaining performance even when target language data quality is questionable.

**MGSM.** MGSM results reveal differentiated patterns based on initial model capabilities and language resources. For high-performing languages, COMPASS provides modest improvements across all models. However, for languages with weak baseline performance, the gains are more variable. Performance on MGSM highlights an important characteristic of COMPASS: its effectiveness scales with the availability of relevant cross-lingual knowledge. Mathematical reasoning, being more language-agnostic than other tasks, benefits from cross-lingual transfer when the target language lacks sufficient training examples, but shows diminishing returns for well-resourced languages where the base model already performs adequately.

**Cross-Model Analysis.** Comparing across the three model architectures reveals consistent patterns. Phi4-Mini has high relative improvements particularly on low-resource languages, suggesting that smaller models benefit more from targeted cross-lingual transfer. The model's initial multilingual regression (noted in the Phi4 technical report) is mitigated by COMPASS. Llama3.1-8B demonstrates balanced improvements across all tasks, with less variance between high- and low-resource languages. Qwen2.5-7B, despite extensive multilingual pre-training, shows substantial uplift across the lower performing languages with COMPASS, particularly for XQuAD. Even models with significant multilingual exposure benefit from distribution-aware fine-tuning, especially when adapting to specific task formats.

### 5.7 Ablations

**Embedding Model.** COMPASS relies on the embedding model's semantic representation capabilities, and a combination of lower quality embeddings and lack of language coverage in the embedding model deteriorates COMPASS' data selection quality and downstream performance. GTE Multilingual Base is competitive with Jina3, with a discrepancy in average performance of 2.8% and 1.2%, respectively, on Global MMLU and MMLU-ProX that is in-line with its worse performance on MMTEB. The greater regression of GTE Multilingual Base on Global MMLU may be driven by its lack of coverage for low-resource languages such as Amharic (ge'ez script) and Hausa (latin script).

Cluster- and sample-level weights assume distance in the embedding space reflects semantic text similarity. We posit that there is an inflection point where differences in embedding quality lead to a degradation in cluster- and sample-level weight computation and a subsequent degradation in performance on par with having neither cluster or sample level weights. To stress test COMPASS, we evaluate two more settings for embedding generation, Paraphrase Mpnet and Distiluse Base, representing greater discrepancy in both language coverage (lacking explicit coverage of one-third of Global MMLU languages and of Bengali, Swahili, and Chinese in MMLU-ProX) and with $> 10\%$ performance discrepancy on STS tasks in MMTEB relative to Jina3. Both result in performance reductions that mitigate the benefit of having fine-tuned on the target language and, in the case of Distiluse Base, worsen performance relative to the pretrained model, which is worse than the random sampling baseline. Standard deviations also increase by 3-4% due to associated

regressions where both embedding models lack coverage. These regressions are worse than the random sampling baseline, and indicate that poor embedding quality results in a net negative.

**COMPASS-D vs. COMPASS-S.** Both the deterministic (COMPASS-D) and stochastic (COMPASS-S) algorithms improve over the baseline for all languages. While COMPASS-S outperforms COMPASS-D, indicating value in training data diversity, COMPASS-D is effective without extra sampling overhead.

| | | Global MMLU | | MMLU ProX | |
|---|---|---|---|---|---|
| Component Varied | Setting | Mean | SD | Mean | SD |
| **Baseline Model** | | | | | |
| | Phi4-Mini (3.8B) | 43.5*** | 9.0 | 27.8*** | 10.9 |
| **COMPASS (Optimal)** | | **52.4** | **8.4** | **33.4** | **10.6** |
| *Components:* | DoRA, Jina3, HDBScan, COMPASS-S | | | | |
| **Ablation: PEFT Method** | | | | | |
| | LoRA (r=16) | 51.2 | 10.2 | 32.5 | 12.1 |
| **Ablation: Embedding Model** | | | | | |
| | GTE Multilingual Base | 49.6* | 9.2 | 32.2 | 10.4 |
| | Distiluse Base | 41.1** | 12.5 | 27.4** | 13.6 |
| | Paraphrase Mpnet | 43.9** | 11.6 | 28.8** | 13.1 |
| **Ablation: Clustering Method** | | | | | |
| | KMeans | 47.1** | 10.8 | 29.0** | 12.7 |
| | Agglomerative (Ward) | 50.7* | 8.9 | 31.7* | 11.3 |
| | Taylor-Butina | 40.6*** | 12.4 | 22.1*** | 13.5 |
| **Ablation: Sampling Strategy** | | | | | |
| | COMPASS-D | 51.0* | 9.4 | 32.1 | 10.6 |
| **Ablation: Sampling Importance Weights** | | | | | |
| | w/o sample-level w. | 48.7* | 8.8 | 31.3* | 11.0 |
| | w/o cluster-level w. | 47.3** | 9.4 | 29.9** | 11.6 |

Table 2: Performance of COMPASS and variants on Global MMLU and MMLU-ProX. Scores are reported as the mean accuracy (%) across all languages. Standard deviation (SD) reflects variability of performance across languages, indicating cross-lingual consistency. Superscripts denote statistical significance vs. COMPASS (Optimal) via permutation tests: $*$ $p<0.05$, $**$ $p<0.01$, $***$ $p<0.001$. All results are from single-seed runs.

**Sampling Weights.** Cluster- and sample-level weights are both important to COMPASS, with the former contributing most to performance. To ablate sample-level weights, we sample randomly within each cluster. To ablate cluster-level weights, we ignore cluster mismatches and sample a fixed number of examples per cluster evenly, picking the samples with highest semantic score in each.

Cluster weights benefit the model's ability to prioritize groupings of source language samples that are more representative of interactions with the target language's users, as reflected by live traffic (or a suitable proxy). Sample-level weights prioritize source language data that has greater affinity to the target language and that are easy-to-learn in low data regimes. When we remove sample-level weights, performance drops by 3.7% (Global MMLU) and 2.1% (MMLU-ProX), whereas removal of cluster-level weights reduces performance by 5.1% (Global MMLU) and 3.5% (MMLU-ProX).

Cluster selection (i.e., which topic to draw from) is more critical than sample-level weighting. Ignoring which clusters are needed (i.e., just taking top examples from each cluster equally) wastes capacity on clusters that already had target data reflective of live traffic, or spends too much on clusters that aren't as useful. This aligns with the intuition that ensuring the model sees the right coverage of topics is paramount; seeing the absolute best example versus a moderately good example in a needed topic is secondary, and likely contributes to performance if sufficient to filter out hard-to-learn or noisy examples existing along the cluster boundary.

**Clustering Method.** HDBScan was effective at handling variable cluster shapes and densities and in filtering noise points that were often dissimilar to others (for instance, some code-switched or garbage text examples introduced by MLQA-en). KMeans and Taylor-Butina clustering both underperform relative to HDBScan, with the latter regressing substantially. MMLU tasks are diverse and, when modified to reflect cultural nuances, diverge further from neat, spherical clusters with consistent density. With KMeans, forcing data into ill-fitting shapes across task/language boundaries with greater sensitive to outliers worsens performance. Taylor-Butina clustering, while also a density-based method, is better suited for high-dimensional binary representations with well-defined, often empirically-derived, pairwise similarity thresholds. Agglomerative clustering is most competitive with HDBScan, but is less robust to noise introduced by out-of-distribution training data (MLQA-en).

**DoRA vs. LoRA.** Across both Global MMLU and MMLU-ProX, DoRA achieves 1.2% and 0.9% higher average accuracy than LoRA (r=16), respectively. While this performance gain is marginal and not statistically significant relative to using LoRA, we chose DoRA for greater demonstrated robustness both across languages and hyperparameter configurations. The cross-lingual standard deviation for DoRA is 8.4% and 10.6% compared to LoRA's 10.2% and 12.1% for Global MMLU and MMLU-ProX respectively, indicating more consistent performance across diverse language families. To quantify hyperparameter sensitivity, we evaluated both methods across a grid of learning rates (1e-6 to 2e-4) and rank values (8, 16, 32, 64), as shown in Figure 9. DoRA exhibited a broader optimal learning rate plateau (spanning 1e-5 to 5e-5), whereas LoRA demonstrated sharper performance peaks requiring precise learning rate tuning.

While DoRA offers improved robustness, additional computations to perform weight decomposition into magnitude and direction result in increased training time. In terms of parameter and inference efficiency, DoRA adds a negligible number of parameters for the magnitude vector and does not increase inference overhead relative to LoRA, as both magnitude and direction components can be merged into the pre-trained weight after training (Liu et al., 2024). For COMPASS's use case, the broader optimal hyperparameter range of DoRA reduces the risk of failures when adapting to new languages. Moreover, DoRA's stability, compared to LoRA's rank and learning rate sensitivity, obviates the need for as extensive hyperparameter sweeps to offset the increased cost of an individual training run.

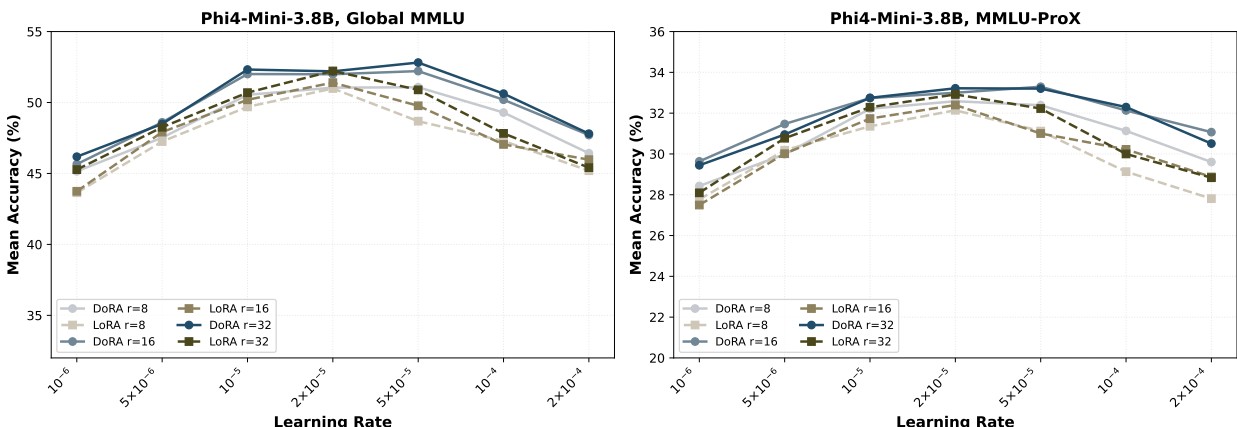

Figure 9: Learning rate sensitivity comparison between DoRA and LoRA across different ranks on (left) Global MMLU and (right) MMLU-ProX. Solid lines with circles represent DoRA, dashed lines with squares represent LoRA. DoRA exhibits a broader optimal plateau and maintains stable performance across learning rates, while LoRA shows sharper peaks requiring precise tuning.

# 6 Discussion

COMPASS is effective for multilingual adaptation, outperforming baselines across diverse model architectures and evaluation benchmarks. By proactively identifying and sampling from under-represented semantic clusters in an auxiliary data pool, COMPASS minimizes the distributional mismatch between the training

data and a target usage distribution. This data-centric approach stands in contrast to prior methods that rely on reactive, computationally intensive gradient manipulation during training or static heuristics such as linguistic similarity, which our experiments show to be less effective. The strategic selection of cross-lingual data maximizes positive transfer while mitigating the negative interference that plagues naive multilingual fine-tuning, a phenomenon starkly observed in the performance degradation of the "All" baseline.

COMPASS-ECDA elevates the framework from a static, one-time adaptation technique to a dynamic, sustainable solution for real-world deployment, where data distributions are inherently non-stationary due to evolving user needs, emerging topics, or shifting demographics. A key advantage is the decoupling of the data selection process from the model training algorithm. The selection strategy, which operates on semantic representations in a pre-processing step, is agnostic to the specific PEFT method employed for adaptation (e.g., DoRA, LoRA). This modularity endows COMPASS with broad applicability, suggesting its potential to enhance not only current but also future adaptation techniques.

### 6.1   Limitations

The performance of any fine-tuning method is ultimately constrained by the properties of the base model and the data ecosystem. First, COMPASS is subject to a tokenization ceiling, where performance is capped by the base model's pre-trained tokenizer. For many LRLs, pre-trained vocabularies are dominated by high-resource language tokens, leading to inefficient "over-fragmentation" where single LRL words are split into multiple sub-word tokens and placing greater strain on the model's context window (Nag et al., 2025). This not only increases sequence lengths and computational costs but can also lead to suboptimal learning (Nicholas & Bhatia, 2023). As our fine-tuning process does not alter the tokenizer, this inherent inefficiency persists.

While COMPASS optimizes the use of existing data, it cannot overcome the primacy of pre-training data scarcity. The foundational reason for the performance gap in LRLs is the scarcity of high-quality, representative data during the initial pre-training stage. Fine-tuning, even with a sophisticated method like COMPASS, is a powerful but partial intervention when compensating for the imbalance in web-scale data that forms the model's core knowledge.

Our evaluation relies on benchmarks like Global-MMLU and MMLU-ProX, yet these still represent only a fraction of the world's linguistic and cultural diversity. The lack of quality, human-translated, and genuinely localized evaluation suites remains a critical challenge for the field. Reliance on translated or synthetic data can obscure a model's true capabilities and risks, particularly for LRLs. These ecosystem-level constraints form a challenging cycle: a lack of pre-training data leads to poor tokenizers, making fine-tuning less effective, while a lack of good evaluation data hinders the measurement of progress. Algorithmic interventions like COMPASS, which improve data efficiency, are a crucial step. However, their full potential can only be realized in tandem with community-driven data curation efforts, which address the data scarcity and evaluation problems directly.

Our framework is also built upon several assumptions. The initial adaptation phase relies on the "proxy for live data" assumption, using a held-out evaluation set to approximate the true "live" usage distribution. This is a pragmatic choice, but the proxy set may not perfectly reflect the topical or stylistic distribution of real-world user queries. This potential mismatch is a primary motivation for the continual learning component of our framework. COMPASS-ECDA is designed to be self-correcting; an imperfect initial adaptation can be refined over time as the system monitors and adapts to the actual incoming data stream.

The continual update mechanism in COMPASS-ECDA is triggered by Jensen-Shannon (JS) divergence, a measure of distribution shift. This trigger is performance-blind. A significant distribution shift might occur that has no material impact on model performance (e.g., a new topic the model already handles well), leading to unnecessary updates. Conversely, a subtle but critical shift, such as the emergence of new adversarial phrasing, might not exceed the threshold, allowing performance to degrade silently.

Furthermore, the continual learning phase relies on incremental clustering implementations that threaten long-term cluster stability. Over extended periods of significant distribution shift, the initial clusters may become "stale," rendering the incremental updates less effective. There likely exists a threshold beyond which performance cannot be improved without re-clustering the entire data pool from scratch (i.e., a "reset" of the

adaptation process). The current work does not define or investigate this threshold, leaving open questions about the framework's stability over very long time horizons.

Finally, the entire data selection strategy is bottlenecked by encoder performance. The quality of the multilingual embeddings produced by the encoder model is paramount. Our ablation studies confirm this dependency, showing that lower-quality embeddings degrade performance. Any biases or weaknesses in the encoder will be propagated and potentially amplified by the selection process.

Our empirical analysis reveals a specific manifestation of this encoder dependency: the "indiscriminate sampling" phenomenon where languages such as Malagasy, Malay, Tamil, Telugu, and Sinhala, which were not included in Jina3's primary tuning set, were sampled disproportionately across unrelated target languages. Consequently, examples from these under-tuned languages contaminate the auxiliary data selection for targets with which they share no linguistic or topical affinity.

This finding demonstrates that COMPASS's sampling is systematically biased by encoder quality disparities across languages. When the encoder fails to capture precise semantic distinctions for certain languages, COMPASS's cluster-based selection inherits and amplifies these flaws. It also highlights the hidden cost of general-purpose multilingual encoders: even when a language is nominally "supported," insufficient tuning can render it harmful to the selection process. Encoder selection and evaluation beyond aggregate metrics (e.g., average performance on MMTEB) would serve as a proactive diagnostic of per-language embedding quality, which might be addressed by either developing encoder-aware sampling weights that down-weight languages with known embedding deficiencies, or investigating whether fine-tuning the embedding model on a subset of the auxiliary data pool could reduce indiscriminate sampling by improving representation quality for under-tuned languages.

## 6.2 Future Work

The current framework's modularity invites exploration into more sophisticated adapter architectures. Instead of treating adapters as monolithic, language-specific modules, future work could view them as carriers of composable "semantic skills." For instance, an adapter trained on Spanish legal text learns both "Spanish grammar" and "legal concepts." This perspective opens up several research directions. One is the development of merged adapters, where router scores from a language identification model could be used to combine multiple adapters at inference time. Another direction is the application of COMPASS to stacked adapters, where a language-specific adapter could be composed with a task-specific one (e.g., Spanish adapter + medica QA adapter) for more fine-grained specialization.

While COMPASS focuses on distributional alignment, its framework could be extended to tackle broader systemic issues in multilingual NLP. For cross-lingual security, the distribution-guided approach could be adapted to select training examples that explicitly cover safety-critical scenarios in multiple languages, potentially reducing the jailbreaking success rates that can jump from $\leq 1\%$ to 79% when translating unsafe inputs (Yong et al., 2024). For tokenization efficiency, COMPASS could be combined with script-aware sampling strategies that ensure balanced representation of non-Latin scripts during fine-tuning, potentially mitigating token length discrepancies that lead to inefficient encoding and decoding of non-Latin inputs (Cui et al., 2024; Ji et al., 2023). Regarding privacy preservation, enhanced multilingual capabilities may improve downstream multilingual preference alignment and safety tuning, strengthening defenses against sensitive training data extraction, personally identifiable information (PII) reconstruction, membership inference, and gradient leakage (Lukas et al., 2023; Li et al., 2024; Nasr et al., 2023).

The principles of distribution-guided sampling are also domain-agnostic and could be extended to new applications. A compelling future direction is to adapt COMPASS for specialized multilingual domains such as multilingual code generation. In this context, the "target distribution" could be defined by a specific codebase, API, or programming style, and the framework could be used to select the most relevant code snippets from a vast, multilingual corpus of open-source projects to fine-tune a code-generating model.

To address the "performance-blind" nature of the current update trigger, a multi-signal trigger system should be developed. The decision to update an adapter could be based on a learned score that integrates multiple signals: distribution divergence, performance degradation, uncertainty increase, and user feedback signals.

Alternatively, consider that the data-centric approach of COMPASS-ECDA is complementary to parameter-centric continual learning methods. A promising avenue is to integrate COMPASS-ECDA with the parameter preservation mechanisms of methods like CURLoRA. A hybrid system combining targeted rehearsal of distributional anchors with principled parameter regularization beyond EWC could prove highly robust.

Addressing bias propagation requires integrating fairness constraints into the sampling objective. Future work should explore debiasing objectives that penalize selection of stereotypical associations and cluster-level auditing to identify whether certain topics (e.g., gender, religion, socioeconomic status) are disproportionately represented or omitted. Privacy-preserving Cluster interpretability methods, such as Clio Tamkin et al. (2024), may provide human oversight of the selection process in real-world applications.

Ultimately, the solution to the LRL data scarcity problem lies beyond any single algorithm, underscoring the importance of large-scale, collaborative data creation and curation efforts. Initiatives like SeaCrowd, which is building comprehensive, standardized corpora for nearly 1,000 Southeast Asian languages, are essential high-quality, localized datasets and benchmarks needed to train and evaluate equitable multilingual models (Lovenia et al., 2025). While COMPASS provides an efficient framework for leveraging existing data, the long-term solution requires expanding the multilingual data ecosystem itself.

**Broader Impact Statement**

The motivation for this work is to address and mitigate linguistic inequality prevalent in LLMs, thereby making advanced technologies more accessible and effective for speakers of low-resource languages. By improving model performance in a parameter- and data-efficient manner, our framework has the potential to foster greater digital inclusion and enable development of culturally and contextually relevant AI applications in underserved communities. Furthermore, enhancing the multilingual capabilities of base models is a crucial prerequisite for developing robust and equitable cross-lingual safety and alignment mechanisms, reducing the risk of users inadvertently bypassing safety guardrails when interacting in non-English languages.

However, improved multilingual generation could be leveraged for malicious purposes, such as creating more convincing and targeted misinformation, spam, or propaganda across a wider range of languages, potentially impacting communities that are not well-equipped to detect or counter such threats. Additionally, our data-centric approach is dependent on the quality and composition of the underlying data sources and embedding models. The framework could inadvertently amplify existing societal biases present in the auxiliary data pool or in the multilingual text encoder used for semantic clustering. COMPASS does not explicitly audit or correct for bias, meaning that if the target data distribution itself reflects biases, the model will learn to replicate them.

Beyond encoder biases, COMPASS's data selection strategy could amplify biases present in the auxiliary data pool. High-performing semantic clusters are preferentially sampled, which may overrepresent majority viewpoints or Western-centric content even within non-Western languages, potentially marginalizing alternative cultural or institutional frameworks in affected communities.

Our distribution-aware sampling prioritizes auxiliary data matching target usage patterns, which may inadvertently sample culturally sensitive content (e.g., indigenous knowledge, minority language expressions, religious texts) without proper cultural context or community consent. While COMPASS does not modify such content, the selection and repurposing of low-resource language data raises ethical questions about data sovereignty and appropriate use that warrant engagement with linguistic communities, especially those that have less agency in dictating data usage concerns.

Lastly, the continual adaptation paradigm in COMPASS-ECDA requires periodic retraining cycles, which has associated environmental costs. While adapter-based approaches are more efficient than repeatedly training full models, production deployments should carefully consider the environmental impact of update frequency and explore strategies such as batching updates across multiple languages to reduce computational overhead.

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

# A   Model Language Exposure

See Table 3. Note that Qwen2.5 advertises explicit support for 29 languages but not all of the supported languages are listed in their documentation.

Table 3: Language Support in Models (▲ = Supported)

| Code | Language | Script | Family | Subgrouping | Phi4-Mini-Instruct-3.8B | LLaMA-3.1-Instruct-8B | Qwen2.5-7B-Instruct | Jina-Embeddings-v3-570M | GlotLID-v3 |
|---|---|---|---|---|---|---|---|---|---|
| ara | Arabic | Arabic | Afro-Asiatic | Semitic | ▲ | | ▲ | ▲ | ▲ |
| ces | Czech | Latin | Indo-European | Balto-Slavic | ▲ | | | ▲ | ▲ |
| deu | German | Latin | Indo-European | Germanic | ▲ | ▲ | ▲ | ▲ | ▲ |
| eng | English | Latin | Indo-European | Germanic | ▲ | ▲ | ▲ | ▲ | ▲ |
| eus | Basque | Latin | Basque | - | | | | ▲ | ▲ |
| fra | French | Latin | Indo-European | Italic | ▲ | ▲ | ▲ | ▲ | ▲ |
| hin | Hindi | Devanagari | Indo-European | Indo-Aryan | | ▲ | | ▲ | ▲ |
| hun | Hungarian | Latin | Uralic | - | ▲ | | | ▲ | ▲ |
| ita | Italian | Latin | Indo-European | Italic | ▲ | ▲ | ▲ | ▲ | ▲ |
| jpn | Japanese | Japanese | Japonic | Japonic | ▲ | | ▲ | ▲ | ▲ |
| nld | Dutch | Latin | Indo-European | Germanic | ▲ | | | ▲ | ▲ |
| pes | Persian | Arabic | Indo-European | Iranian | | | | ▲ | ▲ |
| pol | Polish | Latin | Indo-European | Balto-Slavic | ▲ | | | ▲ | ▲ |
| por | Portuguese | Latin | Indo-European | Italic | ▲ | ▲ | ▲ | ▲ | ▲ |
| rus | Russian | Cyrillic | Indo-European | Balto-Slavic | ▲ | | ▲ | ▲ | ▲ |
| spa | Spanish | Latin | Indo-European | Italic | ▲ | ▲ | ▲ | ▲ | ▲ |
| srp | Serbian | Cyrillic | Indo-European | Balto-Slavic | | | | ▲ | ▲ |
| tur | Turkish | Latin | Turkic | Common Turkic | ▲ | | | ▲ | ▲ |
| vie | Vietnamese | Latin | Austroasiatic | Vietic | ▲ | | | ▲ | ▲ |
| zho | Chinese | Han | Sino-Tibetan | Sinitic | ▲ | | ▲ | ▲ | ▲ |
| ben | Bengali | Bengali | Indo-European | Indo-Aryan | | | | ▲ | ▲ |
| bul | Bulgarian | Cyrillic | Indo-European | Balto-Slavic | | | | ▲ | ▲ |
| ceb | Cebuano | Latin | Austronesian | Malayo-Polynesian | | | | | ▲ |
| dan | Danish | Latin | Indo-European | Germanic | ▲ | | | ▲ | ▲ |
| ell | Greek | Greek | Indo-European | Graeco-Phrygian | | | | ▲ | ▲ |
| fil | Filipino | Latin | Austronesian | Malayo-Polynesian | | | | | ▲ |
| fin | Finnish | Latin | Uralic | Finnic | ▲ | | | ▲ | ▲ |
| heb | Hebrew | Hebrew | Afro-Asiatic | Semitic | ▲ | | | ▲ | ▲ |
| ind | Indonesian | Latin | Austronesian | Malayo-Polynesian | | | | ▲ | ▲ |
| kor | Korean | Hangul | Koreanic | Korean | ▲ | | | ▲ | ▲ |
| lit | Lithuanian | Latin | Indo-European | Balto-Slavic | | | | ▲ | ▲ |
| msa | Malay | Latin | Austronesian | Malayo-Polynesian | | | | | ▲ |
| ron | Romanian | Latin | Indo-European | Italic | | | | ▲ | ▲ |
| swe | Swedish | Latin | Indo-European | Germanic | ▲ | | | ▲ | ▲ |
| tam | Tamil | Tamil | Dravidian | South Dravidian | | | | ▲ | ▲ |
| tha | Thai | Thai | Tai-Kadai | Kam-Tai | ▲ | ▲ | ▲ | ▲ | ▲ |
| ukr | Ukrainian | Cyrillic | Indo-European | Balto-Slavic | ▲ | | | ▲ | ▲ |
| urd | Urdu | Arabic | Indo-European | Indo-Aryan | | | | ▲ | ▲ |
| amh | Amharic | Ge'ez | Afro-Asiatic | Semitic | | | | ▲ | ▲ |
| gle | Irish | Latin | Indo-European | Celtic | | | | | ▲ |
| guj | Gujarati | Gujarati | Indo-European | Indo-Aryan | | | | ▲ | ▲ |
| hat | Haitian Creole | Latin | Indo-European | Italic | | | | | ▲ |
| hau | Hausa | Latin | Afro-Asiatic | Chadic | | | | ▲ | ▲ |
| ibo | Igbo | Latin | Atlantic-Congo | Benue-Congo | | | | | ▲ |
| jav | Javanese | Latin | Austronesian | Malayo-Polynesian | | | | | ▲ |
| kan | Kannada | Kannada | Dravidian | South Dravidian | | | | | ▲ |
| kir | Kyrgyz | Cyrillic | Turkic | Common Turkic | | | | | ▲ |
| kur | Kurdish | Latin | Indo-European | Iranian | | | | | ▲ |
| mal | Malayalam | Malayalam | Dravidian | South Dravidian | | | | | ▲ |
| mar | Marathi | Devanagari | Indo-European | Indo-Aryan | | | | | ▲ |
| mlg | Malagasy | Latin | Austronesian | Malayo-Polynesian | | | | | ▲ |
| mya | Burmese | Myanmar | Sino-Tibetan | Burmo-Qiangic | | | | | ▲ |
| nep | Nepali | Devanagari | Indo-European | Indo-Aryan | | | | | ▲ |
| nor | Norwegian | Latin | Indo-European | Germanic | ▲ | | | ▲ | ▲ |
| nso | Northern Sotho | Latin | Atlantic-Congo | Benue-Congo | | | | | ▲ |
| ny | Nyanja | Latin | Atlantic-Congo | Benue-Congo | | | | | ▲ |
| pan | Punjabi | Gurmukhi | Indo-European | Indo-Aryan | | | | ▲ | ▲ |
| pus | Pashto | Arabic | Indo-European | Iranian | | | | ▲ | ▲ |
| sin | Sinhala | Sinhala | Indo-European | Indo-Aryan | | | | ▲ | ▲ |
| sna | Shona | Latin | Indo-European | Indo-Aryan | | | | | ▲ |
| snd | Sindhi | Arabic | Indo-European | Indo-Aryan | | | | ▲ | ▲ |
| som | Somali | Latin | Afro-Asiatic | Cushitic | | | | ▲ | ▲ |
| sot | Southern Sotho | Latin | Atlantic-Congo | Benue-Congo | | | | | ▲ |
| sqi | Albanian | Latin | Indo-European | Albanian | | | | ▲ | ▲ |
| sun | Sundanese | Latin | Austronesian | Malayo-Polynesian | | | | ▲ | ▲ |
| swa | Swahili | Latin | Atlantic-Congo | Benue-Congo | | | | ▲ | ▲ |
| tel | Telugu | Telugu | Dravidian | South Dravidian | | | | ▲ | ▲ |
| wol | Wolof | Latin | Atlantic-Congo | North-Central Atlantic | | | | | ▲ |
| xho | Xhosa | Latin | Atlantic-Congo | Benue-Congo | | | | ▲ | ▲ |
| yor | Yoruba | Latin | Atlantic-Congo | Benue-Congo | | | | | ▲ |
| zul | Zulu | Latin | Atlantic-Congo | Benue-Congo | | | | | ▲ |

# B  Dataset Language Coverage

See Table 4.

Table 4: Language Resources and Translation Data

| Code | Language | Script | Family | Subgrouping | Resources | Aya (Data▲/Coll.◆) | Global MMLU | MMLU ProX | XNLI | xquad | MGSM8K | OneRuler |
|---|---|---|---|---|---|---|---|---|---|---|---|---|
| ara | Arabic | Arabic | Afro-Asiatic | Semitic | High | ▲◆ | ▲ | ▲◆ | ▲ | ▲ | | |
| ces | Czech | Latin | Indo-European | Balto-Slavic | High | ◆ | ▲◆ | ▲◆ | | | | ▲■ |
| deu | German | Latin | Indo-European | Germanic | High | ▲◆ | ▲◆ | ▲◆ | ▲ | ▲ | ▲ | ▲■ |
| eng | English | Latin | Indo-European | Germanic | High | ▲◆ | ▲◆ | ▲◆ | ▲ | ▲ | ▲ | ▲■ |
| eus | Basque | Latin | Basque | - | High | ▲◆ | | | | | | |
| fra | French | Latin | Indo-European | Italic | High | ▲◆ | ▲ | ▲◆ | ▲ | | ▲ | ▲■ |
| hin | Hindi | Devanagari | Indo-European | Indo-Aryan | High | ▲◆ | ▲ | ▲◆ | ▲ | ▲ | | ▲■ |
| hun | Hungarian | Latin | Uralic | - | High | ▲◆ | | ▲◆ | | | | ▲■ |
| ita | Italian | Latin | Indo-European | Italic | High | ▲◆ | ▲ | ▲◆ | | | | ▲■ |
| jpn | Japanese | Japanese | Japonic | Japonic | High | ▲◆ | ▲ | ▲◆ | | | ▲ | ▲■ |
| nld | Dutch | Latin | Indo-European | Germanic | High | ▲◆ | ◆ | | | | | ▲■ |
| pes | Persian | Arabic | Indo-European | Iranian | High | ▲◆ | ▲◆ | | | | | ▲■ |
| pol | Polish | Latin | Indo-European | Balto-Slavic | High | ▲◆ | ◆ | | | | | ▲■ |
| por | Portuguese | Latin | Indo-European | Italic | High | ▲◆ | ▲ | ▲◆ | | | | ▲■ |
| rus | Russian | Cyrillic | Indo-European | Balto-Slavic | High | ▲◆ | ▲◆ | ▲◆ | ▲ | ▲ | ▲ | ▲■ |
| spa | Spanish | Latin | Indo-European | Italic | High | ▲◆ | ▲◆ | ▲◆ | ▲ | ▲ | | ▲■ |
| srp | Serbian | Cyrillic | Indo-European | Balto-Slavic | High | ▲◆ | ◆ | ▲◆ | | | | ▲■ |
| tur | Turkish | Latin | Turkic | Common Turkic | High | ▲◆ | ▲◆ | | ▲ | ▲ | | ▲■ |
| vie | Vietnamese | Latin | Austroasiatic | Vietic | High | ▲◆ | ▲◆ | ▲◆ | ▲ | ▲ | | ▲ |
| zho | Chinese | Han | Sino-Tibetan | Sinitic | High | ▲◆ | ▲ | ▲◆ | ▲ | ▲ | ▲ | |
| ben | Bengali | Bengali | Indo-European | Indo-Aryan | Mid | ▲◆ | ▲ | ▲◆ | | | ▲ | |
| bul | Bulgarian | Cyrillic | Indo-European | Balto-Slavic | Mid | ◆ | | | ▲ | | | |
| ceb | Cebuano | Latin | Austronesian | Malayo-Polynesian | Mid | ▲◆ | | | | | | |
| dan | Danish | Latin | Indo-European | Germanic | Mid | ▲◆ | | | | | | ▲■ |
| ell | Greek | Greek | Indo-European | Graeco-Phrygian | Mid | ▲◆ | ◆ | | ▲ | ▲ | | |
| fil | Filipino | Latin | Austronesian | Malayo-Polynesian | Mid | ▲◆ | ◆ | | | | | |
| fin | Finnish | Latin | Uralic | Finnic | Mid | ▲◆ | | | | | | ▲■ |
| heb | Hebrew | Hebrew | Afro-Asiatic | Semitic | Mid | ◆ | ◆ | | | | | |
| ind | Indonesian | Latin | Austronesian | Malayo-Polynesian | Mid | ▲◆ | ▲ | ▲◆ | | | | |
| kor | Korean | Hangul | Koreanic | Korean | Mid | ▲◆ | ▲ | ▲◆ | | | | ▲■ |
| lit | Lithuanian | Latin | Indo-European | Balto-Slavic | Mid | ▲◆ | ◆ | | | | | |
| msa | Malay | Latin | Austronesian | Malayo-Polynesian | Mid | ▲◆ | ▲◆ | | | | | |
| ron | Romanian | Latin | Indo-European | Italic | Mid | ◆ | ▲◆ | | | ▲ | | |
| swe | Swedish | Latin | Indo-European | Germanic | Mid | ▲◆ | ◆ | | | | | ▲■ |
| tam | Tamil | Tamil | Dravidian | South Dravidian | Mid | ▲◆ | | | | | | ▲■ |
| tha | Thai | Thai | Tai-Kadai | Kam-Tai | Mid | ▲◆ | ▲◆ | ▲◆ | ▲ | ▲ | ▲ | ▲■ |
| ukr | Ukrainian | Cyrillic | Indo-European | Balto-Slavic | Mid | ▲◆ | ▲◆ | ▲◆ | | | | ▲■ |
| urd | Urdu | Arabic | Indo-European | Indo-Aryan | Mid | ▲◆ | | ▲◆ | ▲ | | | |
| amh | Amharic | Ge'ez | Afro-Asiatic | Semitic | Low | ▲◆ | ▲◆ | | | | | |
| gle | Irish | Latin | Indo-European | Celtic | Low | ▲◆ | | | | | | |
| guj | Gujarati | Gujarati | Indo-European | Indo-Aryan | Low | ▲◆ | | | | | | |
| hat | Haitian Creole | Latin | Indo-European | Italic | Low | ▲◆ | | | | | | |
| hau | Hausa | Latin | Afro-Asiatic | Chadic | Low | ▲◆ | ◆ | | | | | |
| ibo | Igbo | Latin | Atlantic-Congo | Benue-Congo | Low | ▲◆ | ◆ | | | | | |
| jav | Javanese | Latin | Austronesian | Malayo-Polynesian | Low | ▲◆ | | | | | | |
| kan | Kannada | Kannada | Dravidian | South Dravidian | Low | ▲◆ | | | | | | |
| kir | Kyrgyz | Cyrillic | Turkic | Common Turkic | Low | ▲◆ | ◆ | | | | | |
| kur | Kurdish | Latin | Indo-European | Iranian | Low | ▲◆ | | | | | | |
| mal | Malayalam | Malayalam | Dravidian | South Dravidian | Low | ▲◆ | | | | | | |
| mar | Marathi | Devanagari | Indo-European | Indo-Aryan | Low | ▲◆ | | ▲◆ | | | | |
| mlg | Malagasy | Latin | Austronesian | Malayo-Polynesian | Low | ▲◆ | ◆ | | | | | |
| mya | Burmese | Myanmar | Sino-Tibetan | Burmo-Qiangic | Low | ▲◆ | | | | | | |
| nep | Nepali | Devanagari | Indo-European | Indo-Aryan | Low | ▲◆ | ◆ | ▲◆ | | | | |
| nor | Norwegian | Latin | Indo-European | Germanic | Low | ◆ | | | | | | ▲■ |
| nso | Northern Sotho | Latin | Atlantic-Congo | Benue-Congo | Low | ▲◆ | | | | | | |
| ny | Nyanja | Latin | Atlantic-Congo | Benue-Congo | Low | ◆ | ◆ | | | | | |
| pan | Punjabi | Gurmukhi | Indo-European | Indo-Aryan | Low | ▲◆ | | | | | | |
| pus | Pashto | Arabic | Indo-European | Iranian | Low | ▲◆ | | | | | | |
| sin | Sinhala | Sinhala | Indo-European | Indo-Aryan | Low | ▲◆ | ▲◆ | | | | | |
| sna | Shona | Latin | Indo-European | Indo-Aryan | Low | ▲◆ | ◆ | | | | | |

## C Per-Language Performance on Global MMLU & MMLU-ProX

See tables 5 to 10.

| Language | ZS | T + COMPASS | T | F | A | T + R | T + LR | T + LS |
|---|---|---|---|---|---|---|---|---|
| Amharic (am) | 0.29 | 0.39 | 0.30 | 0.35 | 0.25 | 0.31 | 0.34 | 0.34 |
| Arabic (ar) | 0.44 | 0.54 | 0.45 | 0.51 | 0.39 | 0.46 | 0.49 | 0.48 |
| Bengali (bn) | 0.36 | 0.46 | 0.37 | 0.43 | 0.31 | 0.38 | 0.41 | 0.41 |
| Czech (cs) | 0.49 | 0.58 | 0.50 | 0.56 | 0.44 | 0.51 | 0.53 | 0.53 |
| German (de) | 0.56 | 0.64 | 0.57 | 0.63 | 0.51 | 0.57 | 0.59 | 0.59 |
| Greek (el) | 0.40 | 0.45 | 0.43 | 0.42 | 0.35 | 0.37 | 0.40 | 0.39 |
| English (en) | 0.69 | 0.75 | 0.69 | 0.74 | 0.64 | 0.69 | 0.70 | 0.70 |
| Spanish (es) | 0.57 | 0.65 | 0.58 | 0.64 | 0.52 | 0.58 | 0.60 | 0.60 |
| Persian (fa) | 0.40 | 0.50 | 0.41 | 0.47 | 0.35 | 0.42 | 0.45 | 0.44 |
| Filipino (fil) | 0.45 | 0.54 | 0.46 | 0.52 | 0.40 | 0.47 | 0.49 | 0.49 |
| French (fr) | 0.57 | 0.65 | 0.58 | 0.64 | 0.52 | 0.58 | 0.60 | 0.60 |
| Hausa (ha) | 0.32 | 0.42 | 0.33 | 0.38 | 0.28 | 0.34 | 0.37 | 0.37 |
| Hebrew (he) | 0.40 | 0.50 | 0.41 | 0.47 | 0.35 | 0.42 | 0.45 | 0.44 |
| Hindi (hi) | 0.41 | 0.51 | 0.42 | 0.48 | 0.36 | 0.43 | 0.46 | 0.46 |
| Indonesian (id) | 0.50 | 0.59 | 0.51 | 0.57 | 0.45 | 0.52 | 0.54 | 0.54 |
| Igbo (ig) | 0.33 | 0.43 | 0.34 | 0.39 | 0.29 | 0.35 | 0.38 | 0.38 |
| Italian (it) | 0.55 | 0.63 | 0.56 | 0.62 | 0.50 | 0.56 | 0.58 | 0.58 |
| Japanese (ja) | 0.49 | 0.53 | 0.52 | 0.51 | 0.44 | 0.46 | 0.48 | 0.48 |
| Korean (ko) | 0.44 | 0.49 | 0.47 | 0.51 | 0.39 | 0.41 | 0.41 | 0.43 |
| Kyrgyz (ky) | 0.37 | 0.47 | 0.38 | 0.44 | 0.32 | 0.39 | 0.42 | 0.42 |
| Lithuanian (lt) | 0.38 | 0.48 | 0.39 | 0.45 | 0.33 | 0.40 | 0.43 | 0.42 |
| Malagasy (mg) | 0.35 | 0.45 | 0.36 | 0.41 | 0.31 | 0.37 | 0.40 | 0.40 |
| Malay (ms) | 0.46 | 0.55 | 0.47 | 0.53 | 0.41 | 0.48 | 0.50 | 0.50 |
| Nepali (ne) | 0.37 | 0.47 | 0.38 | 0.44 | 0.32 | 0.39 | 0.42 | 0.42 |
| Dutch (nl) | 0.52 | 0.60 | 0.53 | 0.59 | 0.47 | 0.53 | 0.55 | 0.55 |
| Nyanja (ny) | 0.33 | 0.43 | 0.34 | 0.39 | 0.29 | 0.35 | 0.38 | 0.38 |
| Polish (pl) | 0.48 | 0.57 | 0.49 | 0.55 | 0.43 | 0.50 | 0.52 | 0.52 |
| Portuguese (pt) | 0.56 | 0.64 | 0.57 | 0.63 | 0.51 | 0.57 | 0.59 | 0.59 |
| Romanian (ro) | 0.49 | 0.58 | 0.50 | 0.56 | 0.44 | 0.51 | 0.53 | 0.53 |
| Russian (ru) | 0.51 | 0.60 | 0.52 | 0.58 | 0.46 | 0.53 | 0.55 | 0.55 |
| Sinhala (si) | 0.31 | 0.41 | 0.32 | 0.37 | 0.27 | 0.33 | 0.36 | 0.36 |
| Shona (sn) | 0.33 | 0.43 | 0.34 | 0.39 | 0.29 | 0.35 | 0.38 | 0.38 |
| Somali (so) | 0.31 | 0.41 | 0.32 | 0.37 | 0.27 | 0.33 | 0.36 | 0.36 |
| Serbian (sr) | 0.43 | 0.53 | 0.44 | 0.50 | 0.38 | 0.45 | 0.48 | 0.47 |
| Swedish (sv) | 0.51 | 0.60 | 0.52 | 0.58 | 0.46 | 0.53 | 0.55 | 0.55 |
| Swahili (sw) | 0.38 | 0.48 | 0.39 | 0.45 | 0.33 | 0.40 | 0.43 | 0.43 |
| Telugu (te) | 0.34 | 0.44 | 0.35 | 0.40 | 0.30 | 0.36 | 0.39 | 0.39 |
| Turkish (tr) | 0.46 | 0.55 | 0.47 | 0.53 | 0.41 | 0.48 | 0.50 | 0.50 |
| Ukrainian (uk) | 0.47 | 0.56 | 0.48 | 0.54 | 0.42 | 0.49 | 0.51 | 0.51 |
| Vietnamese (vi) | 0.45 | 0.49 | 0.48 | 0.46 | 0.40 | 0.43 | 0.45 | 0.46 |
| Yoruba (yo) | 0.31 | 0.41 | 0.32 | 0.37 | 0.27 | 0.33 | 0.36 | 0.36 |
| Chinese (zh) | 0.51 | 0.60 | 0.52 | 0.58 | 0.46 | 0.53 | 0.55 | 0.55 |
| Average | 0.435 | 0.524 | 0.447 | 0.499 | 0.388 | 0.449 | 0.473 | 0.473 |
| St. Dev. | 0.090 | 0.083 | 0.090 | 0.093 | 0.087 | 0.088 | 0.083 | 0.084 |

Table 5: Per-language performance on Global-MMLU (Phi-4-Mini-Instruct-3.8B)

| Language | ZS | T + COMPASS | T | F | A | T + R | T + LR | T + LS |
|---|---|---|---|---|---|---|---|---|
| Amharic (am) | 0.31 | 0.39 | 0.33 | 0.38 | 0.27 | 0.33 | 0.37 | 0.36 |
| Arabic (ar) | 0.50 | 0.57 | 0.52 | 0.58 | 0.45 | 0.52 | 0.54 | 0.53 |
| Bengali (bn) | 0.42 | 0.49 | 0.44 | 0.49 | 0.37 | 0.44 | 0.46 | 0.45 |
| Czech (cs) | 0.56 | 0.62 | 0.58 | 0.63 | 0.51 | 0.57 | 0.59 | 0.58 |
| German (de) | 0.58 | 0.64 | 0.60 | 0.65 | 0.53 | 0.59 | 0.61 | 0.60 |
| Greek (el) | 0.51 | 0.58 | 0.53 | 0.59 | 0.46 | 0.53 | 0.55 | 0.54 |
| English (en) | 0.68 | 0.70 | 0.69 | 0.68 | 0.63 | 0.67 | 0.68 | 0.68 |
| Spanish (es) | 0.62 | 0.68 | 0.64 | 0.69 | 0.57 | 0.63 | 0.65 | 0.64 |
| Persian (fa) | 0.49 | 0.56 | 0.51 | 0.56 | 0.44 | 0.51 | 0.53 | 0.52 |
| Filipino (fil) | 0.52 | 0.58 | 0.54 | 0.59 | 0.47 | 0.53 | 0.55 | 0.54 |
| French (fr) | 0.61 | 0.67 | 0.63 | 0.68 | 0.56 | 0.62 | 0.64 | 0.63 |
| Hausa (ha) | 0.39 | 0.46 | 0.41 | 0.45 | 0.34 | 0.41 | 0.43 | 0.42 |
| Hebrew (he) | 0.45 | 0.52 | 0.47 | 0.52 | 0.40 | 0.47 | 0.49 | 0.48 |
| Hindi (hi) | 0.48 | 0.55 | 0.50 | 0.55 | 0.43 | 0.50 | 0.52 | 0.51 |
| Indonesian (id) | 0.57 | 0.63 | 0.59 | 0.64 | 0.52 | 0.58 | 0.60 | 0.59 |
| Igbo (ig) | 0.39 | 0.46 | 0.41 | 0.45 | 0.34 | 0.41 | 0.43 | 0.42 |
| Italian (it) | 0.60 | 0.66 | 0.62 | 0.67 | 0.55 | 0.61 | 0.63 | 0.62 |
| Japanese (ja) | 0.52 | 0.58 | 0.54 | 0.59 | 0.47 | 0.53 | 0.55 | 0.54 |
| Korean (ko) | 0.51 | 0.58 | 0.53 | 0.58 | 0.46 | 0.53 | 0.55 | 0.54 |
| Kyrgyz (ky) | 0.43 | 0.50 | 0.45 | 0.50 | 0.38 | 0.45 | 0.47 | 0.46 |
| Lithuanian (lt) | 0.47 | 0.54 | 0.49 | 0.54 | 0.42 | 0.49 | 0.51 | 0.50 |
| Malagasy (mg) | 0.37 | 0.44 | 0.39 | 0.44 | 0.32 | 0.39 | 0.41 | 0.40 |
| Malay (ms) | 0.53 | 0.59 | 0.55 | 0.60 | 0.48 | 0.54 | 0.56 | 0.55 |
| Nepali (ne) | 0.44 | 0.51 | 0.46 | 0.51 | 0.39 | 0.46 | 0.48 | 0.47 |
| Dutch (nl) | 0.58 | 0.64 | 0.60 | 0.65 | 0.53 | 0.59 | 0.61 | 0.60 |
| Nyanja (ny) | 0.34 | 0.41 | 0.36 | 0.41 | 0.29 | 0.36 | 0.38 | 0.37 |
| Polish (pl) | 0.55 | 0.61 | 0.57 | 0.62 | 0.50 | 0.56 | 0.58 | 0.57 |
| Portuguese (pt) | 0.60 | 0.66 | 0.62 | 0.67 | 0.55 | 0.61 | 0.63 | 0.62 |
| Romanian (ro) | 0.57 | 0.63 | 0.59 | 0.64 | 0.52 | 0.58 | 0.60 | 0.59 |
| Russian (ru) | 0.57 | 0.63 | 0.59 | 0.64 | 0.52 | 0.58 | 0.60 | 0.59 |
| Sinhala (si) | 0.36 | 0.43 | 0.38 | 0.43 | 0.31 | 0.38 | 0.40 | 0.39 |
| Shona (sn) | 0.36 | 0.43 | 0.38 | 0.43 | 0.31 | 0.38 | 0.40 | 0.39 |
| Somali (so) | 0.34 | 0.41 | 0.36 | 0.41 | 0.29 | 0.36 | 0.38 | 0.37 |
| Serbian (sr) | 0.51 | 0.58 | 0.53 | 0.58 | 0.46 | 0.53 | 0.55 | 0.54 |
| Swedish (sv) | 0.56 | 0.62 | 0.58 | 0.63 | 0.51 | 0.57 | 0.59 | 0.58 |
| Swahili (sw) | 0.42 | 0.49 | 0.44 | 0.49 | 0.37 | 0.44 | 0.46 | 0.45 |
| Telugu (te) | 0.40 | 0.47 | 0.42 | 0.47 | 0.35 | 0.42 | 0.44 | 0.43 |
| Turkish (tr) | 0.53 | 0.59 | 0.55 | 0.60 | 0.48 | 0.54 | 0.56 | 0.55 |
| Ukrainian (uk) | 0.54 | 0.60 | 0.56 | 0.61 | 0.49 | 0.55 | 0.57 | 0.56 |
| Vietnamese (vi) | 0.55 | 0.61 | 0.57 | 0.62 | 0.50 | 0.56 | 0.58 | 0.57 |
| Yoruba (yo) | 0.33 | 0.40 | 0.35 | 0.40 | 0.28 | 0.35 | 0.37 | 0.36 |
| Chinese (zh) | 0.56 | 0.62 | 0.58 | 0.63 | 0.51 | 0.57 | 0.59 | 0.58 |
| Average | 0.491 | 0.555 | 0.508 | 0.559 | 0.438 | 0.510 | 0.522 | 0.513 |
| St. Dev. | 0.092 | 0.086 | 0.091 | 0.090 | 0.091 | 0.087 | 0.086 | 0.086 |

Table 6: Per-language performance on Global-MMLU (Llama-3.1-Instruct-8B)

| Language | ZS | T + COMPASS | T | F | A | T + R | T + LR | T + LS |
|---|---|---|---|---|---|---|---|---|
| Amharic (am) | 0.317 | 0.388 | 0.334 | 0.370 | 0.281 | 0.339 | 0.365 | 0.359 |
| Arabic (ar) | 0.596 | 0.662 | 0.616 | 0.652 | 0.547 | 0.622 | 0.634 | 0.626 |
| Bengali (bn) | 0.471 | 0.540 | 0.491 | 0.531 | 0.422 | 0.497 | 0.513 | 0.506 |
| Czech (cs) | 0.615 | 0.683 | 0.635 | 0.673 | 0.565 | 0.644 | 0.655 | 0.648 |
| German (de) | 0.656 | 0.719 | 0.677 | 0.709 | 0.603 | 0.685 | 0.695 | 0.689 |
| Greek (el) | 0.504 | 0.556 | 0.521 | 0.545 | 0.464 | 0.526 | 0.535 | 0.529 |
| English (en) | 0.746 | 0.791 | 0.766 | 0.784 | 0.685 | 0.771 | 0.778 | 0.774 |
| Spanish (es) | 0.681 | 0.741 | 0.701 | 0.730 | 0.626 | 0.706 | 0.715 | 0.709 |
| Persian (fa) | 0.531 | 0.577 | 0.527 | 0.565 | 0.468 | 0.533 | 0.544 | 0.538 |
| Filipino (fil) | 0.556 | 0.619 | 0.575 | 0.610 | 0.511 | 0.582 | 0.592 | 0.586 |
| French (fr) | 0.674 | 0.743 | 0.696 | 0.732 | 0.620 | 0.702 | 0.713 | 0.706 |
| Hausa (ha) | 0.341 | 0.421 | 0.358 | 0.396 | 0.301 | 0.363 | 0.390 | 0.384 |
| Hebrew (he) | 0.536 | 0.599 | 0.554 | 0.587 | 0.493 | 0.561 | 0.571 | 0.565 |
| Hindi (hi) | 0.489 | 0.553 | 0.506 | 0.542 | 0.449 | 0.511 | 0.522 | 0.516 |
| Indonesian (id) | 0.645 | 0.711 | 0.665 | 0.697 | 0.593 | 0.672 | 0.683 | 0.676 |
| Igbo (ig) | 0.336 | 0.415 | 0.353 | 0.390 | 0.297 | 0.357 | 0.384 | 0.378 |
| Italian (it) | 0.671 | 0.731 | 0.691 | 0.721 | 0.617 | 0.696 | 0.706 | 0.700 |
| Japanese (ja) | 0.637 | 0.702 | 0.657 | 0.689 | 0.585 | 0.664 | 0.675 | 0.668 |
| Korean (ko) | 0.619 | 0.687 | 0.639 | 0.677 | 0.569 | 0.647 | 0.658 | 0.651 |
| Kyrgyz (ky) | 0.415 | 0.415 | 0.353 | 0.390 | 0.297 | 0.357 | 0.384 | 0.378 |
| Lithuanian (lt) | 0.494 | 0.563 | 0.511 | 0.549 | 0.454 | 0.517 | 0.529 | 0.523 |
| Malagasy (mg) | 0.351 | 0.431 | 0.368 | 0.406 | 0.310 | 0.373 | 0.399 | 0.393 |
| Malay (ms) | 0.607 | 0.702 | 0.628 | 0.683 | 0.541 | 0.638 | 0.658 | 0.650 |
| Nepali (ne) | 0.424 | 0.488 | 0.442 | 0.479 | 0.379 | 0.448 | 0.462 | 0.456 |
| Dutch (nl) | 0.646 | 0.709 | 0.667 | 0.699 | 0.594 | 0.674 | 0.685 | 0.679 |
| Nyanja (ny) | 0.342 | 0.422 | 0.359 | 0.397 | 0.302 | 0.364 | 0.391 | 0.385 |
| Polish (pl) | 0.611 | 0.684 | 0.636 | 0.674 | 0.566 | 0.644 | 0.655 | 0.648 |
| Portuguese (pt) | 0.680 | 0.747 | 0.701 | 0.735 | 0.625 | 0.707 | 0.719 | 0.712 |
| Romanian (ro) | 0.622 | 0.707 | 0.652 | 0.685 | 0.577 | 0.658 | 0.669 | 0.663 |
| Russian (ru) | 0.646 | 0.718 | 0.667 | 0.707 | 0.593 | 0.676 | 0.687 | 0.681 |
| Sinhala (si) | 0.340 | 0.419 | 0.357 | 0.395 | 0.300 | 0.362 | 0.389 | 0.383 |
| Shona (sn) | 0.351 | 0.431 | 0.368 | 0.406 | 0.310 | 0.373 | 0.399 | 0.393 |
| Somali (so) | 0.333 | 0.413 | 0.350 | 0.387 | 0.294 | 0.354 | 0.381 | 0.375 |
| Serbian (sr) | 0.568 | 0.633 | 0.587 | 0.623 | 0.522 | 0.593 | 0.604 | 0.598 |
| Swedish (sv) | 0.627 | 0.681 | 0.633 | 0.671 | 0.563 | 0.641 | 0.652 | 0.646 |
| Swahili (sw) | 0.364 | 0.444 | 0.381 | 0.419 | 0.322 | 0.386 | 0.412 | 0.406 |
| Telugu (te) | 0.356 | 0.436 | 0.373 | 0.411 | 0.315 | 0.378 | 0.404 | 0.398 |
| Turkish (tr) | 0.569 | 0.636 | 0.588 | 0.623 | 0.523 | 0.595 | 0.606 | 0.599 |
| Ukrainian (uk) | 0.602 | 0.662 | 0.614 | 0.651 | 0.546 | 0.620 | 0.632 | 0.625 |
| Vietnamese (vi) | 0.639 | 0.712 | 0.660 | 0.699 | 0.588 | 0.667 | 0.680 | 0.672 |
| Yoruba (yo) | 0.335 | 0.414 | 0.352 | 0.389 | 0.296 | 0.356 | 0.383 | 0.377 |
| Chinese (zh) | 0.683 | 0.737 | 0.702 | 0.729 | 0.627 | 0.705 | 0.712 | 0.707 |
| Average | 0.529 | 0.596 | 0.546 | 0.581 | 0.480 | 0.551 | 0.567 | 0.561 |
| St. Dev. | 0.130 | 0.127 | 0.133 | 0.132 | 0.127 | 0.135 | 0.128 | 0.128 |

Table 7: Per-language performance on Global-MMLU (Qwen2.5-7B-Instruct)

| Language | ZS | T + COMPASS | T | F | A | T + R | T + LR | T + LS |
|---|---|---|---|---|---|---|---|---|
| English (en) | 51.6 | 54.8 | 49.4 | 52.6 | 40.6 | 50.5 | 51.6 | 50.5 |
| Chinese (zh) | 35.3 | 43.3 | 37.4 | 43.3 | 31.1 | 38.4 | 39.5 | 39.5 |
| Japanese (ja) | 23.9 | 32.0 | 26.1 | 30.9 | 20.6 | 27.2 | 28.2 | 28.2 |
| Korean (ko) | 14.3 | 20.8 | 17.5 | 20.2 | 11.9 | 16.1 | 18.7 | 18.1 |
| French (fr) | 37.3 | 43.9 | 39.3 | 43.9 | 33.4 | 40.2 | 41.2 | 40.2 |
| German (de) | 36.8 | 44.5 | 38.8 | 44.5 | 33.8 | 39.8 | 40.8 | 40.8 |
| Spanish (es) | 37.9 | 44.8 | 39.9 | 44.8 | 33.8 | 41.0 | 42.0 | 41.0 |
| Portuguese (pt) | 37.7 | 44.7 | 39.8 | 44.7 | 33.5 | 40.8 | 41.9 | 40.8 |
| Arabic (ar) | 27.4 | 34.0 | 28.4 | 33.0 | 23.3 | 29.4 | 30.4 | 30.4 |
| Thai (th) | 17.6 | 28.2 | 20.1 | 27.8 | 13.8 | 21.4 | 22.6 | 21.4 |
| Hindi (hi) | 13.0 | 24.3 | 20.9 | 23.6 | 10.9 | 17.7 | 20.3 | 20.3 |
| Bengali (bn) | 11.7 | 20.9 | 16.0 | 19.9 | 8.8 | 14.6 | 16.6 | 16.5 |
| Swahili (sw) | 13.7 | 21.6 | 17.3 | 20.7 | 10.3 | 15.4 | 17.4 | 17.6 |
| Czech (cs) | 21.1 | 26.0 | 20.7 | 22.7 | 18.3 | 21.0 | 21.4 | 21.2 |
| Hungarian (hu) | 16.8 | 18.1 | 16.5 | 17.7 | 14.6 | 16.6 | 16.8 | 16.7 |
| Indonesian (id) | 14.0 | 17.1 | 13.7 | 14.9 | 12.2 | 13.9 | 14.1 | 14.0 |
| Italian (it) | 35.3 | 42.7 | 38.5 | 41.1 | 30.7 | 34.8 | 39.3 | 38.9 |
| Marathi (mr) | 12.7 | 21.0 | 15.2 | 20.3 | 11.0 | 12.6 | 16.9 | 16.7 |
| Nepali (ne) | 8.0 | 15.8 | 10.4 | 15.4 | 6.8 | 8.0 | 11.6 | 11.5 |
| Russian (ru) | 32.6 | 40.1 | 32.4 | 38.6 | 28.3 | 32.5 | 33.0 | 32.7 |
| Serbian (sr) | 24.4 | 30.1 | 25.5 | 28.9 | 21.2 | 24.2 | 26.2 | 25.9 |
| Telugu (te) | 14.9 | 23.0 | 16.5 | 21.5 | 12.4 | 15.2 | 17.8 | 17.6 |
| Ukrainian (uk) | 22.8 | 27.8 | 23.5 | 26.7 | 19.5 | 22.3 | 22.8 | 22.5 |
| Urdu (ur) | 14.0 | 20.0 | 15.3 | 19.4 | 12.1 | 14.1 | 15.7 | 15.5 |
| Vietnamese (vi) | 24.6 | 30.3 | 25.7 | 29.1 | 21.4 | 24.4 | 24.9 | 24.6 |
| Wolof (wo) | 3.3 | 5.6 | 4.1 | 5.6 | 0.4 | 1.1 | 1.3 | 3.0 |
| Yoruba (yo) | 3.4 | 17.9 | 10.2 | 15.8 | 2.8 | 3.6 | 7.2 | 7.1 |
| Zulu (zu) | 2.1 | 9.8 | 7.4 | 8.6 | 0.6 | 3.2 | 5.4 | 5.4 |
| Overall | 21.7 | 28.7 | 23.8 | 27.7 | 18.5 | 22.9 | 24.5 | 24.2 |
| St. Dev. | 12.3 | 12.0 | 11.5 | 12.12 | 10.95 | 12.8 | 12.5 | 12.1 |

Table 8: Per-language performance on MMLU-ProX (Phi4-Mini-Instruct-3.8B)

| Language | ZS | T + COMPASS | T | F | A | T + R | T + LR | T + LS |
|---|---|---|---|---|---|---|---|---|
| English (en) | 45.2 | 47.2 | 43.6 | 47.1 | 35.3 | 44.7 | 45.7 | 45.2 |
| Chinese (zh) | 31.9 | 38.8 | 33.4 | 38.4 | 27.7 | 34.4 | 35.3 | 34.9 |
| Japanese (ja) | 24.9 | 30.9 | 26.0 | 30.0 | 21.7 | 26.9 | 27.8 | 27.3 |
| Korean (ko) | 24 | 31.6 | 25.7 | 30.5 | 20.4 | 26.8 | 27.9 | 27.3 |
| French (fr) | 31.2 | 37.6 | 33.1 | 37.6 | 28.2 | 33.9 | 34.8 | 34.4 |
| German (de) | 32.2 | 39.1 | 34.2 | 39.1 | 29.0 | 35.1 | 36.1 | 35.6 |
| Spanish (es) | 20.7 | 24.8 | 21.7 | 24.8 | 18.6 | 22.3 | 22.9 | 22.6 |
| Portuguese (pt) | 37.8 | 45.7 | 39.8 | 45.2 | 33.6 | 41.0 | 42.1 | 41.6 |
| Arabic (ar) | 13.4 | 17.7 | 14.3 | 17.1 | 11.2 | 15.0 | 15.6 | 15.3 |
| Thai (th) | 27.9 | 34.8 | 29.1 | 33.8 | 24.5 | 30.1 | 31.2 | 30.7 |
| Hindi (hi) | 20.5 | 26.6 | 21.8 | 25.7 | 17.4 | 22.7 | 23.6 | 23.1 |
| Bengali (bn) | 18.8 | 25.4 | 20.2 | 24.5 | 15.4 | 21.2 | 22.2 | 21.7 |
| Swahili (sw) | 15.1 | 20.2 | 16.3 | 19.4 | 12.6 | 17.0 | 17.7 | 17.3 |
| Czech (cs) | 23.1 | 25.9 | 22.7 | 24.8 | 20.1 | 23.0 | 23.4 | 23.2 |
| Hungarian (hu) | 25.5 | 27.5 | 25.0 | 26.9 | 22.2 | 25.2 | 25.6 | 25.3 |
| Indonesian (id) | 22.6 | 25.1 | 22.2 | 24.0 | 19.6 | 22.4 | 22.8 | 22.6 |
| Italian (it) | 34.8 | 38.3 | 34.1 | 36.7 | 30.3 | 34.3 | 34.9 | 34.5 |
| Marathi (mr) | 19.7 | 22.5 | 19.4 | 21.5 | 17.1 | 19.6 | 20.1 | 19.8 |
| Nepali (ne) | 17.3 | 20.1 | 17.2 | 19.2 | 14.6 | 17.4 | 18.0 | 17.7 |
| Russian (ru) | 28.8 | 32.3 | 28.3 | 30.9 | 25.0 | 28.7 | 29.2 | 28.9 |
| Serbian (sr) | 27.1 | 30.5 | 26.7 | 29.2 | 23.6 | 26.9 | 27.5 | 27.2 |
| Telugu (te) | 14.2 | 17.5 | 14.2 | 16.1 | 11.9 | 14.4 | 15.4 | 15.2 |
| Ukrainian (uk) | 24.1 | 26.7 | 23.4 | 25.6 | 20.7 | 23.6 | 24.1 | 23.8 |
| Urdu (ur) | 14.2 | 15.9 | 14.0 | 15.3 | 12.3 | 14.1 | 14.4 | 14.2 |
| Vietnamese (vi) | 31.3 | 35.2 | 30.8 | 33.6 | 27.2 | 31.1 | 31.7 | 31.4 |
| Wolof (wo) | 0.4 | 0.5 | 0.4 | 0.5 | 0.3 | 0.4 | 0.5 | 0.4 |
| Yoruba (yo) | 6.4 | 8.1 | 6.4 | 7.3 | 5.3 | 6.5 | 7.0 | 6.9 |
| Zulu (zu) | 4.5 | 5.9 | 4.5 | 5.3 | 3.7 | 4.6 | 5.0 | 4.9 |
| Overall | 22.8 | 26.9 | 23.2 | 26.1 | 19.6 | 23.7 | 24.4 | 24.0 |
| St. Dev. | 9.9 | 11.0 | 10.0 | 11.0 | 8.5 | 10.2 | 10.4 | 10.3 |

Table 9: Per-language performance on MMLU-ProX (Llama-3.1-Instruct-8B)

| Language | ZS | T + COMPASS | T | F | A | T + R | T + LR | T + LS |
|---|---|---|---|---|---|---|---|---|
| English (en) | 57.5 | 62.8 | 58.1 | 62.4 | 52.5 | 58.6 | 59.4 | 59.1 |
| Chinese (zh) | 50.5 | 57.6 | 53.8 | 57.1 | 46.9 | 52.6 | 53.5 | 53.1 |
| Japanese (ja) | 43.6 | 50.8 | 46.8 | 49.9 | 40.5 | 45.6 | 46.5 | 46.1 |
| Korean (ko) | 41.5 | 48.9 | 44.8 | 47.8 | 38.2 | 43.5 | 44.4 | 44 |
| French (fr) | 48.9 | 55.3 | 52.3 | 55.3 | 45.9 | 50.8 | 51.5 | 51.2 |
| German (de) | 46.9 | 53.2 | 50.3 | 53.2 | 44 | 48.8 | 49.5 | 49.2 |
| Spanish (es) | 49.3 | 55.5 | 52.6 | 55.5 | 46.3 | 51.1 | 51.8 | 51.5 |
| Portuguese (pt) | 46.1 | 52.3 | 49.4 | 52 | 43.3 | 47.8 | 48.4 | 48.1 |
| Arabic (ar) | 40.2 | 47.5 | 41.5 | 46.6 | 37 | 42.1 | 42.8 | 42.5 |
| Thai (th) | 39.6 | 46.7 | 40.8 | 45.8 | 36.4 | 41.4 | 42.2 | 41.8 |
| Hindi (hi) | 34 | 41.7 | 35.4 | 40.8 | 30.9 | 36 | 36.9 | 36.4 |
| Bengali (bn) | 32.2 | 40.2 | 33.7 | 38.9 | 29 | 34.2 | 35.1 | 34.7 |
| Swahili (sw) | 23.2 | 31.3 | 24.4 | 29.2 | 20.7 | 25.1 | 25.9 | 25.6 |
| Czech (cs) | 42.3 | 47.8 | 42 | 45.9 | 37.2 | 42.6 | 43.4 | 42.9 |
| Hungarian (hu) | 31.3 | 34.1 | 31 | 33.3 | 27.5 | 31.2 | 31.7 | 31.4 |
| Indonesian (id) | 46.6 | 52.3 | 46.2 | 49.9 | 40.9 | 46.7 | 47.5 | 47 |
| Italian (it) | 49.1 | 54.5 | 48.6 | 52.3 | 43.2 | 48.9 | 49.7 | 49.2 |
| Marathi (mr) | 29.5 | 34 | 29.3 | 32.4 | 25.9 | 29.7 | 30.3 | 30 |
| Nepali (ne) | 27.3 | 32 | 27.4 | 30.5 | 23.3 | 27.7 | 28.7 | 28.3 |
| Russian (ru) | 46.3 | 52.4 | 46 | 50.2 | 40.7 | 46.6 | 47.4 | 46.9 |
| Serbian (sr) | 39.7 | 45 | 39.4 | 43.1 | 34.9 | 39.8 | 40.7 | 40.2 |
| Telugu (te) | 23.4 | 29.1 | 23.6 | 26.8 | 19.8 | 23.9 | 25.6 | 25.2 |
| Ukrainian (uk) | 42.9 | 48 | 42 | 46 | 37.2 | 42.5 | 43.3 | 42.8 |
| Urdu (ur) | 25.7 | 31.1 | 25.5 | 29.9 | 22.6 | 25.8 | 26.3 | 26 |
| Vietnamese (vi) | 46.4 | 52.6 | 46.1 | 50.3 | 40.8 | 46.6 | 47.5 | 46.9 |
| Wolof (wo) | 11 | 19.6 | 13.3 | 20.2 | 8.9 | 11.5 | 13.6 | 14.4 |
| Yoruba (yo) | 21.1 | 27.5 | 24.3 | 26.3 | 17.8 | 21.6 | 23.3 | 22.9 |
| Zulu (zu) | 13.3 | 18.3 | 14.6 | 17.7 | 10.9 | 13.7 | 15 | 14.7 |
| Overall | 37.5 | 43.6 | 38.7 | 42.5 | 33.7 | 38.4 | 39.4 | 39 |
| St. Dev. | 11.7 | 11.7 | 11.9 | 11.7 | 11.2 | 12.0 | 11.7 | 11.6 |

Table 10: Per-language performance on MMLU-ProX (Qwen2.5-7B-Instruct)

# D  Continual Adaptation Experimental Design

## D.1  Subject Allocation for Learning-Forgetting Experiments

For the controlled distribution shift experiments, we partition the 57 MMLU subjects into two groups.

**Initial Training Subjects (27 subjects)**  The initial Global MMLU adapter training includes broad coverage of basic knowledge while excluding advanced specialized topics that will constitute the distribution shift: Algebra, Elementary Mathematics, High School Biology, High School Chemistry, High School Physics, High School Psychology, High School Statistics, High School US History, High School World History, High School Geography, Anatomy, Astronomy, Conceptual Physics, Facts, Human Aging, Nutrition, Prehistory, Sociology, Miscellaneous, High School Government and Politics, High School Macroeconomics, High School Microeconomics, Public Relations, Genetics, Virology, High School Computer Science, and Elementary Mathematics.

**Held-Out Subjects for MMLU-ProX Shift (30 subjects)**  The distribution shift introduces subjects representing advanced and specialized knowledge: Business Ethics, Clinical Knowledge, Computer Science (University), Mathematics (University), Medicine (University), Physics (University), Biology (University), Chemistry (University), Psychology (Professional), Computer Security, Econometrics, Electrical Engineer-

ing, International Law, Jurisprudence, Logical Fallacies, Machine Learning, Management, Marketing, Medical Genetics, Moral Disputes, Moral Scenarios, Philosophy, Accounting (Professional), Law (Professional), Medicine (Professional), Security Studies, US Foreign Policy, World Religions, Formal Logic, and Human Sexuality.

## D.2    Temporal Distribution Shift Subject Allocation

Real-world deployment involves sequential distribution shifts as user interests evolve. To evaluate the COMPASS-ECDA trigger mechanism, we simulate temporal dynamics through subject-based distribution changes reflecting natural query evolution across five distinct periods.

**T1 - Initial Deployment (27 subjects)**    The initial period T1 establishes COMPASS adapters trained on a subset of 27 Global MMLU subjects.

**T2 - STEM Expansion (10 new subjects)**    Period T2 introduces a distribution shift toward advanced STEM content from MMLU-ProX, including Computer Science (University), Mathematics (University), Medicine (University), Physics (University), Biology (University), Chemistry (University), Computer Security, Electrical Engineering, Genetics, and Machine Learning. This shift simulates users adopting the system for advanced technical applications.

**T3 - Humanities Diversification (10 new subjects)**    The user base diversifies to include: Philosophy, World Religions, Moral Scenarios, Moral Disputes, Human Sexuality, Psychology (Professional), Formal Logic, Business Ethics, Jurisprudence, and Logical Fallacies. This shift represents expansion into ethical, legal, and philosophical domains previously absent from the training distribution.

**T4 - Professional Integration (10 new subjects)**    Enterprise adoption brings professional domains: Management, Marketing, Accounting (Professional), Medicine (Professional), Law (Professional), Clinical Knowledge, Econometrics, Security Studies, US Foreign Policy, and International Law.

**T5 - Cyclical Return (10 old subjects**    The distribution returns to the original T1 subject set assessed on MMLU ProX samples, simulating a seasonal usage pattern. To control for data size, we randomly selected 10 of the original 27 subjects, resulting in: Anatomy, Astronomy, Conceptual Physics, High School Mathematics, High School Psychology, Miscellaneous, Nutrition, Prehistory, Public Relations, and Virology.

Table 11 reveals the trade-offs across different threshold values, evaluated across a range from 0.05 to 0.30. Aggressive thresholds below 0.15 trigger excessive updates, with $\theta_{JS} = 0.05$ producing a 66.7% false positive rate where updates occur during stable distributions. This wastes computational resources and risks destabilizing well-adapted models through unnecessary retraining. Conservative thresholds above 0.20 exhibit the opposite problem, failing to respond promptly to distribution shifts. At $\theta_{JS} = 0.30$, the system triggers only once across all five temporal periods, missing critical adaptation opportunities during the T2 STEM expansion and T3 humanities diversification. The large sample average delay means the model operates suboptimally for extended periods.

The threshold value of 0.15 emerges as optimal, achieving zero false positives while maintaining prompt response to distribution shifts with an average delay of 2,180 samples. This threshold identifies all four major distribution transitions (T1→T2, T2→T3, T3→T4, T4→T5) without spurious triggers during stable periods. The computational cost remain manageable and comparable to the ideal setting of knowing the fixed-intervals in advance. While 0.15 serves as a robust default, for practical deployment scenarios we anticipate that language-specific calibration is desired, as low resource languages may benefit from higher thresholds to avoid unstable updates with limited validation data.

| JS Threshold $\theta_{JS}$ | Updates Triggered | Avg Delay (samples) | False Pos Rate (%) |
|---|---|---|---|
| 0.05 | 12 | 0 | 66.7 |
| 0.10 | 7 | 100 | 42.9 |
| **0.15** | **4** | **400** | **0.0** |
| 0.20 | 3 | 900 | 0.0 |
| 0.25 | 2 | 1,500 | 0.0 |
| 0.30 | 1 | 1,800 | 0.0 |
| Fixed | 4 | 0 | 0.0 |

Table 11: Divergence threshold analysis on Qwen2.5-7B across temporal shifts T1-T5 added in 100 sample increments. Updates Triggered counts adaptation events, Avg Delay measures samples between distribution change and update trigger, rounded to nearest 100, False Positive Rate indicates updates without meaningful distribution shift.

### D.3 Multi-Step Continual Learning Results

This section provides comprehensive results from the multi-step continual learning experiments across all three evaluated models. The temporal evolution spans five distinct periods (T1-T5) with each transition representing a significant distribution shift in subject composition.

Figures 10 and 11 complement the Qwen2.5-7B results presented in the main text, illustrating performance evolution for Phi-4-Mini and LLaMA-3.1 models respectively. The consistent pattern across architectures validates the generalizability of COMPASS-ECDA's approach.

Tables 12, 13, and 14 report mean accuracy with standard deviation across 3 random seeds for key checkpoints (T1 and T5) and overall performance across the entire temporal span.

| Method | PROX T1 | PROX T5 | PROX Overall | MMLU T1 | MMLU T5 | MMLU Overall |
|---|---|---|---|---|---|---|
| Naive Fine-tuning | 0.264±0.001 | 0.270±0.024 | 0.277±0.016 | 0.479±0.001 | 0.319±0.058 | 0.363±0.085 |
| Full Retraining | 0.261±0.004 | 0.276±0.008 | 0.275±0.009 | 0.475±0.002 | 0.471±0.003 | 0.470±0.004 |
| EWC | 0.259±0.006 | 0.265±0.012 | 0.272±0.011 | 0.477±0.001 | 0.440±0.017 | 0.448±0.021 |
| Random Rehearsal | 0.257±0.009 | 0.274±0.021 | 0.270±0.018 | 0.481±0.002 | 0.438±0.012 | 0.448±0.023 |
| COMPASS-ECDA | 0.258±0.008 | 0.281±0.014 | 0.276±0.013 | 0.477±0.002 | 0.470±0.006 | 0.469±0.007 |

Table 12: Aggregate performance metrics for Phi-4-Mini-Instruct-3.8B.

| Method | PROX T1 | PROX T5 | PROX Overall | MMLU T1 | MMLU T5 | MMLU Overall |
|---|---|---|---|---|---|---|
| Naive Fine-tuning | 0.233±0.001 | 0.246±0.028 | 0.253±0.020 | 0.519±0.002 | 0.391±0.044 | 0.428±0.067 |
| Full Retraining | 0.234±0.002 | 0.251±0.011 | 0.246±0.010 | 0.517±0.002 | 0.508±0.005 | 0.511±0.006 |
| EWC | 0.235±0.003 | 0.241±0.014 | 0.245±0.010 | 0.520±0.002 | 0.484±0.013 | 0.494±0.018 |
| Random Rehearsal | 0.230±0.002 | 0.236±0.010 | 0.239±0.009 | 0.523±0.002 | 0.485±0.013 | 0.493±0.020 |
| COMPASS-ECDA | 0.232±0.001 | 0.252±0.012 | 0.252±0.015 | 0.519±0.001 | 0.506±0.008 | 0.509±0.007 |

Table 13: Aggregate performance metrics for LLaMA-3.1-Instruct-8B.

### D.4 Memory-Performance Analysis

We investigate the relationship between distributional anchor buffer size and performance retention using the Global MMLU to MMLU-ProX shift scenario. The analysis explores buffer sizes ranging from 0% representing pure regularization without rehearsal to 100% representing full rehearsal of all original training data.

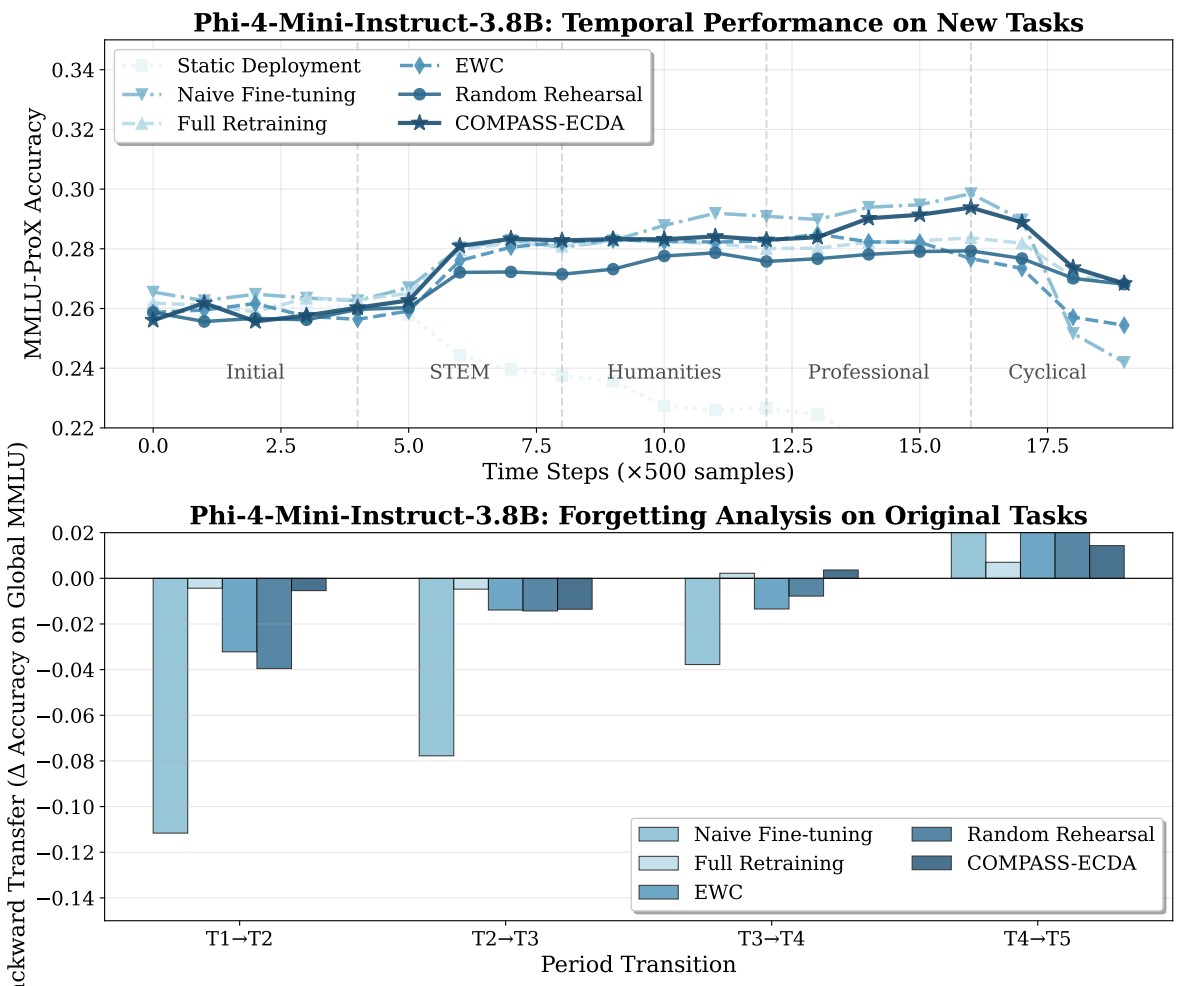

Figure 10: Temporal performance evolution for Phi-4-Mini-Instruct-3.8B across five distribution shifts. Despite limited model capacity, COMPASS-ECDA (dark blue) maintains stable adaptation with minimal forgetting. The pronounced forgetting in naive fine-tuning (light blue) underscores the importance of explicit retention mechanisms for smaller models.

| Method | PROX T1 | PROX T5 | PROX Overall | MMLU T1 | MMLU T5 | MMLU Overall |
|---|---|---|---|---|---|---|
| Naive Fine-tuning | 0.406±0.002 | 0.420±0.026 | 0.430±0.023 | 0.604±0.002 | 0.491±0.036 | 0.522±0.058 |
| Full Retraining | 0.408±0.001 | 0.427±0.010 | 0.425±0.013 | 0.600±0.001 | 0.591±0.005 | 0.590±0.007 |
| EWC | 0.402±0.001 | 0.418±0.016 | 0.422±0.016 | 0.603±0.001 | 0.576±0.011 | 0.580±0.016 |
| Random Rehearsal | 0.402±0.002 | 0.417±0.012 | 0.418±0.013 | 0.600±0.002 | 0.570±0.010 | 0.579±0.014 |
| COMPASS-ECDA | 0.405±0.002 | 0.437±0.015 | 0.431±0.019 | 0.600±0.001 | 0.591±0.006 | 0.592±0.007 |

Table 14: Aggregate performance metrics for Qwen2.5-7B-Instruct.

Table 15 reveals diminishing returns beyond 5% distribution anchor buffer size in terms of mitigating forgetting. Ultimately, the size of the buffer reflects how susceptible the network architecture is to catastrophic forgetting, which remains low due to the nature of PEFT.

## D.5 Hyperparameter Selection for COMPASS-ECDA

To determine optimal regularization strength EWC ($\lambda$) and loss weight for the DAR buffer ($\beta$), we performed a joint hyperparameter sweep over a predefined set of values for both hyperparameters. The search space for

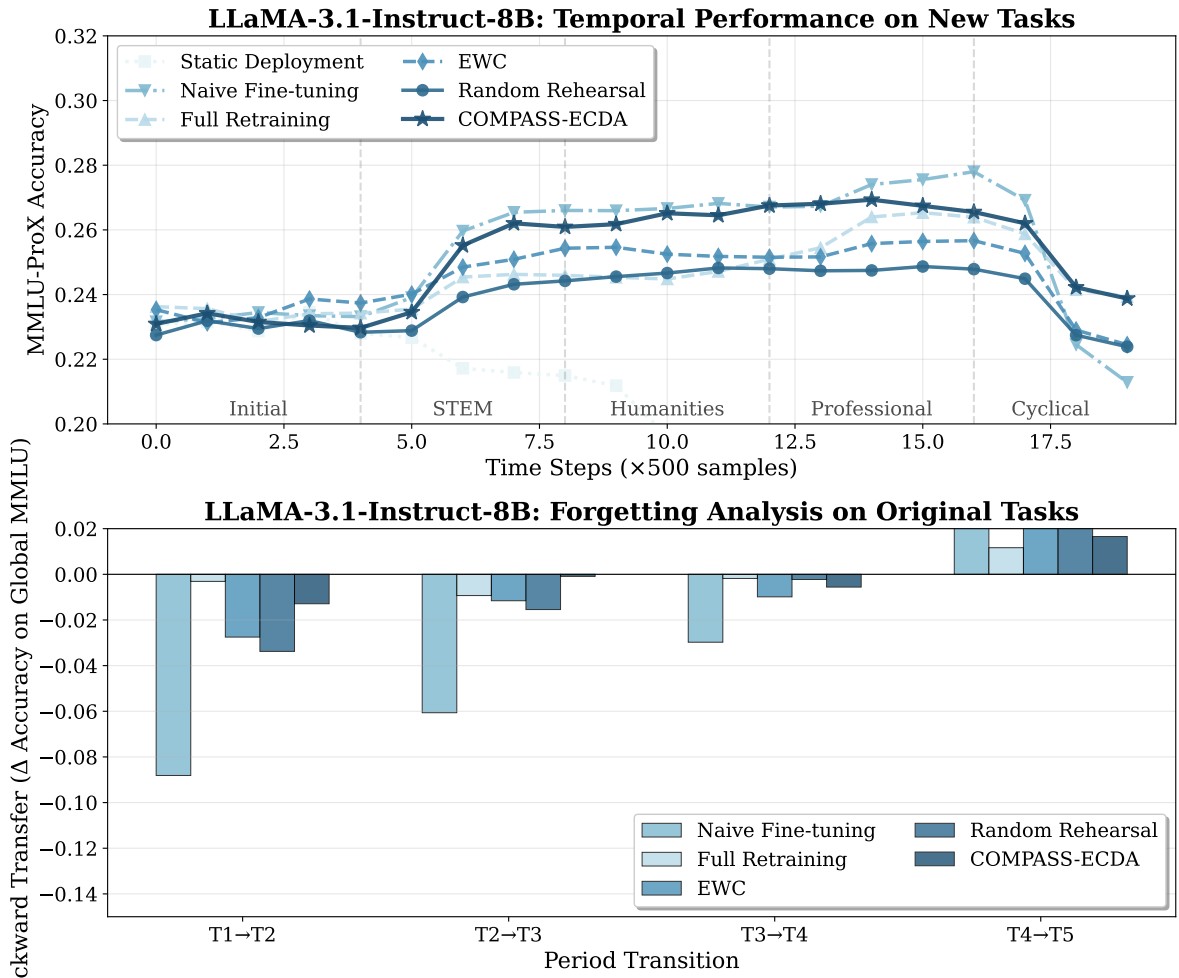

Figure 11: Temporal performance evolution for LLaMA-3.1-Instruct-8B.

| Buffer | Global MMLU | MMLU ProX |
|---|---|---|
| Size (%) | $\Delta$ | $\Delta$ |
| 0 | -10.9 | +6.9 |
| 1 | -5.0 | +6.3 |
| 5 | -1.6 | +5.2 |
| 10 | -1.0 | +2.4 |
| 20 | -0.7 | +1.6 |

Table 15: Memory-performance trade-offs for distributional anchor buffers on Qwen2.5-7B. $\Delta$ shows change from initial performance.

$\lambda$ was set to [0.1, 1, 2, 10, 100, 1000] to explore different orders of magnitude for the regularization penalty. The search space for $\beta$ was set to [0.001, 0.01, 0.1, 0.5] to evaluate different weights for the rehearsal loss.

For each pair of $(\lambda, \beta)$, we trained the adapter on the distribution shift task, adapting from the initial Global MMLU subjects to the new MMLU-ProX subjects. The optimal pair was selected based on its ability to achieve the best Pareto-optimal trade-off on a held-out validation set, i.e., maximized performance on the new MMLU-ProX subjects while ensuring performance degradation on the original Global MMLU subjects

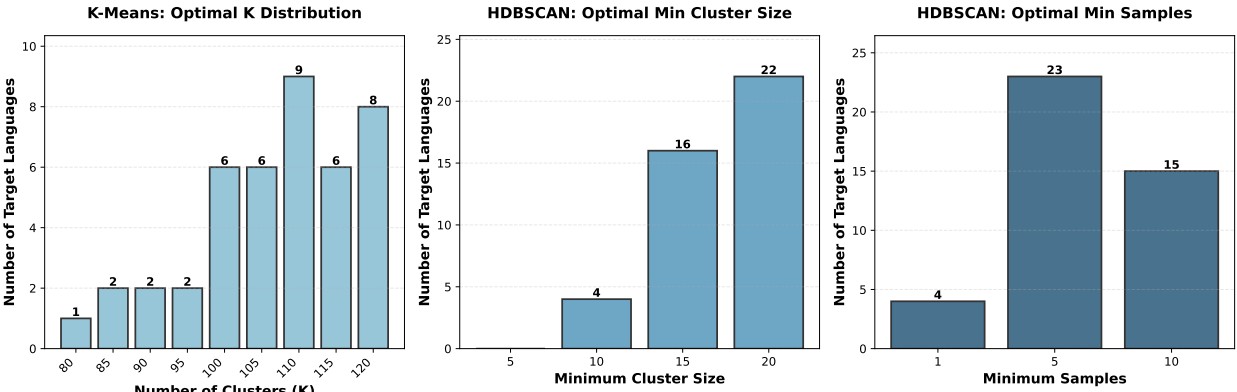

Figure 12: Distribution of optimal clustering parameters across 42 target languages. **Left:** K-means optimal cluster number ($K$) selected by maximizing silhouette score. **Middle:** HDBSCAN optimal minimum cluster size selected by maximizing DBCV score. **Right:** HDBSCAN optimal minimum samples parameter. The balanced distribution between 5 and 10 indicates moderate density requirements for effective semantic clustering.

remained minimal. This process identified $\lambda = 2$ and $\beta = 0.1$ as the most effective combination. These values were subsequently used for all COMPASS-ECDA experiments presented in the main paper.

## E Clustering Methods

COMPASS's performance depends on the quality of semantic clustering. We evaluated multiple clustering algorithms and conducted parameter sweeps to identify optimal configurations for each target language. For practical scenarios, we recommend conducting similar clustering hyperparameter investigations for each language, considering additional hyperparameter values beyond the upper bounds considered in this study. In this study, we use clustering metrics instead of downstream performance for efficiency, as fine-tuning each adapter is relatively costly and labeled test data may not be available prior to deployment.

**K-Means Clustering.** Our K-means implementation was configured to test a range of clusters from $K = 10$ to $K = 120$ in steps of 5. We used the k-means++ initialization method with 10 different seeds for each value of $K$ and evaluated cosine distance metrics. The final number of clusters per target language was selected by maximizing the silhouette score. Across all 42 languages, optimal $K$ ranged from 80 to 120, with no language achieving optimal performance below $K = 80$ (Figure 12, left panel). The distribution is heavily weighted toward higher values ($K \geq 100$) for 29 languages.

**HDBSCAN Clustering.** For HDBSCAN, we conducted a grid search over parameter combinations to find optimal configurations, evaluating cluster quality using the Density-Based Clustering Validation (DBCV) score, which accounts for varying cluster densities and shapes while penalizing noise points. We performed a grid search over minimum cluster sizes of $[5, 10, 15, 20]$ and minimum samples of $[1, 5, 10]$. The optimal minimum cluster size varied across languages (Figure 12, middle panel), though the majority of languages (22/42) achieved optimal clustering with `min_cluster_size`= 20. Only 4 languages required smaller values of 10, corresponding to low-resource languages. The minimum samples parameter, which determines core point density constraints, balanced between 5 and 10 (Figure 12, right panel), with preference for 5 samples. Only 4 languages (9.5%) achieved best performance with `min_samples`= 1. The preference for `min_samples`$\in$ $\{5, 10\}$ indicates that COMPASS requires moderate density thresholds to distinguish meaningful semantic clusters from noise, and larger minimums may net even better clustering.

**Hierarchical Agglomerative Clustering.** For hierarchical agglomerative clustering, we implemented a two-step approach using Ward's linkage method to build a hierarchical cluster tree and `fcluster` to cut the dendrogram at an optimized number of clusters. After clustering, we compute cluster centers as the mean of all embeddings in each cluster for allocating cluster assignments. We evaluated both Euclidean and cosine

distance metrics. For cosine similarity, we normalize embeddings before computing the distance matrix. The number of clusters was selected by maximizing the silhouette score over the range $K = 80$ to $K = 120$, following from initial results on K-means. Optimal $K$ values were distributed similarly to K-means, with majority of languages optimized at values of $K \geq 110$.

**Taylor-Butina Clustering.** Taylor-Butina clustering is a density-based algorithm that iteratively: (1) computes pairwise distances between all embeddings, (2) sorts points by their number of neighbors within a threshold, (3) assigns points as cluster centers if they haven't been claimed, and (4) assigns all neighbors of a center to its cluster. We enhanced this with an adaptive threshold selection mechanism that uses binary search to find the distance threshold within range $[0.70, 0.95]$ that yields an optimal number of clusters. This approach iteratively adjusts the threshold between specified minimum and maximum values until convergence, finding the threshold that produces the largest number of non-singleton clusters while ensuring that at least 95% of the data points were assigned to a cluster.

# F   Computational Efficiency Analysis

**Preprocessing Overhead.** COMPASS incurs a one-time preprocessing cost for embedding generation and clustering, which is amortized across all target languages. Using Jina-Embeddings-v3-570M (A100 GPU, batch size 128), we embed 204K examples from Aya dataet in 42.4 minutes (averaged over 3 embedding runs). HDBScan clustering on 204K 1024-dimensional embeddings required 2.2 hours on CPU. Amortized over 42 target languages, the one-time cost of preprocessing was 4.15 minutes per language.

**Per-Adapter Training Costs.** Per-adapter training time for COMPASS for Phi4-Mini-3.8B, Llama-3.1-8B, and Qwen2.5-7B was 44.9, 104.7, and 86.3 minutes per language, respectively. Comparatively, per-adapter training time was 17.9, 49.7, and 32.8 minutes per language, respectively, when only using the target language training data and not incorporating the auxiliary training data. For full finetuning, we observed increases in training time to 61.3, 133.2, and 110.6 minutes per language, respectively, for Phi4-Mini-3.8B, Llama-3.1-8B, and Qwen2.5-7B. While rigorous comparison to full finetuning requires careful experimental controls (batch size, rank) and depends on hardware configuration and usage of optimized implementations, existing literature suggests LoRA and DoRA typically achieve up to 40% reductions in peak memory compared to full finetuning, though there are certain settings where throughput can decrease by up to 15% (Biderman et al., 2024; Liu et al., 2024).

**Inference Costs.** COMPASS introduces minimal inference overhead through adapter loading and language detection. Given sentence-length inputs, GlotLID-v3 latency averaged 6 milliseconds with the model already loaded in memory. With all adapters pre-loaded, switching is less than 1 millisecond. In aggregate, these add negligible overhead to typical LLM inference latency. When loaded dynamically, the DoRA adapters introduced no more than 3% inference overhead compared to the base model across 3 runs on Global-MMLU. Per-adapter memory overhead for Qwen2.5-7B-Instruct is approximately 40 MB (1.68 GB total for 42 adapters), representing roughly 10% additional storage relative to base model size.

**Scaling Considerations.** Adding support for a new language requires embedding the new language's dev set, computing cluster weights, sampling auxiliary data, and training one adapter. Assuming pre-computed cluster assignments from the initial clustering run on the initial language set, supporting a new language with a dev set of 3,000 examples required 496 seconds for the first three steps: embedding examples (248 sec), computing cluster weights (174 sec), and sampling auxiliary data (74 sec). The primary bottleneck remains adapter training (0.75 to 2.25 hours depending on the base model and data budget). While COMPASS-ECDA update cycles add distribution shift detection and usage of incremental clustering, the scaling bottleneck remains adapter retraining. Adding support for a new language entails linear scaling for storage ( 40 MB per adapter) and constant inference overhead, as language detection and adapter loading times remain unaffected by the total number of supported languages.

# G   Statistical Testing

To assess statistical significance of performance differences between COMPASS and baseline methods, we employ non-parametric statistical tests that leverage cross-language variance from our single-run experiments across 42 languages (Global-MMLU) and 29 languages (MMLU-ProX).

**Permutation tests.** For each pairwise comparison (e.g., COMPASS vs. Target), we perform approximate randomization tests by randomly shuffling the method labels across languages and recalculating the mean performance difference. We repeat this process 10,000 times to construct an empirical null distribution under the hypothesis of no systematic difference between methods. The p-value is computed as the proportion of permutations yielding a difference as large or larger than the observed difference. This approach tests whether COMPASS's improvements are consistent across the language distribution or could arise from random variation.

**Effect sizes.** We report Cohen's d effect sizes to quantify the practical magnitude of improvements beyond statistical significance. Effect sizes are calculated using the pooled standard deviation across languages and interpreted using standard thresholds: small ($d \approx 0.2$), medium ($d \approx 0.5$), and large ($d \geq 0.8$). This addresses the concern that with 42 languages, even small differences may achieve statistical significance despite limited practical importance.

**Sign tests.** To evaluate whether improvements are distributed across languages rather than driven by outliers, we use binomial sign tests that count how many languages improved with COMPASS versus how many regressed. We test against the null hypothesis of equal probability (50%).

