# OpenReview forum: "COMPASS: COntinual Multilingual PEFT with Adaptive Semantic Sampling"
_TMLR — Accepted by TMLR_

### Review · Reviewer_Jy1S · 2025-09-23

**Summary Of Contributions:**

This paper introduces COMPASS, a data-centric framework for adapting large language models (LLMs) to multiple languages. The core idea is selectively integrating data from auxiliary multilingual training set, and use parameter-efficient fine-tuning (PEFT) with language-specific adapters. The selection is guided by a distribution-aware sampling strategy that identifies semantic gaps between the target language’s training data and its expected usage distribution. The method is extended to a continual learning setting (COMPASS-ECDA) to handle real-world distribution shifts over time. The proposed method shows empirical advantage over baselines including monolingual and non-distribution-aware strategies.

Strength:
1. Adapting LLMs to LRLs is valuable topic in real world, and utilizing additional multilingual data is a reasonable solution.
2. The paper provides a complete pipeline for adapting LLM to LRLs by utilizing additional multilingual data. The proposed method is overall reasonable.
3. Extensive empirical studies are provided, showing the proposed method is effective.

Weakness:
1. COMPASS first obtains instance embedding by pre-trained encoder and make clustering, and the entire method is based on this process. It is likely to be sensitive to the encoder and clustering process. For encoder, though sec 4.2 mentioned the choice, no empirical results with other encoders are reported. The sensitiveness could be vital when deploying the method for new dataset/models, without testing data available to choose encoder. Similarly, sensitiveness about the unsupervised clustering algorithm and cluster number K, which have not been mentioned in paper, is also likely to be critical.
2. COMPASS makes improvement with the cost of efficiency. Not only the mentioned introducing additional training data (budget Q), but also the additional embedding-clustering-weighting process on the complete auxiliary set (might be large). No empirical results about the consumption is provided.
3. Lack of proper evaluation of the effect of continual adaptation strategy. In sec 5.5, the synthetic setting only reflects domain-adaptation ability rather than continual learning, i.e., it requires long-time-range distribution shift rather than one-step shift.

**Additional Comments:**

NA

**Audience:**

Yes

**Audience Explanation:**

The topic of adapting LLM to low-resource languages is important in real-world.

**Claims And Evidence:**

Yes

**Claims Explanation:**

Most claims made in the submission are supported by accurate, convincing and clear evidence. But modifications are still necessary as Required Changes.

**Requested Changes:**

Corresponding to the weakness.
1. Provide the clustering algorithm; Provide empirical results of sensitiveness of the choice of encoder, and choice of cluster number K.
2. Provide and properly discuss the training efficiency of COMPASS and baselines.
3. Evaluate the effectiveness of the proposed continually adaptation strategy under proper continual learning setting/senario.

---

> ### Author Response · Authors · 2025-10-31
> **Response to Reviewer Jy1S (Part 1)**
>
> We thank the reviewer for the helpful feedback, especially regarding the detailed commentary on extending our continual learning evaluations to prove more robust adaptation. We believe the following changes have enhanced the overall quality of the manuscript and its utility to the TMLR community.
>
> **Change 1: Encoder and clustering sensitivity**
>
> *“Provide the clustering algorithm; Provide empirical results of sensitiveness of the choice of encoder, and choice of cluster number K.”*
>
> We thank the reviewer for raising these important concerns about methodological sensitivity. To increase prominence of our ablations on the choice of encoder and clustering algorithm, we add a forward reference to section 4.2:
>
>    >We evaluate sensitivity to encoder choice and clustering parameters in Section 5.7.
>
> Section 5.7 (pages 26-28) empirically supports that COMPASS is sensitive to both the choice of encoder and clustering method. Regarding encoder sensitivity, section 5.7 presents ablations across 4 embedding models showing performance ranges from 41.1% to 52.4% on Global MMLU, with discussion of failure modes for under-performing encoders. Regarding clustering algorithm choice, section 5.7 compares 4 clustering methods with performance differences up to 11.8 points, demonstrating sensitivity.
>
> To address the reviewer's concern about cluster number K, we have expanded “Appendix E: Clustering Methods” on pages 51-52 to include greater detail regarding clustering methods and choice of cluster number K and related parameters that affect clustering.
>
> **Change 2: Efficiency measurements**
>
> *“COMPASS makes improvement with the cost of efficiency. Not only the mentioned introducing additional training data (budget Q), but also the additional embedding-clustering-weighting process on the complete auxiliary set (might be large). No empirical results about the consumption is provided. Provide and properly discuss the training efficiency of COMPASS and baselines.”*
>
> We agree that this is essential for practical deployment assessment. This concern overlaps with Reviewer 6ZzK 's Change 1. As detailed in that response, we have added a 'Computational Efficiency Analysis' appendix section (Appendix F) covering preprocessing costs, baseline comparisons, memory footprint, inference overhead, and scaling analysis.. Please see our response to Reviewer 6ZzK, Change 1 for further details.
>
> **Change 3: Continual learning evaluation**
>
> *“Lack of proper evaluation of the effect of continual adaptation strategy. In sec 5.5, the synthetic setting only reflects domain-adaptation ability rather than continual learning, i.e., it requires long-time-range distribution shift rather than one-step shift. Evaluate the effectiveness of the proposed continually adaptation strategy under proper continual learning setting.”*
>
> We thank the reviewer for this insightful critique and the chance to better demonstrate our method’s value. We have addressed this concern by implementing a multi-step evaluation that demonstrates long-time-range adaptation capabilities.
> We add a new subsection, “Multi-Step Temporal Evaluation” to section 5.5 (page 23-25). In this additional subsection, we now evaluate COMPASS-ECDA across five sequential distribution shifts (T1→T2→T3→T4→T5), representing realistic temporal evolution in deployed systems:
>
> - T1: Initial deployment (27 base subjects)
> - T2: STEM expansion (+10 advanced technical domains)
> - T3: Humanities diversification (+10 ethical/philosophical domains)
> - T4: Professional integration (+10 enterprise domains)
> - T5: Cyclical return (testing retention after intervening shifts)
>
> Each period consists of 2K samples with performance (both adaptation and retention) measured at 500-sample increments. We find that COMPASS-ECDA demonstrates the best balance of adaptation, retention, and computational efficiency, successfully readapting to T1 distribution at T5 after intervening shifts (T2-T4). We evaluate all three models (Phi-4-Mini, LLaMA-3.1, Qwen2.5), showing COMPASS-ECDA's effectiveness across different capacities, with more detailed results reported in Appendix D (pages 46-51).

---

### Review · Reviewer_AfXw · 2025-10-15

**Summary Of Contributions:**

This paper introduces COMPASS, a distribution-aware sampling method for selecting multilingual training data to fine-tune large language models (LLMs) using DoRA, with the goal of improving performance on targeted non-English languages. The core motivation stems from the well-documented problem of negative interference in multilingual training: when models are exposed to heterogeneous language data without regard to task or distributional alignment, dominant languages can overshadow under-resourced ones, leading to suboptimal representations and degraded performance.

The proposed approach is conceptually straightforward and intuitively appealing. It estimates the cluster-based task distribution of a target language using its evaluation (or validation) set, then selectively augments the training data with complementary examples from a multilingual corpus that fill underrepresented clusters. By aligning the training distribution more closely with the downstream task distribution, COMPASS aims to reduce noise and conflicting gradients during fine-tuning. Empirical results on Global MMLU and MMLU-ProX demonstrate consistent gains over strong baselines, including other data sampling strategies. The use of LoRA for parameter-efficient adaptation is well justified, as it enables scalable multilingual tuning while preserving pretrained knowledge.

However, the paper’s adoption of DoRA—a recent variant of LoRA that decomposes weight updates into magnitude and direction components—lacks sufficient justification. The authors do not provide ablation studies comparing DoRA against standard LoRA in the multilingual setting, nor do they quantify the trade-offs in terms of training/inference overhead. Given that DoRA introduces additional computational latency without demonstrated multilingual-specific benefits, this design choice appears under-motivated.

**Audience:**

Yes

**Audience Explanation:**

- The development of multilingual LLMs is an important issue.
- The application of LoRA is an interesting topic for the PEFT community.

**Broader Impact Concerns:**

No concerns.

**Claims And Evidence:**

Yes

**Claims Explanation:**

The authors have conducted comprehensive experiments to support their claimed contributions listed in the introduction.

However, it remains unclear whether the observed performance gain stems from training on more data within clusters that have higher $w_k$. Moreover, the training hyperparameters employed for FFT are not specified and should differ significantly from those used by DoRA to achieve optimal performance.

**Requested Changes:**

More critically, the introduction presents a significant mismatch between the problems it foregrounds and the technical solution offered. The opening paragraphs devote considerable space to a compelling but unrelated set of challenges in multilingual LLMs, including:

- Cross-lingual security vulnerabilities (e.g., jailbreaking via translation, with success rates jumping from ≤1% to 79% in LRLs; Yong et al., 2024),

- Tokenization inefficiencies for non-Latin scripts due to subword fragmentation (Cui et al., 2024; Ji et al., 2023),
Privacy risks such as PII leakage and membership inference in under-resourced languages (Lukas et al., 2023; Li et al., 2024),

- Cultural and semantic mismatches, including translational ambiguity and locale-specific behavioral norms.

These are serious, real-world issues that indeed exacerbate inequities in global AI deployment. Yet COMPASS does not address any of them. The method is purely a data selection strategy based on task-distribution alignment; it does not incorporate mechanisms for improving tokenization, enhancing safety alignment across languages, mitigating privacy leakage, or modeling typological or cultural differences. Even the core issue of linguistic interference—often rooted in syntactic divergence, morphological complexity, or script differences—is treated in the introduction as a structural problem, while COMPASS tackles it only indirectly (and arguably superficially) through distributional coverage.

This creates a misleading narrative: the introduction implies that the paper contributes to solving deep, systemic inequities in multilingual NLP, when in fact it offers a narrow—but useful—optimization for data-efficient fine-tuning. If the authors’ intent is to position distributional mismatch as one contributor to poor LRL performance (among many), that is reasonable. But the current framing overstates the scope of the problem addressed and risks conflating data distribution misalignment with inherent linguistic or systemic disparities.

To improve clarity and scholarly rigor, the authors should:

Sharply refocus the introduction to center on distributional mismatch as the primary challenge, relegating broader issues (security, tokenization, privacy) to a brief contextual paragraph—or removing them entirely if they are not addressed by the method.
Explicitly clarify the scope of COMPASS: Does better data coverage indirectly alleviate some symptoms of linguistic interference (e.g., by reducing noise from irrelevant tasks), or is the method agnostic to true linguistic structure? The current text leaves this ambiguous.
Avoid implying that distributional alignment resolves deep cross-lingual challenges like cultural nuance or script-based tokenization inefficiencies, unless empirical evidence supports such a claim.

---

> ### Author Response · Authors · 2025-10-31
> **Response to Reviewer AfXw (Part 1)**
>
> We are delighted that the reviewer found our method intuitively appealing and experimentally convincing. In the following, we respond to the comments in detail.
>
> **Change 1: Introduction mismatch**
>
> *“To improve clarity and scholarly rigor, the authors should sharply refocus the introduction to center on distributional mismatch as the primary challenge, relegating broader issues (security, tokenization, privacy) to a brief contextual paragraph. Explicitly clarify the scope of COMPASS: Does better data coverage indirectly alleviate some symptoms of linguistic interference (e.g., by reducing noise from irrelevant tasks), or is the method agnostic to true linguistic structure? The current text leaves this ambiguous. Avoid implying that distributional alignment resolves deep cross-lingual challenges like cultural nuance or script-based tokenization inefficiencies, unless empirical evidence supports such a claim.”*
>
> We thank the reviewer for this important feedback. We agree that our original framing may have overstated the novelty of addressing multilingual challenges, and we have revised the manuscript to better position our specific contributions. We have condensed the discussion of security, tokenization, and privacy concerns into a single contextual paragraph (now paragraph 2 in the Introduction), removing the extended motivation that could be interpreted as claiming these as novel contributions.
>
> We have also clarified in two pre-existing, subsequent paragraphs (now paragraphs 3-4 in the Introduction) that COMPASS specifically targets distributional mismatch in cross-lingual transfer, rather than attempting to solve multilingual NLP broadly. Specifically, we point out the following revised text that now explicitly states:
>
>    >COMPASS does not directly tackle tokenization inefficiencies, safety vulnerabilities, or inherent linguistic structure differences – these remain important open problems. Instead, COMPASS operates on the principle that better coverage of the target distribution, achieved through semantically-guided auxiliary data selection, can improve model performance by reducing task-irrelevant noise and ensuring exposure to under-represented usage patterns. This improved distributional alignment may indirectly reduce some symptoms of linguistic interference (e.g., by filtering irrelevant cross-lingual examples), but the method is fundamentally task-distribution-driven rather than linguistically-informed.
>
> To acknowledge the broader landscape of multilingual NLP challenges, we have also added a new paragraph to the Future Work (page 30, paragraph 2 under Future Work) section, discussing how future research could extend our distributional analysis framework to investigate systemic issues. This positions our work as one methodological contribution within a larger research agenda. We believe these revisions appropriately scope our claims while maintaining the paper's core contributions.
>
> **Change 2: DoRA justification**
>
> *“The authors do not provide ablation studies comparing DoRA against standard LoRA in the multilingual setting, nor do they quantify the trade-offs in terms of training/inference overhead. Given that DoRA introduces additional computational latency without demonstrated multilingual-specific benefits, this design choice appears under-motivated.”*
>
> We acknowledge this gap and have added further comparison of DoRA and LoRA across a range of rank and learning rate values to motivate our choice of DoRA (pages 27-28), which we found to be more stable across varied hyperparameter values (i.e., having a flatter peak). The analysis, which includes a new figure, is in the ablations subsection of the manuscript and also acknowledges the advantages of LoRA in terms of competitive performance with DoRA when comparing optimal hyperparameters and when accounting for training overhead of DoRA.

---

> ### Author Response · Authors · 2025-10-31
> **Response to Reviewer AfXw (Part 2)**
>
> **Change 3: FFT hyperparameters**
>
> *“The training hyperparameters employed for FFT are not specified and should differ significantly from those used by DoRA to achieve optimal performance.”*
>
> We thank the reviewer for catching this omission as FFT does require different hyperparameters. We have added additional specifications to its mention in the experimental set-up section (page 15). The updated description now reads:
>
>    >COMPASS full fine-tuning (COMPASS-FFT): Fine-tune the entire pretrained model on the COMPASS-supplied datasets, leveraging the model’s full parameter space to improve at the task. While this approach allows for maximum adaptation to the target language, it entails substantial memory overhead to store a complete set of model parameters for each target language and increases overfitting risk. Consistent with prior comparisons of learning rate sensitivity of LoRA-related methods to FFT (Biderman et al., 2024), we identified optimal FFT performance with reduced learning rates of 5e-5 for Phi-4-mini, 1e-5 for LLaMA-8B, and 2e-5 for Qwen2.5-7B, along with a reduced warm-up ratio of 0.05 and a batch size of 16 across all models.

---

### Review · Reviewer_6ZzK · 2025-10-18

**Summary Of Contributions:**

This paper introduces COMPASS, a framework for multilingual adaptation of large language models via parameter-efficient fine-tuning (PEFT) coupled with adaptive semantic sampling. The key innovation is a distribution-aware data selection strategy that identifies semantically underrepresented regions in the target language and selectively samples auxiliary multilingual data to fill these gaps. The paper further extends this to a continual learning variant (COMPASS-ECDA), which monitors distributional shifts in production data via Jensen–Shannon divergence and updates adapters.
Empirical results across Phi-4-Mini, LLaMA-3.1-8B, and Qwen2.5-7B on Global-MMLU, MMLU-ProX, and OneRuler show consistent improvements, highlighting strong cross-lingual transfer and efficient low-resource language adaptation.

**Audience:**

Yes

**Audience Explanation:**

The work directly contributes to multilingual LLM adaptation, a central and timely topic. It bridges data selection, continual learning, and PEFT, offering a practical, data-centric approach to improving multilingual fairness and sustainability, issues highly relevant to both academic and applied researchers. Readers in multilingual NLP, instruction tuning, and model maintenance would find this work significant and broadly applicable.

**Broader Impact Concerns:**

Others things that could be explicitly considered:

1. Bias propagation from multilingual embeddings and clustering errors that may favor high-resource languages.

2. Ethical implications of data re-sampling if low-resource language data contain sensitive cultural or identity-linked content.

3. Environmental costs of continual adaptation(repeated fine-tuning cycles), especially for large-scale deployment.

**Claims And Evidence:**

Yes

**Claims Explanation:**

The methodology is well-motivated and empirically validated across multiple models and benchmarks. The experimental setup is thorough, covering ablation studies, auxiliary data budgets, and distribution shift simulations, and results are consistent with the claims of improved multilingual adaptation and sustainability.
Results, however, could benefit from stronger statistical analysis (e.g., significance testing) and more transparency regarding computational overheads and scalability of the continual learning component (e.g., incremental clustering efficiency). Overall, evidence is convincing, with well-chosen baselines and detailed comparisons.

**Requested Changes:**

Clarify computational cost: quantify training/inference overhead per language adapter and scaling with the number of languages. Include concrete measurements of training time, memory footprint, and wall-clock time for data selection. Compare against full fine-tuning and other baselines.

Report variance metrics: include error bars or multiple random seeds to validate robustness.

Failure analysis: The paper shows that languages like Greek, Japanese, Korean, and Vietnamese have marginal gains . Provide deeper analysis of why COMPASS underperforms for these isolate/unique script languages. Is it embedding quality, lack of linguistic relatives in the auxiliary pool, or something else?

Improve readability: the paper is quite dense; summarizing the algorithm steps and key equations in a compact table or pseudocode would enhance clarity. Figure 4 is very difficult to read

---

> ### Author Response · Authors · 2025-10-31
> **Response to Reviewer 6ZzK (Part 1)**
>
> We thank the reviewer for several important comments. We believe these revisions have substantially strengthened the manuscript.
>
> **Change 1: Clarify computational cost**
>
> *“Clarify computational cost: quantify training/inference overhead per language adapter and scaling with the number of languages. Include concrete measurements of training time, memory footprint, and wall-clock time for data selection. Compare against full fine-tuning and other baselines.”*
>
> We have added a "Computational Efficiency Analysis" section to Appendix F (page 52) that provides detailed quantitative analysis across all requested dimensions. Our summary of changes made:
>
> 1. Preprocessing costs: We report embedding time (42.4 minutes for 204K examples), clustering time (2.2 hours), and amortized per-language preprocessing cost (4.15 minutes).
>
> 2.	Baseline comparisons: While rigorous controlled comparison requires extensive experimentation beyond scope, we provide training costs associated with COMPASS per-adapter finetuning, target only finetuning without the added auxiliary budget, and full finetuning. We clarify that COMPASS’ data budget increases per-adapter training time by approximately 75%-90% relative to target only adapter finetuning.
>
> 3.	Memory footprint: We report per-adapter memory overhead (40 MB), aggregate overhead for 42 adapters (1.68 GB), and percentage relative to base model size (10%).
>
> 4.	Inference overhead: We quantify language detection latency (6ms), adapter switching time (<1ms), dynamic loading overhead (3%), and total added latency (negligible relative to LLM inference).
>
> 5.	Scaling analysis: We break down the cost of adding a new language into constituent operations: embedding (248 sec), cluster weight computation (174 sec), sampling (74 sec), and adapter training (1.3-2.75 hours). We clarify that storage scales linearly (40 MB per language) while inference overhead remains constant.
>
> **Change 2: Report variance metrics**
>
> *"Results, however, could benefit from stronger statistical analysis (e.g., significance testing)...Report variance metrics: include error bars or multiple random seeds to validate robustness."*
>
> We thank the reviewer for this important suggestion and have strengthened the statistical rigor of our evaluation.
>
> 1.	We now report statistical significance for all pairwise comparisons between COMPASS and baseline methods using non-parametric permutation tests. Results are marked with significance indicators in Tables 1-2. These tests leverage the distribution of performance across 42 languages (Global-MMLU) and 29 languages (MMLU-ProX).
>
> 2.	We also report Cohen's d effect sizes for major comparisons, quantifying the practical magnitude of improvements.
>
> 3.	We report binomial sign tests showing that COMPASS improves performance on the majority of languages, demonstrating that improvements are broadly distributed rather than concentrated in a few outliers.
>
> 4.	The per-language performance tables in the Appendix now explicitly report standard deviations across languages, providing transparent variance metrics.
>
> 5.	We have incorporated additional continual learning experiments (Section 5.5) with 3 random seeds to provide additional robustness validation for this component.
>
> **Change 3: Failure analysis**
>
> *“The paper shows that languages like Greek, Japanese, Korean, and Vietnamese have marginal gains. Provide deeper analysis of why COMPASS underperforms for these isolate/unique script languages.”*
>
> We thank the reviewer for the opportunity to expand on this observation. We have expanded the auxiliary budget results section (section 5.2, page 17-18) to detail our rationale for why lack of linguistic relatives in the auxiliary pool are a primary contributor to marginal performance gains in unique script languages. We note that embedding quality is a secondary factor, though this is discussed in further detail within the Discussion section. Our expanded auxiliary budget section identifies two primary factors contributing to marginal gains in these languages: (1) lack of linguistic relatives in the auxiliary pool, and (2) auxiliary budget saturation at lower budget thresholds.
>
> **Change 4: Readability**
>
> *“The paper is quite dense; summarizing the algorithm steps and key equations in a compact table or pseudocode would enhance clarity.”*
>
> We agree that algorithmic clarity is crucial. We have added formal pseudocode (page 10) to clarify the precise specification of COMPASS in the methodology section.

---

> ### Author Response · Authors · 2025-10-31
> **Response to Reviewer 6ZzK (Part 2)**
>
> **Change 5: Additional impact concerns**
>
> *“Other things that could be explicitly considered: Bias propagation from multilingual embeddings and clustering errors that may favor high-resource languages. Ethical implications of data re-sampling if low-resource language data contain sensitive cultural or identity-linked content. Environmental costs of continual adaptation(repeated fine-tuning cycles), especially for large-scale deployment.”*
>
> We have substantially expanded the Broader Impact Statement and Discussion sections, strengthening the paper’s consideration of real-world deployment implications. To the broader impact statement (page 31), we have added paragraphs to promote awareness of bias propagation an amplification, ethical considerations in data resampling, and environmental costs associated with our work:
>
>    >Beyond encoder biases, COMPASS's data selection strategy could amplify biases present in the auxiliary data pool. High-performing semantic clusters are preferentially sampled, which may overrepresent majority viewpoints or Western-centric content even within non-Western languages, potentially marginalizing alternative cultural or institutional frameworks in affected communities.
>
>    >Our distribution-aware sampling prioritizes auxiliary data matching target usage patterns, which may inadvertently sample culturally sensitive content (e.g., indigenous knowledge, minority language expressions, religious texts) without proper cultural context or community consent. While COMPASS does not modify such content, the selection and repurposing of low-resource language data raises ethical questions about data sovereignty and appropriate use that warrant engagement with linguistic communities, especially those that have less agency in dictating data usage concerns.
>
>    >Lastly, the continual adaptation paradigm in COMPASS-ECDA requires periodic retraining cycles, which has associated environmental costs. While adapter-based approaches are more efficient than repeatedly training full models, production deployments should carefully consider the environmental impact of update frequency and explore strategies such as batching updates across multiple languages to reduce computational overhead.
>
> To the Discussion section, we have added a paragraph proposing bias mitigation strategies:
>
>    >Addressing bias propagation requires integrating fairness constraints into the sampling objective. Future work should explore debiasing objectives that penalize selection of stereotypical associations and cluster-level auditing to identify whether certain topics (e.g., gender, religion, socioeconomic status) are disproportionately represented or omitted. Privacy-preserving Cluster interpretability methods, such as Clio Tamkin et al. (2024), may provide human oversight of the selection process in real-world applications.

---

### Review · Reviewer_DbVY · 2025-10-25

**Summary Of Contributions:**

The core contributions are:

1.  **Adaptive Semantic Sampling:** This is the main technical contribution. The framework uses a multilingual embedding model to map all training data ($D_t$), auxiliary data ($D_{aux}$), and a proxy for the target usage distribution ($E_t$) into a shared semantic space. It then clusters this space to identify "semantic gaps"—topics or concepts present in the target usage data ($E_t$) but under-represented in the target training data ($D_t$). COMPASS then samples data from the auxiliary pool ($D_{aux}$) to fill these specific gaps, maximizing positive cross-lingual transfer and minimizing interference.
2.  **Instance-Level Curriculum:** Within a high-priority cluster, the sampling strategy follows a conservative "easy-to-hard" curriculum, initially prioritizing prototypical examples near the cluster centroid before cautiously sampling more ambiguous examples near the decision boundary.
3.  **Continual Learning Framework (COMPASS-ECDA):** The paper extends COMPASS to a continual learning setting to handle real-world distribution shifts. This extension features:
    * A **distribution mismatch trigger** that uses Jensen-Shannon (JS) divergence to detect when the incoming data stream has shifted significantly from the adapter's training distribution.
    * A hybrid update strategy, **Elastic Consolidation and Distributional Anchoring (ECDA)**, which balances plasticity and stability. It combines a standard task loss with Elastic Weight Consolidation (EWC) and a "Distributional Anchor Replay" (DAR) loss, which rehearses a small buffer of prototypical examples from previously learned clusters.

**Additional Comments:**

No.

**Audience:**

Yes

**Audience Explanation:**

1.  **Core ML:** The paper tackles two fundamental challenges in machine learning: **negative interference** in multi-task/multilingual learning and **catastrophic forgetting** in continual learning. The proposed data-centric solutions are relevant beyond just NLP.
2.  **LLMs:** The work is extremely timely. As LLMs become globally deployed, "multilinguality" is a paramount concern. The paper provides a practical, parameter-efficient path to improving performance for low-resource languages (LRLs), a notoriously difficult problem.

**Broader Impact Concerns:**

No.

**Claims And Evidence:**

Yes

**Claims Explanation:**

* **Claim 1: COMPASS provides superior performance by maximizing positive transfer and minimizing negative interference.**
    * **Evidence:** This is best supported by **Table 1**. The evidence is convincing because COMPASS (using PEFT) consistently outperforms *all* key baselines:
        * It beats **`Pretrained (ZS)`**, showing fine-tuning is necessary.
        * It beats **`Target` (monolingual tuning)**, proving that intelligent cross-lingual transfer is superior to using only target-language data.
        * It beats **`All` (naive multilingual tuning)**, decisively demonstrating that it avoids the negative interference that plagues indiscriminate data mixing. The `All` baseline often performs terribly, validating the paper's core motivation.
        * It beats **`Random` (sampling)**, proving that the performance gain is not just from adding more data, but from *strategic selection*.
        * It beats **`LangSim` and `LangRank`**, suggesting that semantic, distribution-aware sampling is a more effective strategy than relying on static, pre-defined linguistic similarity.
        * It achieves performance comparable to **`COMPASS-FFT`** (full fine-tuning) while being vastly more parameter-efficient.

* **Claim 2: The continual learning framework (COMPASS-ECDA) effectively adapts to new distributions while mitigating catastrophic forgetting.**
    * **Evidence:** This is clearly demonstrated in **Figure 6**. The plot of "learning vs. forgetting" shows that COMPASS-ECDA is **Pareto-optimal**. It achieves high accuracy on the new task (y-axis, MMLU-ProX) while simultaneously retaining high performance on the original task (x-axis, Global-MMLU). It clearly provides a better stability-plasticity trade-off than naive fine-tuning (which forgets) or EWC-only (which fails to learn).

* **Claim 3: The components of the sampling strategy are all necessary.**
    * **Evidence:** The ablation studies in **Table 2** are thorough and provide clear answers.
        * **Weighting:** Removing *cluster-level weights* (i.e., not knowing *which topics* to sample from) causes a major performance drop (5.1% on Global MMLU). Removing *sample-level weights* also hurts (3.7%). This cleanly proves that "what to sample" (cluster) is most important, and "which example to sample" (instance) provides further benefit.
        * **Encoder Quality:** The ablations show that the framework is *highly* dependent on a quality embedding model. Using worse encoders (Distiluse, Paraphrase Mpnet) leads to performance that is *worse than random sampling*. This is a crucial and honest finding.

* **Claim 4: The benefits of COMPASS generalize to unseen tasks and formats.**
    * **Evidence:** This is supported by **Figure 5** (long-context) and **Figure 7** (XNLI, XQUAD, MGSM). The fact that fine-tuning on short-context instruction data (Aya) can improve performance on long-context "needle-in-a-haystack" tasks (OneRuler) for low-resource languages is a strong, non-obvious result. The broad gains on XNLI, XQUAD, and MGSM confirm the adaptation is improving the model's core multilingual understanding, not just overfitting to the MMLU task format.

**Requested Changes:**

1.  **Clarify the "Cold-Start" Problem for $E_t$.** The method's initial adaptation phase hinges on having a representative proxy for the "live usage distribution," $E_t$. The paper uses the dev sets of the benchmarks for this. In a real-world "cold start" scenario for a new language, *such a set would not exist*.
    * The paper acknowledges this as a limitation and a motivation for the ECDA extension, but it should be addressed more directly in the methodology (Section 3.1).
    * Please add a brief discussion on practical strategies for bootstrapping $E_t$. For example: Can a human provide just a few dozen "gold" example prompts? What happens if you simply set $E_t = D_t$ initially and rely on COMPASS-ECDA to correct it over time?

2.  **Address the "Promiscuous Language" Finding.** In Section 5.4, the authors note that some LRLs (e.g., Malagasy, Tamil) are "promiscuous," meaning they are sampled by many unrelated languages. They hypothesize this is an *artifact* of the Jina3 encoder having lower-quality embeddings for those languages.
    * This is an important weakness. It implies the sampling is not purely semantic, but is also influenced by encoder flaws. This should be more explicitly framed as a limitation in Section 6.1, as it underscores the encoder's role as a potential bottleneck.
    * This finding also strengthens the case for the next suggestion.
. Please consider using a more standard diverging colormap (e.g., red-white-green) and increasing the font size within the heatmap cells.

---

> ### Author Response · Authors · 2025-10-31
>
> We thank the reviewer for their thoughtful comments, which helped enhance the quality of the paper in terms of practical usage (cold-start problem) and emphasis on additional findings relevant to the multilingual NLP community. We have individually addressed the reviewers’ requested changes.
>
> **Change 1: “Cold-Start” problem clarification**
>
> This is an important practical concern, and we have added a new section (3.2.2 on page 8) on practical bootstrapping strategies for those developing real-world applications. The new text reads:
>
>    >In our experiments, we use held-out evaluation sets (dev sets of Global-MMLU and MMLU-ProX) as proxies for the live usage distribution $E_t$. This approximation is pragmatic for research but raises practical questions for real-world deployment, particularly in "cold-start" scenarios where a model is being adapted to a new language without extensive prior user data.
>
>    >For cold-start deployment, a practical approach is to collect a few hundred representative "seed" examples through human curation. $E_t$ must span the anticipated semantic space of user queries, as COMPASS's distribution-aware sampling requires diversity across semantic clusters rather than volume within clusters. In the extreme case where no proxy is available, setting $E_t = D_t$ (using the training data itself as the usage proxy) provides a baseline. This reduces COMPASS to an approximately uniform-sampling regime in the initial phase, as cluster weights become nearly equal when train and eval distributions are identical ($\rho_k \approx 1$ for all clusters). While this eliminates distributional guidance initially, it does not harm performance relative to target-only training and allows the COMPASS-ECDA extension to refine $E_t$ over time based on observed usage. Alternatively, for languages within well-represented families and overlapping usage within defined geographic locales, usage distributions from related languages can serve as initial proxies (e.g., Spanish for Catalan within Spain), leveraging the multilingual embedding space's cross-lingual semantic similarities to borrow distributional knowledge from higher-resource relatives.
>
> **Change 2: “Promiscuous Language” finding**
>
> We thank the reviewer for this observation, which highlights an important limitation; we have elevated the “promiscuous language” finding from a passing note to an explicit discussion in the limitations section within the Discussion (page 30 last 2 paragraphs under the Limitations subsection). The new text reads:
>
>    >Our empirical analysis reveals a specific manifestation of this encoder dependency: the "promiscuous language" phenomenon where languages such as Malagasy, Malay, Tamil, Telugu, and Sinhala, which were not included in Jina3's primary tuning set, were sampled disproportionately across unrelated target languages. Consequently, examples from these under-tuned languages contaminate the auxiliary data selection for targets with which they share no linguistic or topical affinity.
>
>    >This finding demonstrates that COMPASS's sampling is systematically biased by encoder quality disparities across languages. When the encoder fails to capture precise semantic distinctions for certain languages, COMPASS's cluster-based selection inherits and amplifies these flaws. It also highlights the hidden cost of general-purpose multilingual encoders: even when a language is nominally "supported," insufficient tuning can render it harmful to the selection process. Encoder selection and evaluation beyond aggregate metrics (e.g., average performance on MMTEB) would serve as a proactive diagnostic of per-language embedding quality, which might be addressed by either developing encoder-aware sampling weights that down-weight languages with known embedding deficiencies, or investigating whether fine-tuning the embedding model on a subset of the auxiliary data pool could reduce promiscuity by improving representation quality for under-tuned languages.

---

### Decision · Action_Editor_qpWs · 2025-11-24

**Recommendation:** Accept with minor revision

**Additional Comments:**

The paper proposes COMPASS, a framework for adapting LLMs to target languages using parameter-efficient fine-tuning on a subset of auxiliary multi-lingual data. To address the challenge of distributional mismatch between training data and real-world usage, the authors introduce a distribution-aware sampling strategy. This strategy uses a pre-trained multilingual encoder to map training, auxiliary, and target usage data to the same embedding space. Embeddings are then clustered to identify semantic gaps between the training
and the target usage data. Auxiliary data is sampled to fill these gaps. The authors use the weight-decomposed low-rank adaptation (DoRA) method to fine-tune language-specific adapters. They also extend COMPASS to a continual learning framework to handle real-world distribution shifts,  which monitors these shifts via Jensen–Shannon divergence and updates adapters accordingly. Extensive experiments demonstrate that COMPASS outperforms baseline methods on various benchmarks.

Below is a summary of the main strengths and weaknesses highlighted by reviewers.

Strengths:
- Multilingual LLM adaptation is an important and timely topic.
- The experiments provided are thorough.
- The proposed method outperforms baseline methods across multiple models and benchmarks.
- Improvements generalize to unseen tasks and formats (short to long context).

Weaknesses:
- The proposed method is sensitive to the choice of encoder and clustering method
- The method requires a proxy for the live usage distribution. Though the authors added a discussion of practical strategies to handle cases when such a proxy is not available, as recommended by Reviewer DbVY.
- The data selection strategy could amplify biases present in the auxiliary data pool.
The authors properly acknowledged and discussed all these limitations in the paper.

All reviewers recommended to accept. The authors addressed in their responses all reviewers' concerns and edited the paper accordingly.

I am recommending to accept with the following minor revision:
I suggest changing the term "promiscuous" languages to a more neutral alternative such as "indiscriminately sampled" languages.

**Audience:**

Yes

**Audience Explanation:**

The paper addresses an important real-world problem in multi-language LLMs. It will be of interest to researchers working on this topic, or related ones like parameter-efficient fine-tuning, multi-task learning, and continual learning.

**Claims And Evidence:**

Yes

**Claims Explanation:**

The authors supported all their claims by extensive empirical evidence, as confirmed by all reviewers.

---

> ### Comment · Action_Editor_qpWs · 2025-12-04
> **Additional minor revision**
>
> Thank you for submitting the camera-ready version with the requested revision. I have an additional minor revision request: can you please increase the font size of the text in Figure 4.

---

> > ### Author Response · Authors · 2025-12-08
> >
> > We thank the action editor for the additional recommendation that helps improve the legibility of our manuscript. We have increased the text size for all text items in figure 4 and we have updated the color map to a perceptually uniform color bar.